# THE DIRECTIONALITY OF OPTIMIZATION TRAJECTORIES IN NEURAL NETWORKS

**Sidak Pal Singh**\*
ETH Zürich & MPI-IS Tübingen

**Bobby He**
ETH Zürich

**Thomas Hofmann**
ETH Zürich

**Bernhard Schölkopf**
MPI-IS Tübingen & ETH Zürich

## ABSTRACT

The regularity or implicit bias in neural network optimization has been typically studied via the parameter norms or the landscape curvature, often overlooking the trajectory leading to these parameters. However, properties of the trajectory — particularly its directionality — capture critical aspects of how gradient descent navigates the landscape to converge to a solution. In this work, we introduce the notion of a *Trajectory Map* and derive natural complexity measures that highlight the directional characteristics of optimization trajectories. Our comprehensive analysis across vision and language modeling tasks reveals that **(a)** the trajectory's directionality at the macro-level saturates by the initial phase of training, wherein weight decay and momentum play a crucial but understated role; and **(b)** in subsequent training, trajectory directionality manifests in micro-level behaviors, such as oscillations, for which we also provide a theoretical analysis. This implies that *neural optimization trajectories have, overall, a more linear form than zig-zaggy,* as evident by high directional similarity, especially towards the end. To further hone this point, we show that when the trajectory direction gathers such an inertia, optimization proceeds largely unaltered even if the network is severely decapacitated (by freezing $> 99\%$ of the parameters), — thereby demonstrating the potential for significant computational and resource savings without compromising performance.

## 1 INTRODUCTION

**Loss landscapes imply optimization trajectories, *but also vice versa*.** Given a network architecture and the training task, the loss landscape is the high-dimensional surface whose each point characterizes the fit of the parameters to the task objective. A priori, the loss landscape entails the possible trajectories (or paths) that might be followed by an optimization algorithm, such as stochastic gradient descent (SGD). However, the particular sets of optimization trajectories that are realized depend on the particular optimization choices and hyperparameters, such as the learning rate, momentum, batch size, weight decay, and more. Such is the importance of trajectory, that it could be argued the regions of the landscape that are never encountered or realized in typical optimization trajectories, might as well not be in the landscape at all. *Essentially, the possible optimization trajectories that are realized by an optimization method determine its effective loss landscape.*

**Implicit bias of optimization and what it means for trajectories.** Consequently, a significant body of literature builds around the principle of *implicit bias* facilitated by standard optimization algorithms (Gunasekar et al., 2018; Li et al., 2019; 2020; Moroshko et al., 2020; Barrett and Dherin, 2020), which loosely speaking refers to an undesigned and oftentimes beneficial simplicity that arises in optimized trajectories. This preferential access of realized landscapes should be contrasted with the established non-convexity of global neural network (NN) landscapes (Dauphin et al., 2014; Choromanska et al., 2015; Kawaguchi, 2016; Li et al., 2018), and suggests that implicit bias may lend a formal and reasonable support to the empirical success of massively over-parameterized NNs. As a result, one might expect to see signs and hallmarks of regularity in the sequence of steps that comprise realized NN loss landscape trajectories. In other words, the following questions, which form the basis of this work, naturally arise:

---

\*Correspondence to contact@sidakpal.com

> *What, if any, directional structure exists in neural trajectories? Do these paths have a lot of zigzags and bends, reaching the solution winding and coiling, or are they straight and direct? How do these properties evolve as optimization progresses?*

**Mapping the Trajectory Properties.** More precisely, we explore and develop key qualitative as well as quantitative indicators (hallmarks) about the directional complexity/regularity of the optimization trajectory. Towards this end, we analyze and compare multiple intermediate checkpoints amongst themselves, across different scenarios and large-scale case studies. A qualitative hallmark, which we call the *'Trajectory Map'*, conveys the directional (dis)similarity of the parameters and visually depicts the nature of optimization within and across various stages of training, i.e., at a pan-trajectory level. Our quantitative hallmarks are functions of these trajectory maps, measuring various notions of angles, over and above the sequence of steps in the trajectory.

**Benefits of the Trajectory View.** We believe the trajectory view has several advantages over other measures to study NN training landscapes, such as loss-based linear interpolation (Goodfellow et al., 2014; Frankle et al., 2020) or hidden representation based metrics (Kornblith et al., 2019; Nguyen et al., 2022; Lange et al., 2023). For example, it: (a) brings in a level of architecture agnosticity and helps unlock shared insights onto features of optimization, (b) contains an intrinsic independence to loss and data that necessitates no explicit inference over additional data samples, (c) allows analyzing and prognosing the developing solution strategy on-the-fly instead of waiting until convergence, and (d) provides potential hints at the bottlenecks and redundancies in the optimization procedure.

***Our contributions are*:**

**(1)** We propose the novel perspective of trajectory maps, and showcase how it uncovers new insights into the nature of directional evolution in parameter space during neural network training.

**(2)** We use this to uncover that optimization trajectories possess, in general, highly directional similarity, but simultaneously there reside oscillations at micro-levels — which we study theoretically.

**(3)** We show how model scale in LLMs has a regularizing effect on the directional complexity of trajectories, and that directional measures have a trend which is more predictive of performance than traditional norm-based complexity measures.

**(4)** We provide evidence that after the trajectory direction saturates, a significant amount of the network capacity can be frozen without compromising performance while making training efficient.

## 2 METHODOLOGY

**Matrix representation of Trajectory.** Let us assume the optimization trajectory consists of a set $\mathcal{T}$ of points $\{\boldsymbol{\theta}_t\}_{t=0}^T$, each denoting the (flattened) parameters of the network encountered at some step and which live in the parameter space $\boldsymbol{\theta} = \mathbb{R}^p$, i.e., $\mathcal{T} \subseteq \boldsymbol{\theta}$. This set of points need not contain the entire set of points visited in the course of optimization but instead can represent a subset of points, possibly sampled at an interval of $k$ points. It will be convenient to organize this set of points, which define the trajectory, in the form of a matrix, $\boldsymbol{\Theta} \in \mathbb{R}^{(T+1) \times p}$, whose first dimension $T + 1$ makes explicit the inclusion of the initialization $\boldsymbol{\theta}_0$.

**Trajectory Map** Analyzing the matrix $\boldsymbol{\Theta}$, on its own, might get cumbersome as the size of modern networks ranges in millions and billions of parameters. Hence, we will resort to looking at functions of the kernel matrix $\mathbf{K} = \boldsymbol{\Theta}\boldsymbol{\Theta}^\top$ which would be a square matrix of shape $n = T + 1$. Further, it will also be helpful to isolate and analyze the directional aspect of the trajectory, for which we will normalize the set of points by their norm, and in effect, consider the set $\widehat{\mathcal{T}} = \{\boldsymbol{\theta}_t/\|\boldsymbol{\theta}_t\|_2\}_{t=0}^T$ with the respective matrix $\widehat{\boldsymbol{\Theta}}$. As a result, the ensuing kernel matrix $\widehat{\boldsymbol{\Theta}}\widehat{\boldsymbol{\Theta}}^\top$, which we will refer to as $\mathbf{C}$ will contain the cosine similarities between every pair of points in the trajectory, i.e.,

$$(\mathbf{C})_{ij} = \text{cos-sim}(\boldsymbol{\theta}_i, \boldsymbol{\theta}_j) = \langle \boldsymbol{\theta}_i, \boldsymbol{\theta}_j \rangle / \|\boldsymbol{\theta}_i\|_2 \|\boldsymbol{\theta}_j\|_2 .$$

Hereafter, we will refer to $\mathbf{C}$ as the *Trajectory Map* (TM). We remark that, although not necessary, here we are essentially considering linear kernels, which we will see provide a variety of insights by themselves. The TM will be our qualitative hallmark of choice for analyzing optimization trajectories.

**Quantitative Hallmarks.** Besides visualizing the TM as the qualitative hallmark, we will consider the following set of indicators for quantitatively hallmarking the optimization trajectories.

*(i) Mean Directional Similarity (MDS):* We take the cosine similarity averaged over the entire trajectory map, i.e., over every pair of points in the trajectory. This can be written as, $\omega := \frac{1}{n^2} \mathbb{1}_n^\top \cdot \mathbf{C} \cdot \mathbb{1}_n$, where $\mathbb{1}_n^\top = (1 \cdots 1)^\top \in \mathbb{R}^{1 \times n}$ denotes the vector of all ones and $n = |\mathcal{T}|$ is the cardinality of the trajectory. By using the form of the matrix $\mathbf{C}$ discussed before, we can further rewrite MDS as, $\omega = \left\| \frac{1}{n} \widehat{\boldsymbol{\Theta}}^\top \mathbb{1}_n \right\|^2$. Now, it becomes apparent that MDS essentially projects all the trajectory points onto the unit sphere, computes their average and finally takes the squared norm.

To get a better sense of MDS, we can consider its two possible extremes: (a) all the parameter unit-vectors cancel out, yielding a value of $\omega = 0$. For instance, this would happen in the scenario when the points in the trajectory are exactly following a circular orbit around the origin; or, (b) when each of the parameters point in the same direction, implying that the trajectory is simply a linear path, with $\omega = 1$. Knowing the nature of these two extremes, we can expect neither to be desirable in an ideal trajectory which leads up to a generalizing solution. *Thus, hitting its sweet spot would be the target, or where that is unknown, at least avoiding these extremes.*

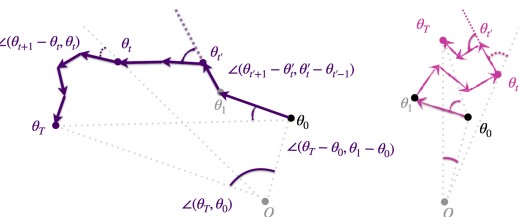

Figure 1: Illustration of two trajectories and angular measures.

(ii) *Angular Measures:* One such key measure would be the angle between consecutive (net) updates, i.e., $\angle(\boldsymbol{\theta}_{t+1} - \boldsymbol{\theta}_t, \boldsymbol{\theta}_t - \boldsymbol{\theta}_{t-1})$, which will measure the extent of directional movement along the trajectory. Next, we can consider a cone with the vertex at at origin and compute the apex angle there, $\angle(\boldsymbol{\theta}_t, \boldsymbol{\theta}_0)$. This measure will give us an idea of the amount of directional movement at a global scale.

(iii) *Norm-based Measures:* Besides, for baselines, we also include measures such as: parameter norms $\|\boldsymbol{\theta}_t\|_2$, distance from initialization $\|\boldsymbol{\theta}_t - \boldsymbol{\theta}_0\|_2$, norm between consecutive points $\|\boldsymbol{\theta}_{t+1} - \boldsymbol{\theta}_t\|_2$.

**Remark 1.** For extremely large neural networks, building the underlying kernel matrices can start becoming resource-expensive. In principle, there is a rich body of work in kernel methods that has focused on developing efficient approximations (Davis et al., 2014; Chen et al., 2021). We do not resort to such approximations for the sake of accuracy.

**Remark 2.** While beyond the scope of the main text, the appendix also catalogs results for other variants of angular/norm-based measures, as well as a relativized version of trajectory map (Section D).

## 3 GAINING INSIGHTS VIA TRAJECTORY EXEMPLARS

**Setup for obtaining the trajectory exemplars.** To begin with, in order to obtain a better grasp of the directionality of trajectories, let us contrast the trajectory map obtained with an optimal choice of hyperparameters against those with sub-optimal ones. As an exemplar of a desirable trajectory map, we employ the standard SGD-based training recipe[1] for ResNet50 on ImageNet that achieves a top-1 accuracy of $\sim 76\%$. To compare against sub-optimal trajectory maps, we analyze the effects of disabling[2] momentum and weight decay one by one, since empirically they lead to significant deterioration in performance. The resulting trajectory maps, built from checkpoints taken at each epoch, are shown in Figure 2.

**Observations.** We can make out that the *trajectory maps for the sub-optimal hyperparameter choices develop a darker hue much more quickly into training*, indicating a significantly higher directional similarity. To the extent that, when put on the same scale as in Figure 2, the rightmost (worst hyperparameter choice) trajectory map shows almost no directional variation as compared to the leftmost (optimal hyperparameter choice) map. Further, for the leftmost trajectory map, the

---

[1]Namely, this consists of training for 90 epochs with a learning rate $\eta = 0.1$ (decayed multiplicatively by a factor of 0.1 at epochs 30 and 60), momentum $\mu = 0.9$, batch size $B = 256$, and weight decay $\lambda = 0.0001$.

[2]In the Appendix F, we also present the directional effects of hyper-parameters such as learning rate and batch size, as well as recent regularizers like Sharpness-Aware Minimization (Foret et al., 2021).

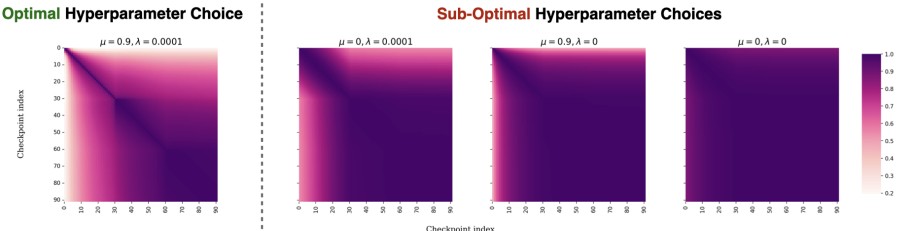

Figure 2: Trajectory Maps of ResNet50 trained on ImageNet with optimal and sub-optimal (in the order of disabling momentum $\mu = 0$, weight decay $\lambda = 0$, or both) hyperparameter choices. The MDS values are, $0.764, 0.901, 0.931, 0.979$ respectively. A darker hue represents higher cosine similarity.

directional movement shows a phase-wise trend[3], with its last phase $60 - 90$ epochs having a similar behaviour as the entire rightmost plot in the figure above. We emphasize that the onset of this dark hue doesn't mean the network has converged, as evident from Figure 3(b) which shows the trajectory length covered over time. Rather, only the *directional movement reaches a saturation by the end of the initial phase* (which is $\sim 30$ epochs) as shown in Figure 3(a).

*Qualitatively different solutions.* More than just the deterioration of the performance ($\sim 8\%$), we conclude that this toggling of the hyperparameters leads to solutions that are qualitatively different in terms of the directionality of their trajectories. In particular, the presence of weight decay or momentum encourages more directional exploration, while in their absence, the trajectory latches on to a solution which is nearby in a directional sense.

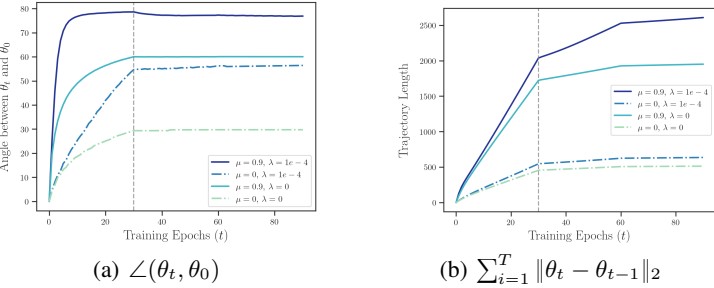

(a) $\angle(\theta_t, \theta_0)$      (b) $\sum_{i=1}^{T} \|\theta_t - \theta_{t-1}\|_2$

Figure 3: Evolution of the angle traced at origin by the trajectory and its path length over training across different momentum and weight decay settings. The left figure suggests that around epoch 30, marked by the gray dotted line, the directional movement saturates, while based on the right figure, we see that the trajectory still goes further. Set-up: ResNet50 on ImageNet.

**Directional Redundancy.** Another visible aspect from the trajectory maps is that, in general, neural optimization trajectories seem to possess a high mean directionality score (MDS), with $\omega = 0.764$ being the least amongst these exemplars. To contextualize this better, note that the parameters live in $\sim 25.6$ million dimensional space. This strongly suggests that network optimization trajectories do not merely comprise random points[4] in high-dimensions whose cosine similarity might go to zero, but rather the sequence of points encountered in a trajectory must be highly structured to yield such directionally redundant trajectories. In the Appendix F, we also present a wider set of results, namely, with adaptive optimizers like AdamW as well as more datasets/architectures such as Vision Tranformers on ImageNet and VGG16 on CIFAR10, and for different amounts of label noise — all of which support this finding of directional redundancy.

*A word of caution.* However, the above experiment also suggests that extreme values of MDS ($\geqslant 0.9$) should be avoided[5]. This can also be understood from the fact that trajectory maps here also comprise

---

[3]While 3 phases can be spotted distinctly, if we look closely, there seems to be another phase transition neighbouring the initialization and the subsequent couple epochs, giving rise to a thin horizontal and vertical sliver of relatively lighter hue. Empirically, we find in this transient phase, the distance to the solution rapidly decreases.

[4]In Appendix C, we analyze the relative trajectory map and MDS for a random walk/Brownian motion, and in comparison we find that (expectedly) the trajectory maps of neural networks are more directionally redundant.

[5]Besides, a certain amount of directional exploration is crucial as evident from our analysis of the grokking (Power et al., 2022) phenomenon discussed in Appendix I.

the random initialization, and hence a trajectory with the maximum possible MDS value of 1 would have as its end point a scaled version of the random initialization.

> **Section 3 Key Takeaways:**
>
> - Optimization trajectories have, overall, a high directional similarity (MDS > 0.75 across cases).
> - Disabling momentum and weight decay reduces the directional movement in the trajectories.
> - Directional movement saturates by the early phase, despite the trajectory going longer than that.

## 4    UNDERSTANDING DIRECTIONAL MOVEMENT

In this section, our aim is to investigate the nature of directional movement in further depth, and in particular, focus on the below two questions:

**Question 1:** *How exactly does the directional nature of trajectories manifest in the late phase of training*, when we know from Figure 3(a) that the overall directionality does not change significantly?

**Initial Thoughts.**  A tempting first guess is that perhaps the trajectories are just linear in this phase. But, if that were the case, we would be just using line-search (Zhang and Hager, 2004; Vaswani et al., 2019) to train neural networks and skip this otherwise slow second phase of training.

**Question 2:** *How do weight decay and momentum promote directional movement and lower MDS?*

**Initial Thoughts.**    The current results might even seem to run against the intuitive pictures of momentum and weight decay. For instance, intuitively, the use of momentum should strengthen the previous gradient directions and lead to increased directional similarity, but we find the opposite. Similarly, in the absence of weight decay, the network is not constrained to remain in a ball around the origin and, in principle, there should be more license to explore in the landscape.

*What are these intuitive impressions not taking into account?*

### 4.1    DIRECTIONAL MOVEMENT AT DIFFERENT LEVELS AND OSCILLATIONS

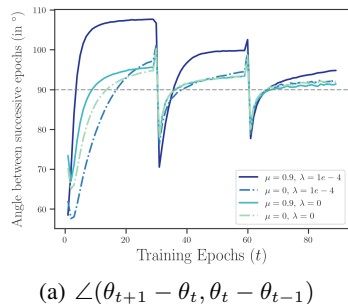 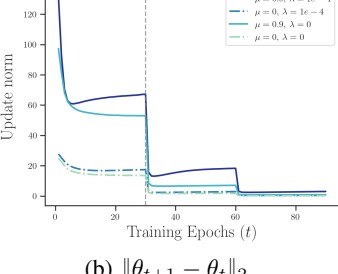

(a) $\angle(\theta_{t+1} - \theta_t, \theta_t - \theta_{t-1})$            (b) $\|\theta_{t+1} - \theta_t\|_2$

Figure 4: (a): Oscillatory nature of updates. (b): Beyond the first phase (after 30 epochs), the updates reside at a micro-scale with smaller update norms. The different curves show the effect of momentum and weight decay hyperparameters. The setup is same as before: ResNet50 on ImageNet.

**Answer 1.** We find that the late phase of training is *not so straightforward*. Far from co-linear updates, it is marked by significant noise and oscillation in the (stochastic) gradient descent updates. *The answer to this conundrum then lies in the fact that such noisy directional movement resides at a 'micro'-level but does not surface up to the 'macro'-level.*[6]

To see this, let us first take a general overview of the Figure 4, without looking at each of the individual curves. We see in Figure 4(a) that for a significant part of the training, the angle between successive update directions[7] is obtuse (gray horizontal line marks $90°$). Apart from short intervals

---

[6]We use this macro-micro distinction in terms of the angle traced at origin by the trajectory, see Figure 3(a).

[7]These updates should be thought of as 'net' or 'aggregate' updates, since they are at an per-epoch granularity for tractability reasons, and not at every single step taken by the optimizer. Empirically, the current granularity is still rich enough in that the presented trends persist even if we go $2\times$ to $5\times$ more coarser.

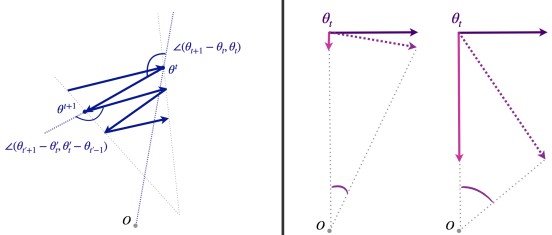

Figure 5: **(Left):** *Directional movement due to oscillations* results in obtuse angles; **(Right):** *Added directional movement due to weight decay.* Here, the downward vector represents the pull towards the origin (O) due to weight decay, while the rightward vector the force due to the loss.

near learning rate decay, this obtuse angle holds almost throughout the late phase. However, the extent to which this angle is obtuse decreases over the phases and further, Figure 4(b) reveals that beyond the early phase (marked by the vertical gray line), these oscillatory updates have much smaller norm.[8] Consequently, the late-phase oscillations reside at a micro-scale and therefore leave the macro-level directionality unaffected.

**Answer 2.** Interestingly, when we look closely at Figure 4(a), we find that *the angle between updates is more obtuse when momentum is turned on versus when off* (compare the solid lines versus dashed-dotted lines). Also, rather visibly, *the angle tends to be larger when weight decay is enabled* additionally (compare the solid **dark blue** and **turquoise** lines) , suggesting that weight decay and momentum are closely intertwined. Crucially, this observation points to how there is more direction movement when these hyperparameters are enabled and thereby decreasing MDS.

Furthermore, weight decay by itself has added directional effects, which we illustrate through a simple physics-based intuition in the Figure 5, right. In particular, we can think of the loss gradient pulling the network parameters rightwards, while the force exerted by weight decay tries to pull the network downwards. The joint presence of these forces lends more directionality to the trajectory, and is accentuated with increasing weight decay (downward force). See Appendices E.2 and E.4 for more.

## 4.2 THEORETICAL MODELLING OF THE UNDERLYING MECHANISM

To further understand the mechanism underlying the oscillatory nature and the intertwined role of momentum and weight decay, we turn to the simplest and oft-employed model of a quadratic problem.

**Lemma 1.** *Given a quadratic problem with $\ell_2$ regularization of strength $\alpha > 0$, namely, $\min_{\boldsymbol{\theta} \in \mathbb{R}^d} \frac{1}{2}\boldsymbol{\theta}^\top \mathbf{M}\boldsymbol{\theta} + \frac{1}{2}\alpha\|\boldsymbol{\theta}\|^2$ , with $\mathbf{M} \in \mathbb{R}^{d \times d}$ symmetric with eigenvalues $\lambda_1 \geqslant \cdots \geqslant \lambda_d$, the angle between successive steps $\Delta_t = \boldsymbol{\theta}_t - \boldsymbol{\theta}_{t-1}, \Delta_{t+1} = \boldsymbol{\theta}_{t+1} - \boldsymbol{\theta}_t$, when using gradient descent with a one-step momentum ($\mu > 0$) and learning rates $\eta_t, \eta_{t+1}$, can be upper and lower bounded as follows:*

$$\langle \Delta_t, \Delta_{t+1} \rangle \leqslant \eta_t \eta_{t+1}(1 - \eta_t(\mu + \alpha + \lambda_d))(\lambda_d + \alpha)^2 \|\boldsymbol{\theta}_{t-1}\|^2$$
$$\langle \Delta_t, \Delta_{t+1} \rangle \geqslant \eta_t \eta_{t+1}(1 - \eta_t(\mu + \alpha + \lambda_1))(\lambda_1 + \alpha)^2 \|\boldsymbol{\theta}_{t-1}\|^2$$

The proof, in Appendix A, inherently considers the solution at $\boldsymbol{\theta}^\star = \mathbf{0}$, but if that is not the case, we can substitute it in the objective and our derived bounds would scale in the squared distance to the solution, i.e. $\|\boldsymbol{\theta}_{t-1} - \boldsymbol{\theta}^\star\|^2$. Besides, in the above proof, we consider a one-step momentum, which inherently means resetting the momentum after every 2 steps. This is done for convenience, as our main purpose is to anyways gain insights into the phenomenon and not provide its ultimate proof.

Turning to the bounds themselves, notice that if the learning rate $\eta_t \geqslant 1/(\lambda_1 + \mu + \alpha)$, the lower bound will turn negative and will be multiplied by a factor of $(\lambda_1 + \alpha)^2 \|\boldsymbol{\theta}_{t-1}\|^2$. On the other hand, although the first term of the upper bound might still be positive, importantly, it is scaled by a factor of $(\lambda_d + \alpha)^2 \approx \alpha^2$ for matrices $\mathbf{M}$ which are close to degenerate ($\lambda_d \to 0$).

**Low-rank Hessian and Edge of Stability (EoS).** In NNs, the Hessian of the loss with respect to the parameters will play the role of the matrix $\mathbf{M}$, since we can assume a second-order Taylor series will hold across the two steps. But it is also known through prior empirical work that the Hessian is significantly degenerate (Sagun et al., 2017), which has also been proven rigorously for deep linear fully-connected and convolutional networks (Singh et al., 2021). Furthermore, this requirement on the learning rate $\eta_t$ is actually looser than the adaptivity of the largest eigenvalue of the Hessian to the learning rate $\lambda \approx 2/\eta$, as shown in the recent work on Edge of Stability (Cohen et al., 2022).

---

[8]The smaller update norms and the micro-level directionality in the late phase do not simply arise due to lower learning rates, but also are closely related to the loss of plasticity (Lyle et al., 2023; Dohare et al., 2024) .

**Explaining the Obtuse angles.** Owing to these facts, we will have that $\lambda_d(\mathbf{M}) \approx 0$, and which further implies that the upper bound on the inner-product between the updates will be approximately zero (as typically the regularization strength $\alpha$ is also small, e.g., $\alpha = 0.0001$ in the ResNet setting), and the lower-bound will be large in absolute value but negative. Therefore, this explains how the angles between consecutive epochs can be obtuse. More broadly, the obtuse angle indeed implies that there are oscillations, especially along the direction of the largest Hessian eigenvector. Further, from Lemma 1, we see that the magnitude of the inner-product of the updates scales in proportion to $\eta_{t+1}$. Hence, a way to dampen the oscillations is to decrease the learning rate, and as can be seen in Figure 4(a), the learning rate decay at epochs 30 and 60 is followed right after with the angles turning from obtuse to acute. Lastly, here in the constraint on the learning rate (the additive terms $\alpha$ and $\mu$), we can also see *momentum and weight decay go hand-in-hand, each accentuating the effect of the other.*

> Section 4 Key Takeaways:
>
> - Training close to EoS causes obtuse angles between updates, leading to directional exploration.
> - Weight decay and momentum are closely intertwined, and further enhance the oscillations.
> - The late-stage oscillations/bouncing reside at a micro-scale, as evident from smaller update norms (Figure 4(b)) and angles between consecutive epochs (Figure 4(a)).

## 5 LLMs AND DIRECTIONAL HALLMARKS OF TRAJECTORIES

Having better understood the nature of directional movement, we would like to know how generalizable are our findings. So, given the increasing relevance of Large Language Models (LLMs), in this section, we study whether language modelling tasks result in a similar trajectory structure as exhibited in vision tasks. Besides, in connection to LLMs, scaling comes up as a natural question. Hence, we would like to investigate how scale affects the directionality of trajectories: does it provide more directions for learning and increase the complexity of trajectories or if instead it has a regularizing effect?

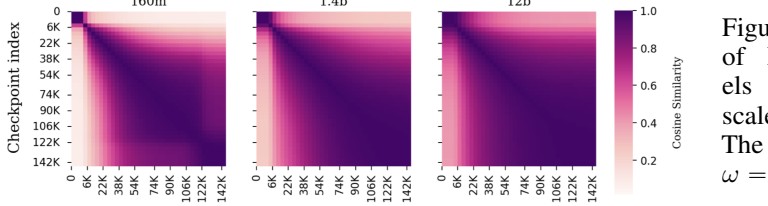

Figure 6: Trajectory Maps of Pythia GPT-NeoX models across two orders of scales trained on Pile. The corresponding MDS $\omega = 0.678, 0.759, 0.815$.

Thanks to Pythia's (Biderman et al., 2023) publicly released model checkpoints over training, for GPT-NeoX (Black et al., 2022) models — ranging in sizes from 14 Million (M) to 12 Billion (B) — we can provide answers to the above questions. For tractability, we select every fourth available checkpoint and consider models of sizes: 14M, 70M, 160M, 410M, 1.4B, 2.8B, 6.9B, 12B. The results for a shortlist of these experiments can be found in Figure 6 (for more, see Figure 69). First up, these *results establish that even the LLM trajectory maps have high directional redundancy across scales*, — suggesting a widespread applicability of our key finding.

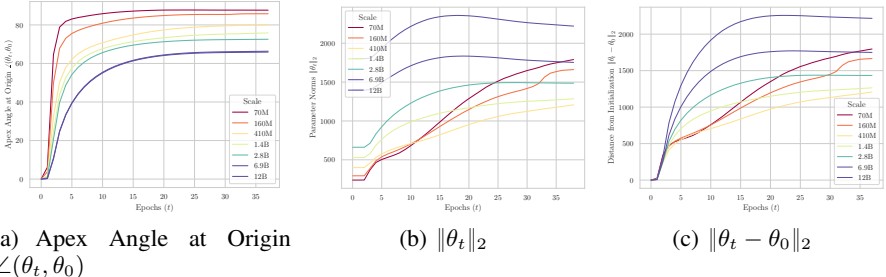

(a) Apex Angle at Origin $\angle(\theta_t, \theta_0)$

(b) $\|\theta_t\|_2$

(c) $\|\theta_t - \theta_0\|_2$

Figure 7: Directional complexity measures correlate better with performance than norm-based ones.

**Detailed Observations.** Elaborating further, in Figure 6, we find a tiny dark square grid in the upper-left corner, which delineates precisely the learning rate warmup phase. Next, the trajectory

map for the 160M model (and smaller) seems to have a discernible substructure, but which becomes more homogeneous with increasing model scale. In general, *increasing scale* lends an intense dark hue to the trajectory maps, suggesting *rising directional similarity*. Even the horizontal and vertical slivers (see the first few rows and columns) corresponding to the warmup phase start to assume a higher cosine similarity with scale. Overall, this seems to suggest that the larger models are biased towards directionally regular optimization trajectories.

**Directional measures correlate with performance better than norm-based measures.** In fact, Figure 7(a) shows that the apex angle made at the origin by the trajectory monotonically decreases with increasing scale. This is in contrast to norm-based complexity measures such as overall parameter norm (Gunasekar et al., 2018) and distance from initialization (Nagarajan and Kolter, 2019) which seem to be poorly correlated to performance, as visible in Figures 7(b) and 7(c). Norm-based measures fail to capture the monotonic trend, and even have the largest complexity value for the best-performing model.

**Why do parameters become aligned with scale? A Theoretical Argument.** We prove, in Appendix B, that this surprising finding about the progressive increase in cosine similarity with scale has a relatively simple explanation, at least in the case of the large-width limit of deep networks. The gist of our argument is that *in the large width limit, any parameter updates that lead to stable feature updates must necessarily yield updated parameters that are identically aligned with their initialisation.* While this is not so surprising for lazy learning regimes like the Neural Tangent Kernel (Jacot et al., 2020) where feature learning does not occur, *a more surprising aspect of our finding is that this is necessarily true also for feature learning regimes*, such as $\mu$P (Yang et al., 2022).

---

**Section 5 Key Takeaways:**

- LLMs up to 12B parameters exhibit like patterns of directionally redundant trajectory maps.
- Apex angle traced at origin monotonically decreases with increasing model scale, as opposed to traditional norm-based measures, thus showing promise as a useful complexity measure.

---

## 6 LEVERAGING DIRECTIONAL REDUNDANCY FOR EFFICIENT TRAINING

Our trajectory map analyses have revealed significant directional redundancy in the optimization trajectories of a broad range of neural networks, which is especially prominent later into training. While we have seen that the macro-level directional movement saturates early (Section 3), micro-level oscillations continue to persist in the late phases (Section 4). We thus hypothesize that *once this directional saturation is achieved, the full capacity of a network might not be required for training.* If true, this would motivate optimization hybrids for efficient training.

**How to best decapacitate the network?** As the primary desiderata, we want to (i) retain the least number of parameters possible, (ii) ensure that the capacity is removed in a structured manner to realize speed-ups (unlike, unstructured sparsity), (iii) intervene minimally in the implementation workflow. Clearly, our desiderata is stringent; but it allows testing the above hypothesis more compellingly.

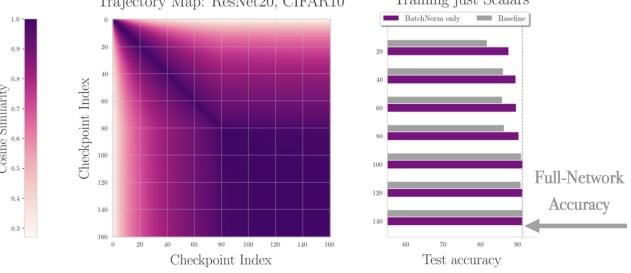

Figure 8: *Training only the BN scalar parameters suffices after the directional movement saturates.* The trajectory map of the original, full network, training is shown on the left and, in the bar chart to the right, the bars denoting the performance achieved by training BN parameters from a particular epoch. 'Baseline' denotes the network accuracy just before switching to optimizing the BN parameters only.

**Approach: Training only Normalization Layers.** This meets our desiderata since (i) normalization layers have typically $< 1\%$ of the parameters, (ii) all other layer types are frozen and do not need gradient updates, thus ensuring speed-up is realizable, (iii) this is essentially a $1$ *line change in code*. Also, (batch/layer/group) normalization layers are ubiquitous in modern architectures. Further, this re-purposing of normalization layers was inspired by Frankle et al. (2021) who empirically demonstrated the better-than-expected performance when training just the scalar parameters in batch-normalization (BN) layers from initialization, as well as theoretical expressivity results (Burkholz, 2024).

**Experimental Setup.** We experiment with ResNet20 on CIFAR10 using SGD over 160 epochs, freezing non-BN layers at different points in training, and then using only the $1,376$ scalar parameters in the BN layers for the remaining training duration. Figure 8 presents the results alongside the trajectory map of the full network training.

**Observations.** (i) Training just BN layers from initialization, as claimed in Frankle et al. (2021), results in a network with a test accuracy of $54.8\%$, which although interesting falls considerably short of the $91.2\%$ test accuracy obtained by training the entire network. In a way, from the horizontal strip of trajectory map around $0$, we see that at this stage the directional movement is yet to saturate. (ii) However soon after, from around $40$ epochs, where the trajectory map develops a dark hue (and the cosine similarity to the final parameters is $\sim 0.9$), training BN parameters alone leads to within $2\%$ of the full-network accuracy. (iii) From around $80$-th epoch, this gets to within $1\%$, and completely matches thereafter. Notably, this feat is *achieved by training only $0.5\%$ of the overall parameters,* and this fares even better than training all the parameters in the bulkier last layer as shown in Figure 77.

**Similar findings on ImageNet.** Likewise, in our experiments with ResNet50 on ImageNet as in Figure 2, we found that although training BN layers right from initialization gets a top-1 accuracy of just $\sim 6.4\%$, but this decapacitated training from epoch 30 gets to within $\sim 10\%$ of the full-network's accuracy and from epoch 60 to within about $\sim 2\%$ of it.This is again a striking result, considering only $0.18\%$ of the parameters are being trained.

**Benefits for Efficient Training.** Besides the *runtime saved in backward pass*, this also leads to savings in the *GPU memory consumption* as the optimization buffers (like the momentum buffer or those used for preconditioning in adaptive methods) now need to be of significantly smaller size (typically $< 1\%$, i.e., $99\%$ savings), which could, in turn, make *larger batch sizes feasible*.

**Broader Implications.** The above results pointedly demonstrate that *the above hypothesis holds through the strong desiderata fairly well*. More broadly, we envision that this proof-of-concept can be readily adapted into a hybrid optimization scheme, and possibly with more *flexible desiderata* that possibly allow for interleaved training with other parameters or equip more parameters to start with — thereby contributing to non-trivial cost savings. While our computational resources restrict us to vision models, we anticipate many of these insights will carry over to LLMs, given the observed similarities in their trajectory maps. We encourage the broader research community to explore the decapacitation idea as a means to optimize LLM training and spark further innovation in this respect.

---

**Section 6 Key Takeaways:**

- When directional movement saturates, neural networks can be severely decapacitated and trained only through parameters in normalization layers.
- Our results on CIFAR10 and ImageNet show that this comes with little compromise in performance, while making training much more efficient.

---

# 7 RELATED WORK

**Directional Redundancy and Trajectory Structure.** Prior work (Ji and Telgarsky, 2020) has theoretically noted a notion of directional convergence, wherein the parameters of simple networks and classifiers converge quickly, in terms of their direction. Likewise (Merrill et al., 2020) have observed that cosine similarity between subsequent parameter checkpoints during T5 (Raffel et al., 2023) pre-training rapidly approaches one. In a similar vein, Guille-Escuret et al. (2023) find that gradients tend to have a moderate but positive alignment with the solution direction, all throughout training. Further, in a recent work (Dherin and Rosca, 2024), propose the existence of regions called 'corridors' in (full-batch) gradient descent training, whereby the trajectory behaves like line segments.

All of these can be seen as diverse facets of the core phenomenon of redundancy in the directional movement, which we elaborate in depth here. In comparison, our work lays out a comprehensive analysis of the directional aspects of trajectories, which we study through trajectory maps and related directional measures, and where we uncover different levels of directional movement and their mechanisms — thus contributing significant nuance over existing literature.

**Implicit Effects of Hyperparameters.** Implicit bias (Gunasekar et al., 2018; Li et al., 2019; 2020; Moroshko et al., 2020; Barrett and Dherin, 2020) has emerged as one of the main contenders for explaining the success of deep neural networks. This principle has also inspired several works which seek to uncover the implicit effects that might be latent in the regular working of hyperparameters. To name a few, Andriushchenko et al. (2023) for instance suggest a loss stabilization mechanism behind weight decay, while (Liu et al., 2023) attempt to characterize the implicit bias of large learning rates in terms of resulting in a flatter solution, and Jelassi and Li (2022); Cao et al. (2023) explain the implicit bias of momentum and batch normalization with regards to margin. In contrast, we showed the implicit effects of momentum and weight decay from a trajectory perspective, i.e., on the increased directional movement, as well as how these two hyperparameters interact with each other.

**Decapacitation.** This might be reminiscent of the lottery-ticket hypothesis (Frankle and Carbin, 2019), with a subtle but key distinction that the lottery-tickets are concerned with sub-networks (certain neurons are set to zero) *while here the network structure remains the same and the capacity is stripped by freezing the parameters*. Also, the mechanism of finding lottery-tickets is retrospective in nature; while here we present a mechanism to decapacitate that depends on the current network parameters. However, it would be worth investigating the connection further, especially given their shared reliance on the early phase of training — albeit through directional saturation versus linear mode connectivity (Frankle et al., 2020). Besides, a similar exploration of tuning normalization parameters has been tried for fine-tuning (Zhao et al., 2023), although not in the context of (pre-)training which is our focus here.

## 8    DISCUSSION

**Summary.** We observe that optimization trajectories in neural networks exhibit a rich directional structure, across a wide range of architectures, datasets, optimizers, and hyperparameters. This is firstly visible through the trajectory maps, which generally have high mean dimensionality scores. Next, as the angular measures reveal, the directional movement persists throughout training, but resides at varying levels, with the macro-level behaviour in the early training phase and micro-level oscillations subsequently. We utilize this to come up with and demonstrate network decapacitation, whereby once the directional saturation is reached, freezing significant network capacity leaves the performance unaltered — thus showcasing the potential for efficient training of neural networks.

**Future Work.** The two most interesting directions of future work are as follows:

*(A). Testing the Generalization Potential of Directional Measures.* We have seen in Sections 3 and 5, that directional measures are correlated with generalization. In particular, they appear to exhibit a U-shape trend, where the extreme values should clearly be avoided. A dedicated study focusing exclusively on the generalization potential of directional measures, and comparing these findings to flatness-based analyses, would be highly insightful. Intuitively, flatter solutions will contain several low curvature directions in their vicinity, which would align well with the discussion here.

*(B). Analysis of Layerwise dynamics.* Another interesting aspect of trajectory maps is that they can be easily extended to hone into directional dynamics at a layerwise level. Our initial analysis into GPTNeoX models discussed in Section 5 shows that the layerwise trajectory maps inhabit a rich structure, as shown in Figure 75. In particular, for models with less than a billion parameters, the Q,K,V trajectory maps show significant heterogeneity in the timescales of directional convergence, with middle layers converging the last directionally. Strikingly, the Q,K,V dynamics homogenize across depth with higher scale. Moreover, these intriguing observations suggest utilizing trajectory maps for (data-free) mechanistic understanding, complementing  (Elhage et al., 2021; Grosse et al., 2023).

**Conclusion.** Overall, we have merely scratched the surface of this trajectory perspective into understanding optimization behaviour in neural networks. We hope this work will bring further nuance in these areas and contribute towards hybrid optimization schemes that can exploit the showcased directional redundancy.

## ACKNOWLEDGMENTS

We would like to thank Benoit Dherin and Tiago Pimentel for reading a draft of the paper and providing useful comments. Also, we thank Weronika Ormaniec for catching typographical mistakes and making useful remarks. Sidak Pal Singh would also like to acknowledge the financial support from Max Planck ETH Center for Learning Systems.

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

# Part I

# Appendix

## Table of Contents

## A OMITTED PROOFS

**Lemma 2.** *Given a quadratic problem with $\ell_2$ regularization of strength $\alpha > 0$, namely, $\min_{\boldsymbol{\theta} \in \mathbb{R}^d} \frac{1}{2}\boldsymbol{\theta}^\top \mathbf{M}\boldsymbol{\theta} + \frac{1}{2}\alpha\|\boldsymbol{\theta}\|^2$, with $\mathbf{M} \in \mathbb{R}^{d \times d}$ symmetric with eigenvalues $\lambda_1 \geqslant \cdots \geqslant \lambda_d$, the angle between successive steps $\Delta_t = \boldsymbol{\theta}_t - \boldsymbol{\theta}_{t-1}, \Delta_{t+1} = \boldsymbol{\theta}_{t+1} - \boldsymbol{\theta}_t$, when using gradient descent with a one-step momentum ($\mu > 0$) and learning rates $\eta_t, \eta_{t+1}$, can be upper and lower bounded as follows:*

$$\langle \Delta_t, \Delta_{t+1} \rangle \leqslant \eta_t \eta_{t+1}(1 - \eta_t(\mu + \alpha + \lambda_d))(\lambda_d + \alpha)^2 \|\boldsymbol{\theta}_{t-1}\|^2$$
$$\langle \Delta_t, \Delta_{t+1} \rangle \geqslant \eta_t \eta_{t+1}(1 - \eta_t(\mu + \alpha + \lambda_1))(\lambda_1 + \alpha)^2 \|\boldsymbol{\theta}_{t-1}\|^2$$

*Proof.* Given function $f(\boldsymbol{\theta}) = \frac{1}{2}\boldsymbol{\theta}^\top \mathbf{M}\boldsymbol{\theta} + \frac{1}{2}\alpha\|\boldsymbol{\theta}\|^2$, the gradient at $\boldsymbol{\theta}$ will be $\nabla f(\boldsymbol{\theta}) = (\mathbf{M} + \alpha\mathbf{I})\boldsymbol{\theta}$. Then at the first optimization step, we do

$$\boldsymbol{\theta}_t = \boldsymbol{\theta}_{t-1} - \eta_t(\mathbf{M} + \alpha\mathbf{I})\boldsymbol{\theta}_{t-1}$$

The particular update being $\Delta_t := \boldsymbol{\theta}_t - \boldsymbol{\theta}_{t-1} = -\eta_t(\mathbf{M} + \alpha\mathbf{I})\boldsymbol{\theta}_{t-1}$. The next update is similar, but now we also have to factor in the momentum,

$$\boldsymbol{\theta}_{t+1} = \boldsymbol{\theta}_t - \eta_{t+1}\left(\nabla f(\boldsymbol{\theta}_t) - \mu\eta_t(\mathbf{M} + \alpha\mathbf{I})\boldsymbol{\theta}_{t-1}\right)$$

$$\begin{aligned}
\Delta_{t+1} := \boldsymbol{\theta}_{t+1} - \boldsymbol{\theta}_t &= -\eta_{t+1}\left((\mathbf{M} + \alpha\mathbf{I})\boldsymbol{\theta}_t - \mu\eta_t(\mathbf{M} + \alpha\mathbf{I})\boldsymbol{\theta}_{t-1}\right) \\
&= -\eta_{t+1}\left((\mathbf{M} + \alpha\mathbf{I})\boldsymbol{\theta}_{t-1} - \eta_t(\mathbf{M} + \alpha\mathbf{I})^2\theta_{t-1} - \mu\eta_t(\mathbf{M} + \alpha\mathbf{I})\boldsymbol{\theta}_{t-1}\right) \\
&= -\eta_{t+1}\left((1 - \mu\eta_t - \alpha\eta_t)\mathbf{I} - \eta_t\mathbf{M}\right)(\mathbf{M} + \alpha\mathbf{I})\boldsymbol{\theta}_{t-1}
\end{aligned}$$

Now, let us evaluate the inner-product $\langle \Delta_t, \Delta_{t+1} \rangle$,

$$\langle \Delta_t, \Delta_{t+1} \rangle = \eta_t \eta_{t+1}\boldsymbol{\theta}_{t-1}^\top \underbrace{(\mathbf{M} + \alpha\mathbf{I})\left((1 - \mu\eta_t - \eta_t\alpha)\mathbf{I} - \eta_t\mathbf{M}\right)(\mathbf{M} + \alpha\mathbf{I})}_{\mathbf{Z}}\boldsymbol{\theta}_{t-1}$$

Now without loss of generality we can consider $\mathbf{Z}$ to be a diagonal matrix, as $\mathbf{Z}$ is symmetric since $\mathbf{M}$ is symmetric, we can consider its spectral decomposition $\mathbf{Z} = \mathbf{U}\mathbf{D}\mathbf{U}^\top$ and project $\boldsymbol{\theta}_{t-1}$ onto its eigenvectors contained in $\mathbf{U}$. With this the matrices in the middle are diagonal and we can commute them, which yields us the following matrix:

$$\mathbf{Z} = \mathrm{diag}\begin{pmatrix} (1 - \mu\eta_t - \eta_t\alpha - \eta_t\lambda_1)(\lambda_1 + \alpha)^2 \\ \vdots \\ (1 - \mu\eta_t - \eta_t\alpha - \eta_t\lambda_d)(\lambda_d + \alpha)^2 \end{pmatrix}$$

where, we have denoted the eigenvalues of $\mathbf{M}$ as $\lambda_1 \geqslant \cdots \geqslant \lambda_d$.

Since the inner product of the updates is a quadratic form, we can upper and lower bound it based on the maximum and minimum eigenvalues of $\mathbf{Z}$, thus giving:

$$\eta_t \eta_{t+1}\lambda_{\min}(\mathbf{Z})\|\boldsymbol{\theta}_{t-1}\|^2 \leqslant \langle \Delta_t, \Delta_{t+1} \rangle \leqslant \eta_t \eta_{t+1}\lambda_{\max}(\mathbf{Z})\|\boldsymbol{\theta}_{t-1}\|^2$$

Because of the above form of eigenvalues of $\mathbf{Z}$ (diagonal matrices have their eigenvalues as their diagonal entries), we will have:

$\lambda_{\max}(\mathbf{Z}) = (1 - \mu\eta_t - \eta_t\alpha - \eta_t\lambda_d)(\lambda_d + \alpha)^2$ and $\lambda_{\min}(\mathbf{Z}) = (1 - \mu\eta_t - \eta_t\alpha - \eta_t\lambda_1)(\lambda_1 + \alpha)^2$ $\quad\square$

# B  WHY COSINE SIMILARITIES INCREASE WITH SCALE?

Note, we assume that the majority of the parameter norm lies in the square hidden matrices, and not the input or output layers. Moreover, we use $o, O, \theta$ to denote standard mathematical notation with regards to scaling in the limit width $n \to \infty$. For vectors, this notation is entry-wise.

Suppose we have a hidden layer with input $x_0 \in \mathbb{R}^n$ for width $n$, that is acted on by (without loss of generality) a square matrix $W_0 \in \mathbb{R}^{n \times n}$ to give:

$$h_0 = W_0 x_0$$

We suppose $x$ has $\theta(1)$ entries, as is the case with standard initialisations/parameterisations He et al. (2015). We suppose $W_0$ has i.i.d. elements with initialisation that is $O(1/\sqrt{n})$ in order to ensure that each element of the features $h$ has entries $\theta(1)$.

Now, if we take a gradient update with learning rate $\eta$ on some downstream loss $L$ that depends on $h$ (and not $W$ or $x$), we get:

$$W_1 = W_0 - \eta \mathrm{d}h \cdot x_0^\top$$

where $\mathrm{d}h = \frac{\partial L}{\partial h} \in \mathbb{R}^{n \times 1}$ is our feature derivative.

Then if we have new input $x_1$ (wlog $x_1 = x_0$), we have new features:

$$h_1 = x_1 W_1 = h_0 - n\eta \mathrm{d}h \cdot \frac{x_0^\top x_0}{n}$$

For our features to be stable (i.e. $\theta(1)$) after the update, we need $n\eta \mathrm{d}h$ to be $O(1)$, because $\frac{x_0^\top x_0}{n} = \theta(1)$ by assumption on $x$. NB: if $n\eta \mathrm{d}h = o(1)$ we have no feature learning (ie NTK regime Jacot et al. (2018)), and if $n\eta \mathrm{d}h = \theta(1)$ we have feature learning (ie $\mu$P Yang et al. (2022)).

In any case, $\eta \mathrm{d}h = O(1/n)$ entry-wise, which means that $W_1 - W_0 = -\eta \mathrm{d}h \cdot x_0^\top$ has $O(1/n)$ entries, again by assumption on the scale of elements of $x_0$.

But because $W_0 = \theta(1/\sqrt{n})$, the initialisation will elementwise-dominate the $O(1/n)$ update for the first training step (and more training steps follows by induction). As a result, the update $W_T - W_0$ will always be an order of at least $\sqrt{n}$ smaller than the initialisation, and hence the new parameters $W_T$ will be exactly aligned with the initialisation $W_0$ for all $T$ in the large width limit, i.e. the cosine similarities will be 1.

## C  COMPARING GRADIENT TRAJECTORIES WITH RANDOM WALKS

The structure that we observe in trajectory maps following gradient trajectories raises the question of if we would observe similar structure in a random walk.

If we have $T$ timesteps or epochs, with parameter space $\boldsymbol{\theta} \in \mathbb{R}^p$ and "learning rate" schedule $(\eta_t)_{t=1}^T$, we can consider a random walk with updates:

$$\boldsymbol{\theta}_t - \boldsymbol{\theta}_{t-1} \overset{\text{ind.}}{\sim} \mathcal{N}(0, \eta_t^2 I_p)$$

which is to say that at time step $t$, each parameter coordinate in the parameter vector is updated independently with a Gaussian of variance $\eta_t^2$, and the updates are independent across different time steps.

Then, if $\theta^i$ denotes a *single* parameter coordinate for a dimension $i \leqslant p$, for two time steps $s < t$, we have:

$$(\theta_s^i, \theta_t^i) \sim \mathcal{N}(0, \left( \begin{smallmatrix} H_s & H_s \\ H_s & H_t \end{smallmatrix} \right))$$

where $H_u = \sum_{t'=1}^u \eta_{t'}^2$ is the cumulative squared learning rate from $t' = 1$ to $t' = u$.

Then, by the strong law of large numbers we have for the large parameter space $p \to \infty$ limit:

$$\frac{1}{p}\|\boldsymbol{\theta}_s\|_2^2 = \frac{1}{p}\sum_{i=1}^p (\theta_s^i)^2 \overset{a.s.}{\to} H_s \,, \quad \frac{1}{p}\|\boldsymbol{\theta}_t\|_2^2 = \frac{1}{p}\sum_{i=1}^p (\theta_t^i)^2 \overset{a.s.}{\to} H_t$$

$$\frac{1}{p}\langle \boldsymbol{\theta}_s, \boldsymbol{\theta}_t \rangle = \frac{1}{p}\sum_{i=1}^p \theta_t^i \theta_s^i \overset{a.s.}{\to} H_s$$

and by the property of composing almost sure limits, we also have almost sure convergence in the cosine similarity:

$$\frac{\langle \boldsymbol{\theta}_s, \boldsymbol{\theta}_t \rangle}{\|\boldsymbol{\theta}_s\|_2 \|\boldsymbol{\theta}_t\|_2} \overset{a.s.}{\to} \frac{H_s}{\sqrt{H_s H_t}} = \sqrt{\frac{H_s}{H_t}}$$

in the large parameter space limit, which we use as an approximation to give analytic formulas for the trajectory map and MDS that we can compare to gradient trajectories.

One thing to note, is that this cosine similarity $\sqrt{\frac{H_s}{H_t}}$ becomes invariant to the *scale* of the learning rates $\eta$, and instead it is the relative rate of decay in the learning rate schedule that matters.

### C.1  EXPERIMENTAL RESULTS

The first thing to note is that our empirical simulation of the random walk matches the theoretical limit described in the section above, for a finite parameter count (such as $10,000$). Next, comparing the these relative trajectory maps with those for ResNet50 (Figure 10) we find that the latter reveal a much more directional redundancy component to their trajectories as opposed to random walks. This further lends support to the thesis that optimization trajectories ensued when training neural networks are highly structured and have significant directional redundancy.

An additional thing to note is that the above relative trajectory map for random walks covers the setting of decreasing the step size to mirror how the optimization procedure is setup for ResNet50. In the case of no such step size decay, the analytic and empirical versions of the relative trajectory map are depicted in the Figure 12.

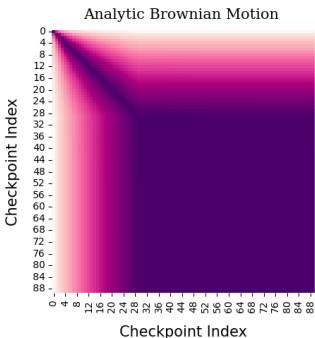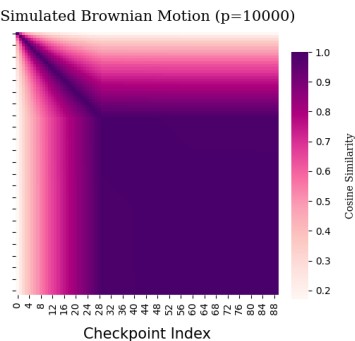

Figure 9: With step size decay: Relative Trajectory Map for a Random Walk/Brownian motion in both analytic and empirically simulated settings.

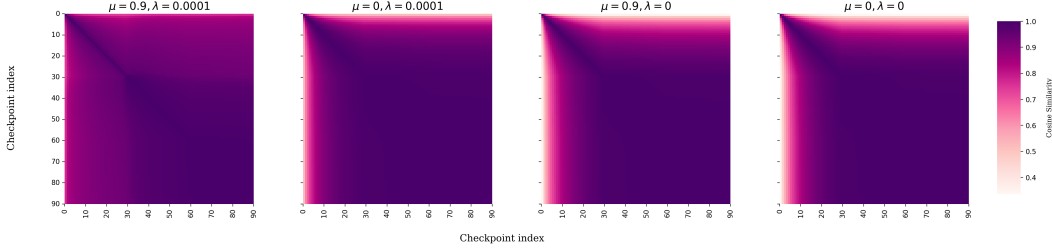

Figure 10: Relative Trajectory Maps, with respect to initialization, of ResNet50 models for different amounts of momentum and weight decay.

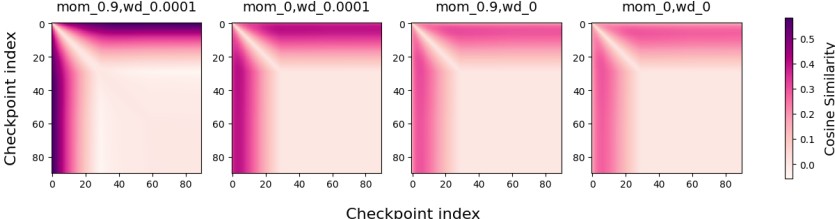

Figure 11: Relative Trajectory Maps, with respect to initialization, of ResNet50 models when subtracting the trajectory map of a Random Walk (shown in Figure 9) with a corresponding learning rate schedule. As before, this is carried out for different amounts of momentum and weight decay.

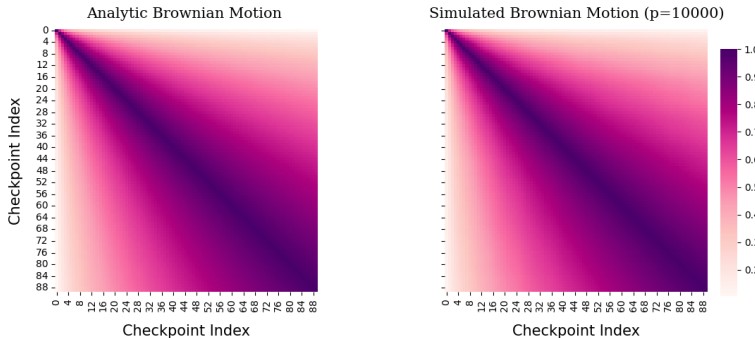

Figure 12: *No Step size decay*: Relative Trajectory Map for a Random Walk/Brownian motion in both analytic and empirically simulated settings.

## C.2 EXPECTED MDS VALUES FOR RANDOM WALK

Effectively, to get the MDS, we need to compute the expected mean entry of the matrix whose $(s, t)$ entry contains the cosine similarity between $\boldsymbol{\theta}_s$ and $\boldsymbol{\theta}_t$ in the described random walk, we'll start by noting the following:

In the large parameter space limit ($p \to \infty$), the cosine similarity between $\boldsymbol{\theta}_s$ and $\boldsymbol{\theta}_t$ converges almost surely to

$$\cos \theta_{s,t} = \sqrt{\frac{\min(H_s, H_t)}{\max(H_s, H_t)}}$$

where $H_u = \sum_{t'=1}^{u} \eta_{t'}^2$ is the cumulative squared learning rate up to time $u$.

For simplicity, let's consider the case where the learning rate is constant: $\eta_t = \eta$ for all $t$. Then, $H_t = t\eta^2$, and the cosine similarity simplifies to

$$\cos \theta_{s,t} = \sqrt{\frac{\min(s, t)}{\max(s, t)}}$$

The expected mean entry of the matrix is the average of all these cosine similarities:

$$\text{Mean} = \frac{1}{T^2} \sum_{s=1}^{T} \sum_{t=1}^{T} \sqrt{\frac{\min(s, t)}{\max(s, t)}}$$

Due to the symmetry of the cosine similarity, we can write this as:

$$\text{Mean} = \frac{2}{T^2} \sum_{s=1}^{T-1} \sum_{t=s+1}^{T} \sqrt{\frac{s}{t}} + \frac{1}{T^2} \sum_{s=1}^{T} \sqrt{\frac{s}{s}}$$

$$\text{Mean} = \frac{2}{T^2} S + \frac{1}{T}$$

where

$$S = \sum_{s=1}^{T-1} \sum_{t=s+1}^{T} \sqrt{\frac{s}{t}}$$

To compute $S$, we approximate the sum by an integral for large $T$:

$$S \approx \int_{s=1}^{T} \int_{t=s}^{T} \sqrt{\frac{s}{t}} \, dt \, ds$$

Calculating the integral:

$$I(s) = \int_{t=s}^{T} \sqrt{\frac{s}{t}} \, dt = 2s^{1/2}(T^{1/2} - s^{1/2})$$

$$S = \int_{s=1}^{T} 2s^{1/2}(T^{1/2} - s^{1/2}) \, ds = 2T^{1/2} \int_{s=1}^{T} s^{1/2} \, ds - 2 \int_{s=1}^{T} s \, ds$$

Computing the integrals:

$$\int_{s=1}^{T} s^{1/2} \, ds = \frac{2}{3}(T^{3/2} - 1), \quad \int_{s=1}^{T} s \, ds = \frac{1}{2}(T^2 - 1)$$

Plugging back:

$$S = 2T^{1/2} \left( \frac{2}{3}(T^{3/2} - 1) \right) - 2 \left( \frac{1}{2}(T^2 - 1) \right)$$

$$S = \frac{4}{3}T^{1/2}(T^{3/2} - 1) - (T^2 - 1)$$

$$S = \frac{4}{3}(T^2 - T^{1/2}) - (T^2 - 1)$$

$$S = \left(\frac{4}{3}T^2 - T^2\right) - \frac{4}{3}T^{1/2} + 1$$

$$S = \frac{1}{3}T^2 - \frac{4}{3}T^{1/2} + 1$$

Now, under the large T assumption, we have as the MDS:

$$\text{MDS} = \frac{2}{T^2}S + \frac{1}{T} \approx \frac{2}{T^2}\left(\frac{1}{3}T^2 - \frac{4}{3}T^{1/2} + 1\right) + \frac{1}{T}$$

$$\text{MDS} \approx \frac{2}{3} - \frac{8}{3}\frac{1}{T^{3/2}} + \frac{2}{T^2} + \frac{1}{T} \tag{1}$$

As $T \to \infty$, the terms involving $\frac{1}{T}$, $\frac{1}{T^{3/2}}$, and $\frac{1}{T^2}$ vanish, leaving:

$$\text{MDS} \approx \frac{2}{3}$$

## D  RELATIVE TRAJECTORY MAPS

In some cases, it might be useful to analyze the trajectory relative to some point $\boldsymbol{\theta}_\tau$ as the origin, so there we will instead consider the set of points $\mathcal{T}_\tau = \{\boldsymbol{\theta}_t - \boldsymbol{\theta}_\tau\}_{t=0}^T$, and correspondingly organize it in the matrix $\boldsymbol{\Theta}_\tau \in \mathbb{R}^{(T+1)\times p}$. When $\tau$ is itself one of the points of the trajectory, then we will omit the row of zeros and shape the matrix as $\mathbb{R}^{T\times p}$. A natural point from where to contextualize the trajectory would be the initialization $\boldsymbol{\theta}_0$, and this relative trajectory will then be denoted as $\boldsymbol{\Theta}_0$ (where the subscript 0 is not to be confused for the usual origin $O$, namely, $\boldsymbol{\theta}_O = \mathbf{0}$).

**Trajectory Map.** Analyzing the matrix $\boldsymbol{\Theta}$ or $\boldsymbol{\Theta}_\tau$, on its own, might get cumbersome as the size of modern networks ranges in millions and billions of parameters. Hence, we will resort to looking at functions of the kernel matrix $\mathbf{K} = \boldsymbol{\Theta}\boldsymbol{\Theta}^\top$ which would be a square matrix of shape $n = T + 1$, or for $\tau \neq O$, the relative kernel matrix $\mathbf{K}_\tau = \boldsymbol{\Theta}_\tau\boldsymbol{\Theta}_\tau^\top$ of shape $n = T + 1$ or $n = T$ depending if the point $\boldsymbol{\theta}_\tau$ is a part of the trajectory or not. Further, it will also be helpful to isolate and analyze the directional aspect of the trajectory, for which we will normalize the set of points by their norm, and in effect, consider the set $\widehat{\mathcal{T}}_\tau = \{\frac{\boldsymbol{\theta}_t - \boldsymbol{\theta}_\tau}{\|\boldsymbol{\theta}_t - \boldsymbol{\theta}_\tau\|_2}\}_{t=0}^T$ with the respective matrix $\widehat{\boldsymbol{\Theta}}_\tau$. As a result, the ensuing kernel matrix $\widehat{\boldsymbol{\Theta}}_\tau\widehat{\boldsymbol{\Theta}}_\tau^\top$, which we will refer to as $\mathbf{C}_\tau$ (or $\mathbf{C} := \mathbf{C}_O$ for the usual origin $\tau = O$), will contain the relative cosine similarities between every pair of points in the trajectory. So, $(\mathbf{C}_\tau)_{ij}$ is,

$$\text{cos-sim}(\boldsymbol{\theta}_i - \boldsymbol{\theta}_\tau, \boldsymbol{\theta}_j - \boldsymbol{\theta}_\tau) = \frac{\langle \boldsymbol{\theta}_i - \boldsymbol{\theta}_\tau, \boldsymbol{\theta}_j - \boldsymbol{\theta}_\tau \rangle}{\|\boldsymbol{\theta}_i - \boldsymbol{\theta}_\tau\|_2 \|\boldsymbol{\theta}_j - \boldsymbol{\theta}_\tau\|_2}.$$

We will refer to $\mathbf{C}$ as the *Trajectory Map* (TM) and $\mathbf{C}_\tau$ (for $\tau \neq O$) as the *Relative Trajectory Map* (RTM).

## E  DETAILED EXPERIMENTAL RESULTS

### E.1  RESNET50: SWITCHING OFF THE HYPERPARAMETERS

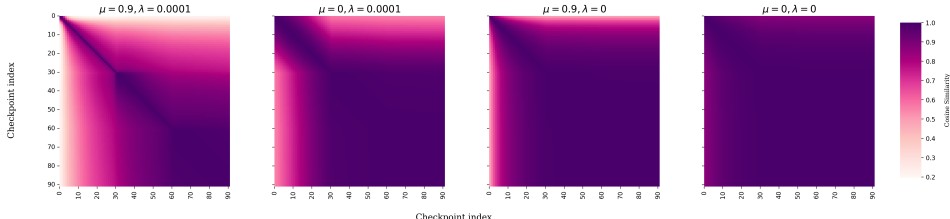

Figure 13: Trajectory Maps of ResNet50 models across different amounts of momentum and weight decay

The relative trajectory maps can be found in Figure 10.

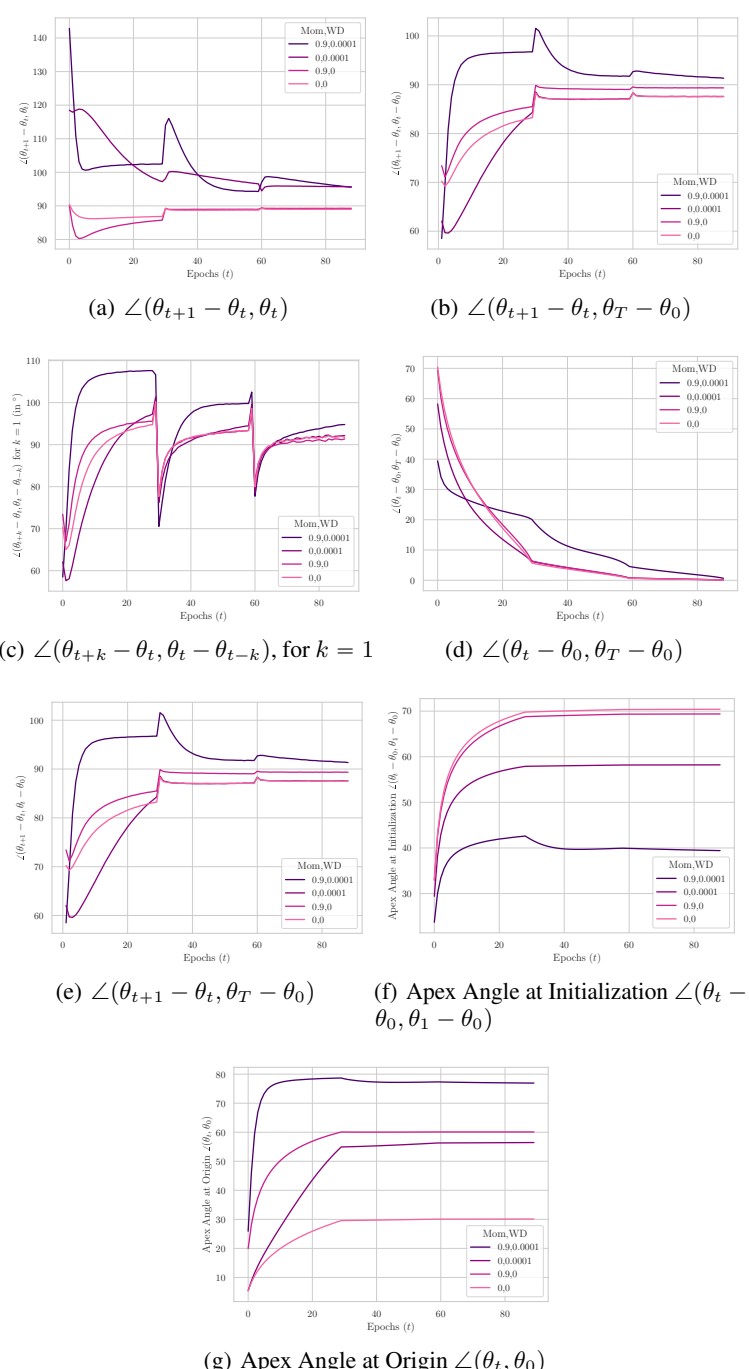

(a) $\angle(\theta_{t+1} - \theta_t, \theta_t)$

(b) $\angle(\theta_{t+1} - \theta_t, \theta_T - \theta_0)$

(c) $\angle(\theta_{t+k} - \theta_t, \theta_t - \theta_{t-k})$, for $k = 1$

(d) $\angle(\theta_t - \theta_0, \theta_T - \theta_0)$

(e) $\angle(\theta_{t+1} - \theta_t, \theta_T - \theta_0)$

(f) Apex Angle at Initialization $\angle(\theta_t - \theta_0, \theta_1 - \theta_0)$

(g) Apex Angle at Origin $\angle(\theta_t, \theta_0)$

Figure 14: Angular measures of the Trajectory for ResNet50 trained on ImageNet

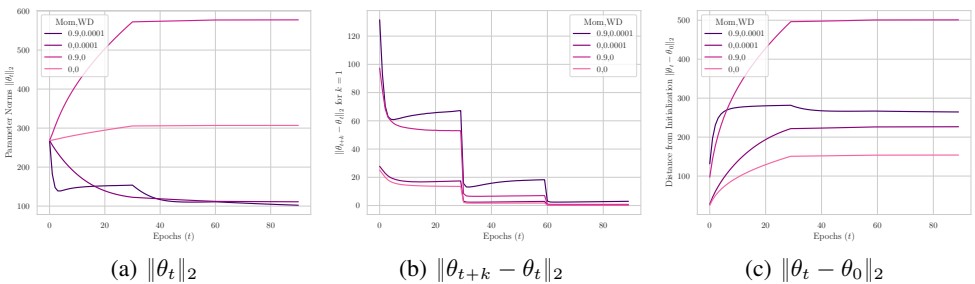

Figure 15: Norm-based measures of the Trajectory for ResNet50 trained on ImageNet

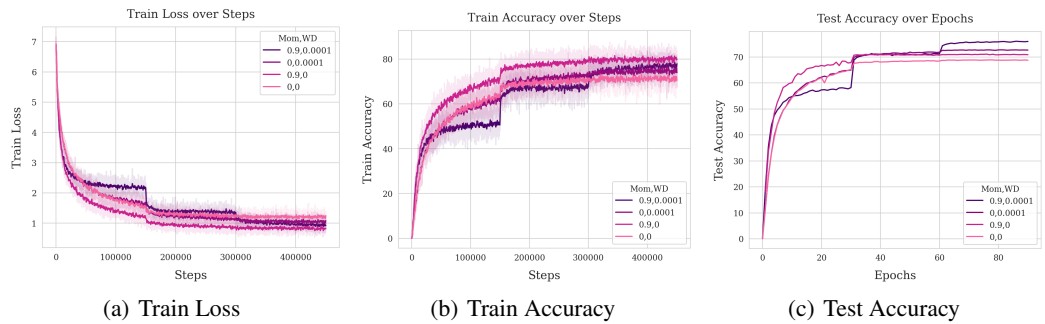

Figure 16: Loss and accuracy values for Momentum, Weight Decay experiments. The final test accuracy values are $75.96, 72.63, 70.86, 68.73$, in the order listed in the figure legend.

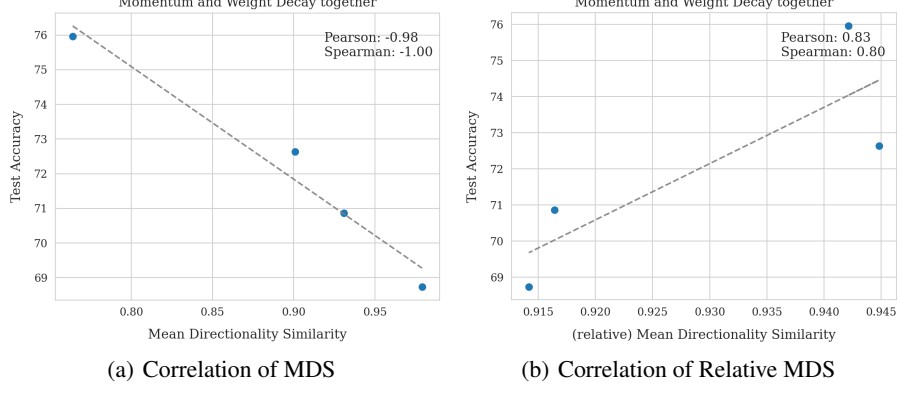

Figure 17: Correlation plots of MDS and relative MDS with test performance.

### E.2 Additional Discussion on Effects of Momentum

Besides, in Figure 3(a) and 14(f), we find that in the presence of momentum, a larger angle is traced at the origin by the trajectory, suggesting a more directional exploration, while the angle traced at initialization is smaller. The latter can also be seen from Figure 15(b), since with momentum, the trajectory moves further away from the initialization. Apart from this, in the absence of weight decay, the updates seem to be strengthening with momentum and the parameter norm rises 15(a) as well, giving rise to a mental picture of a trajectory similar to that left purple trajectory in Figure 1, at least until the training hits EoS.

With weight decay, as there is a decrease in parameter norm Figure 15(a) alongside the EoS process, as well as due to the presence of larger obtuse angles, we expect a reasonable affinity with our illustration, where we see the updates oscillating and slowly drifting towards the origin $O$ below.

### E.3 ResNet50: Weight Decay, AdamW

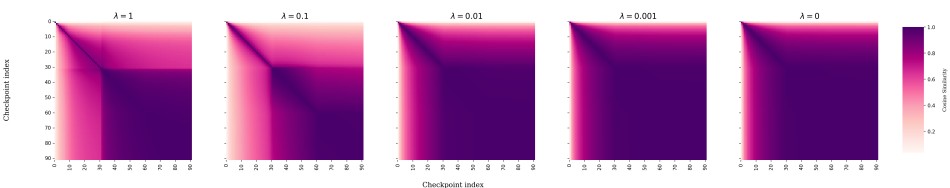

Figure 18: Trajectory Maps of ResNet50 models across different amounts of weight decay

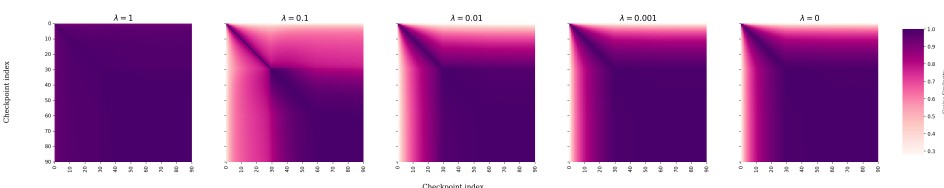

Figure 19: Relative Trajectory Maps, with respect to initialization, of ResNet50 models across different amounts of weight decay

### E.4 Additional Discussion on Weight Decay

Let us turn to weight decay alone and understand its directional effect. We have already noticed the increase in mean directional similarity (MDS) when weight decay is disabled for ResNet50 trained with SGD on ImageNet. In fact, we find a similar effect with an adaptive optimizer, like AdamW (Loshchilov and Hutter, 2017) — the trajectory maps for which are shown in Figure E.3. Here, we used regularization constants from $\lambda = 0$ until the first value where we witness a decrease in test performance, which in this case was $\lambda = 1$. Specifically, we analyze the weight decay coefficients in $\lambda \in \{1, 0.1, 0.01, 0.001, 0\}$. The corresponding MDS come out to be, $\omega = 0.731, 0.679, 0.844, 0.882, 0.885$. We notice that, as before, increasing weight decay leads to a heightened directional exploration, or lower MDS; except $\lambda = 1$ being the seeming anomaly.

But we find that this can be remedied simply by looking at the relative trajectory maps (Figure E.3), and computing the relative MDS, i.e., $\omega_0$ is $0.985, 0.807, 0.862, 0.897, 0.900$ for $\lambda = 1, 0.1, 0.01, 0.001, 0$ respectively. This occurs since such a high weight decay $\lambda = 1$, causes this particular network to underfit (train/test top-1 accuracy are $54.63\%, 50.52\%$). The performance for the rest of the networks improves, more or less, as expected with weight decay, and in particular, achieve accuracies of $75.45\%, 73.38\%, 71.03\%, 71.41\%$.

Having reaffirmed our results extensively about the directional exploration due to weight decay, we can understand it through a simple physics-based intuition, as shown in the Figure 5. In particular, we can think of the loss gradient pulling the network parameters rightwards, while the force exerted by weight decay tries to pull the network downwards. The relative strengths of these two 'forces'

have been represented by the lengths of the two vector arrows. We notice that as the weight decay strength is increased, from the left subfigure to the right, the angle traced at the origin (O) also increases. This explains how weight decay can contribute towards directional exploration.

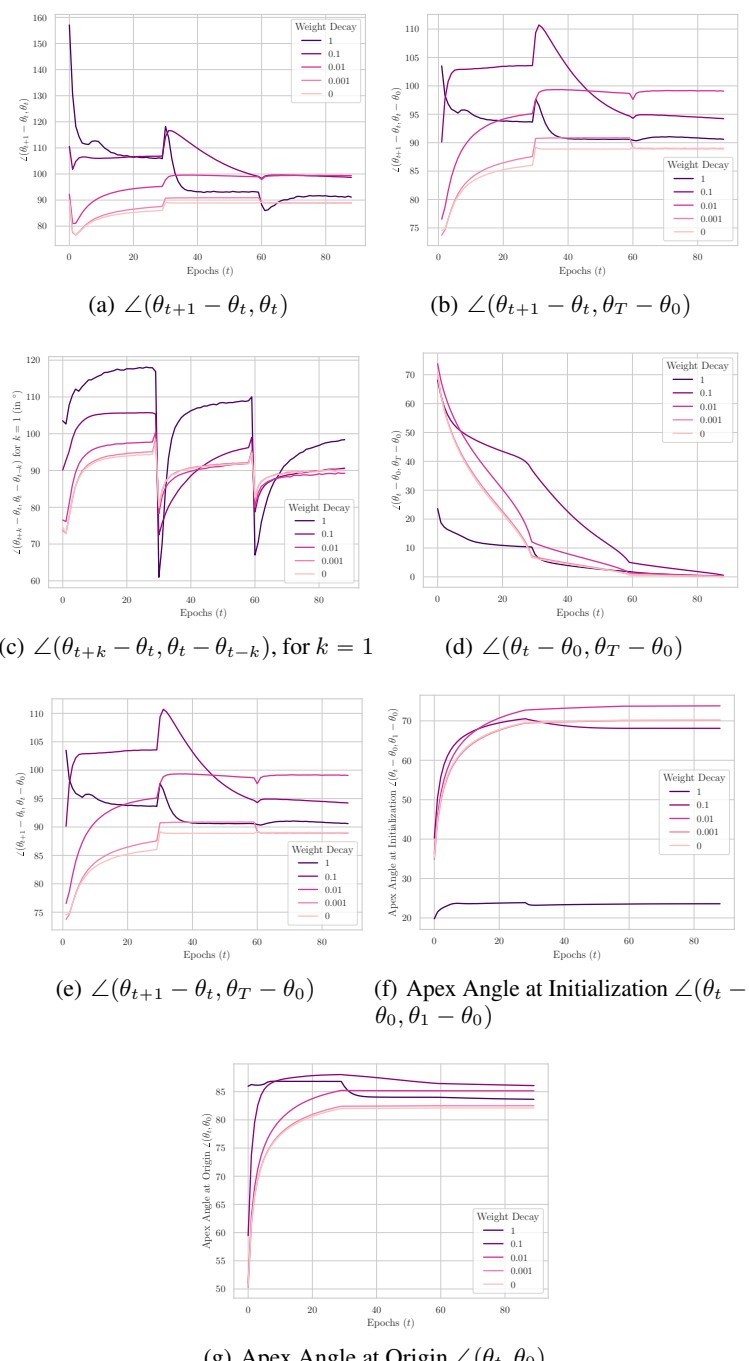

Figure 20: Angular measures of the Trajectory for ResNet50 trained on ImageNet

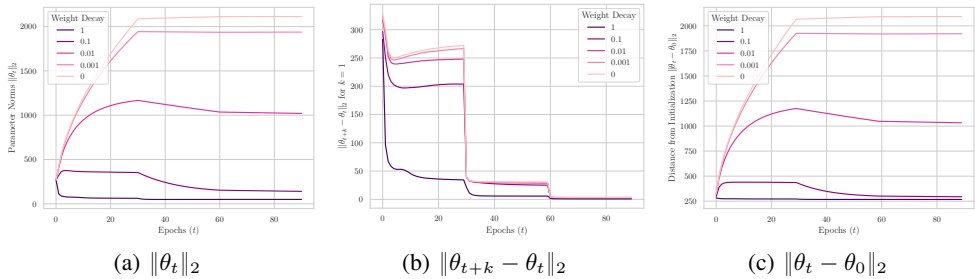

Figure 21: Norm-based measures of the Trajectory for ResNet50 trained on ImageNet

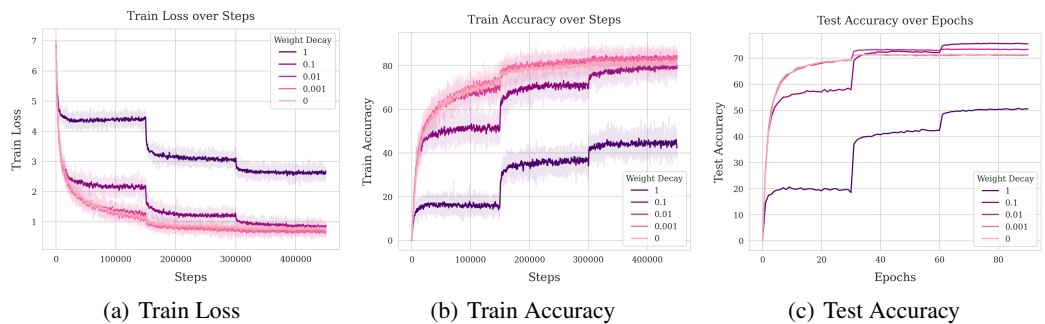

Figure 22: Loss and accuracy values for Weight Decay experiments. The final test accuracy values are $50.54, 75.45, 73.38, 71.03, 71.41$, in the order listed in the figure legend.

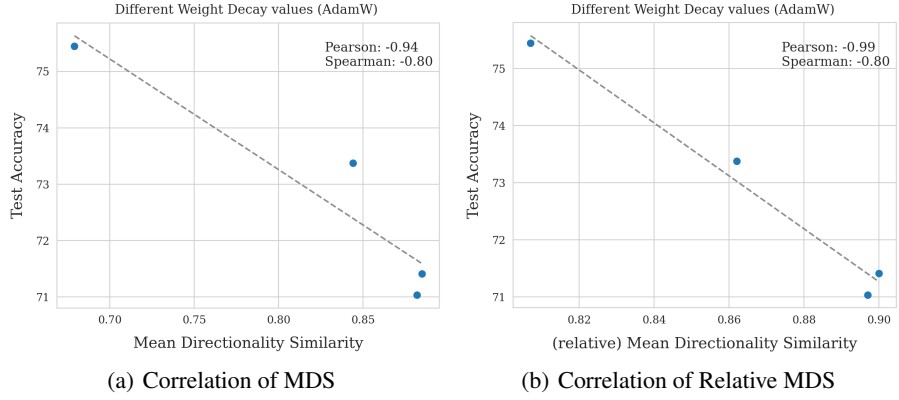

Figure 23: Correlation plots of MDS and relative MDS with test performance. We omit the case where the weight decay coefficient is set to the extremely large value of 1 such that the network severely underfits (train accuracy $< 40\%$).

## F    DIRECTIONAL EFFECTS IN OTHER KEY SETTINGS

In addition to the momentum and weight decay, there are other crucial hyperparameters, such as learning rate and batch size, whose directional effects warrant a mention. We carry out additional experiments in these settings, and from where, the key findings are that the **learning rate**, as it would be easy to guess, indeed encourages directional exploration leading to low MDS scores. But, somewhat more interestingly, we find that increasing the **batch size** also helps further exploration and thereby decreases the MDS scores. The trajectory maps can be found in the Figure 64 While we encourage the curious reader to have a look at the Appendix F.9, we find that with increased batch size, the angle between the updates as well as the angle between the update and the current location become increasingly obtuse, and thus making room for a wider directional exploration. In contrast, for smaller batch sizes these angles are closer to $90°$. We hypothesize that a similar mutual interaction, as observed with weight decay and momentum, also occurs with batch size is considered. A detailed analysis, however, remains outside the current scope.

Lastly, we also experimented with **Sharpness-Aware Minimization** (Foret et al., 2021) (SAM), where we found that a higher value of the SAM regularization coefficient leads to a slightly increased directional similarity, which could potentially be related to SAM directing optimisation to flatter basins wherein the individual points are more directionally alike and have higher cosine similarities. The detailed results can be found in the Appendix F.3.

**Other Settings and Datasets.**    As a final remark for this section, we would like to emphasize that similar results for weight decay as well as momentum, can be found under different hyperparameter settings in the supplementary material. In particular, we analyze the qualitative and quantitative hallmarks for multiple values of learning rate, weight decay, and momentum for VGG16 on CIFAR10 as well as other values for momentum and weight decay in the case of ResNet50 trained with SGD, and even Vision Transformer trained with AdamW on ImageNet across varying weight decay, but these have to be omitted here due to space constraints.

### F.1    VIT: WEIGHT DECAY, ADAMW

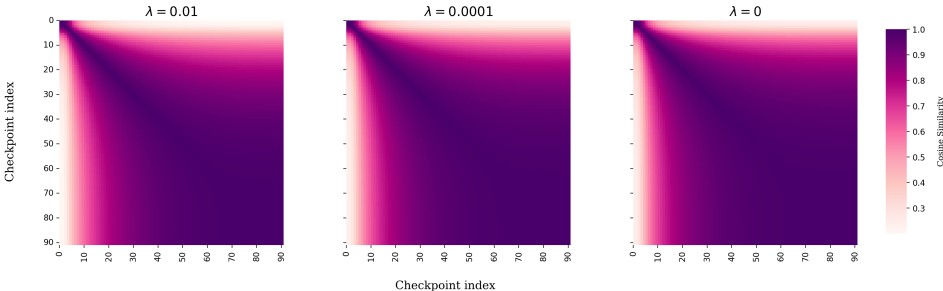

Figure 24: Trajectory Maps of ViT models across different amounts of weight decay

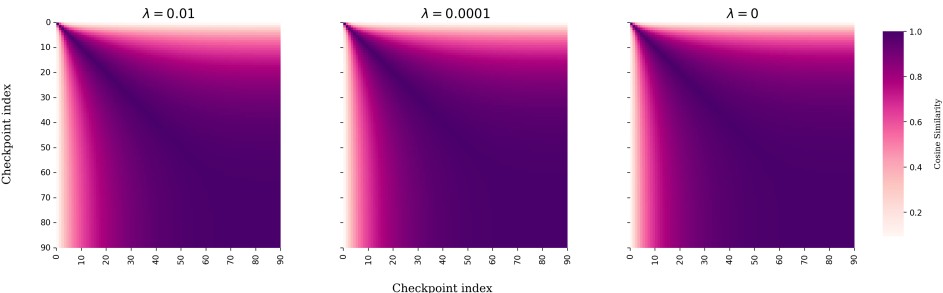

Figure 25: Relative Trajectory Maps, with respect to initialization, of ViT models across different amounts of weight decay

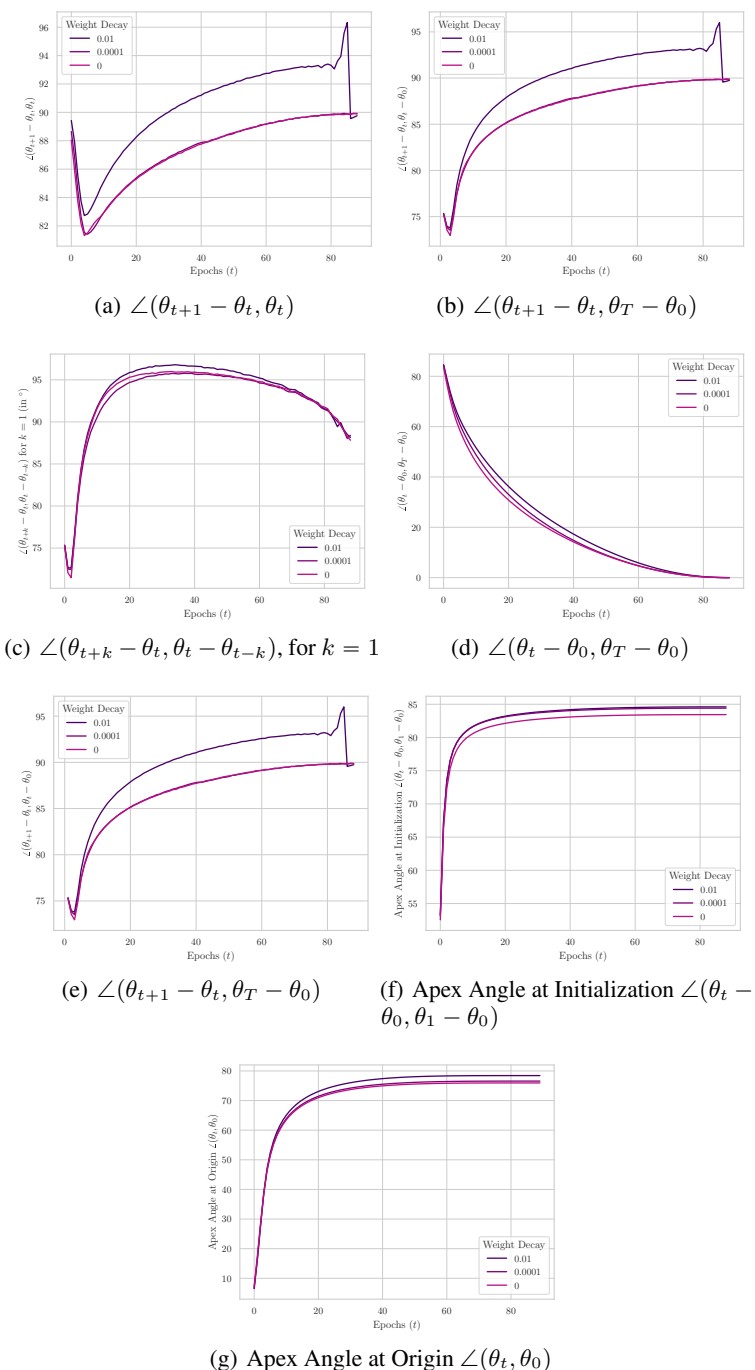

Figure 26: Angular measures of the Trajectory for ViT trained on the ImageNet dataset

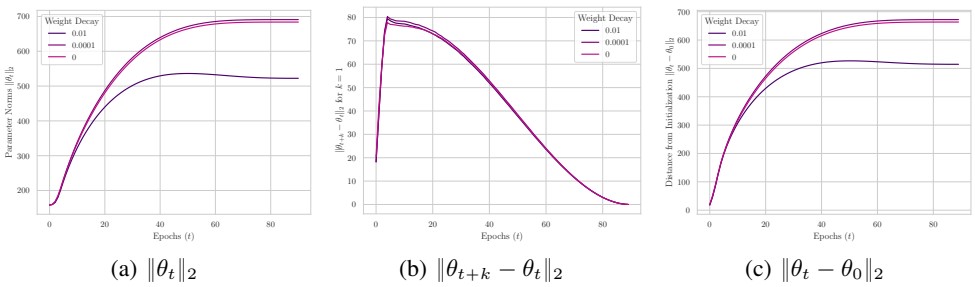

(a) $\|\theta_t\|_2$        (b) $\|\theta_{t+k} - \theta_t\|_2$        (c) $\|\theta_t - \theta_0\|_2$

Figure 27: Norm-based measures of the Trajectory for ViT trained on the ImageNet dataset

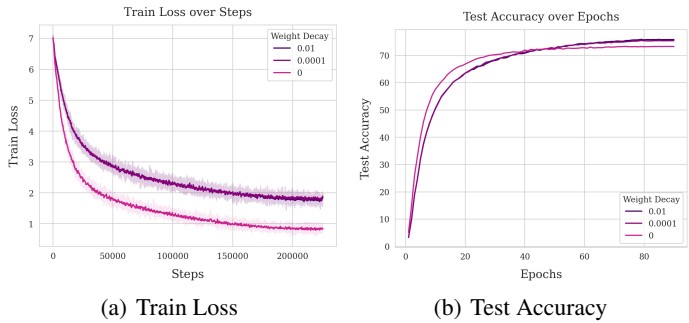

(a) Train Loss        (b) Test Accuracy

Figure 28: Loss and accuracy values for Weight Decay experiments. The final test accuracy values are $75.81, 75.36, 73.26$, in the order listed in the figure legend. The figures do contain the trends for both $0.01$ and $0.0001$, but they often lie on top of each other, so zooming-in might be needed to see them separately.

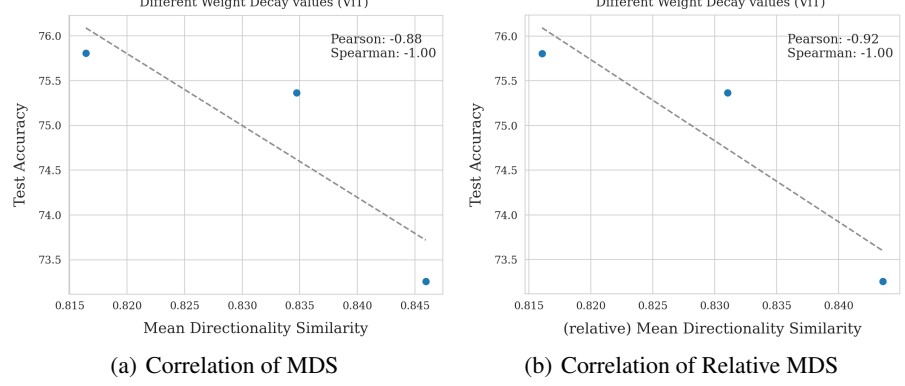

(a) Correlation of MDS        (b) Correlation of Relative MDS

Figure 29: Correlation plots of MDS and relative MDS with test performance.

## F.2 RESNET50: WEIGHT DECAY, SGD

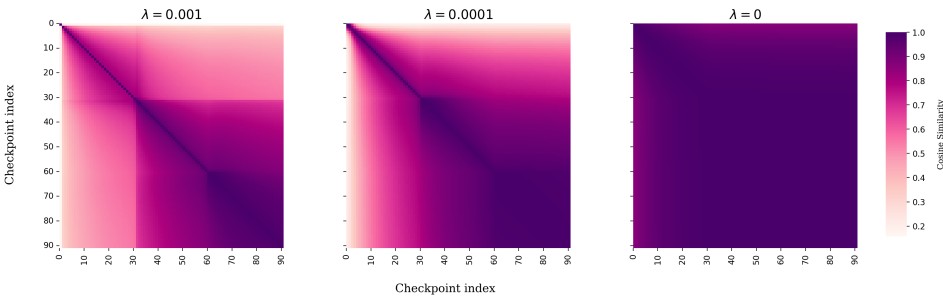

Figure 30: Trajectory Maps of ResNet50 models across different amounts of weight decay

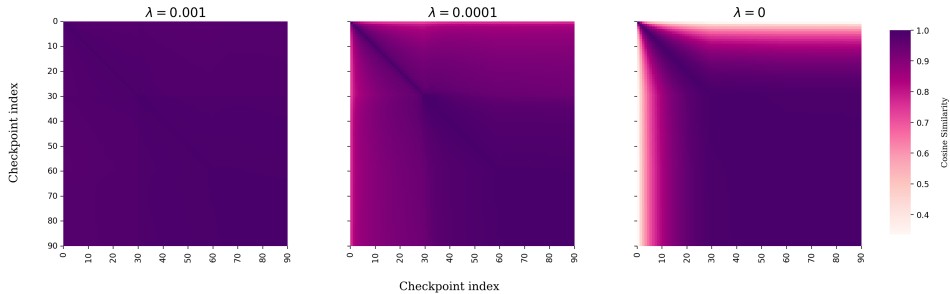

Figure 31: Relative Trajectory Maps, with respect to initialization, of ResNet50 models across different amounts of weight decay

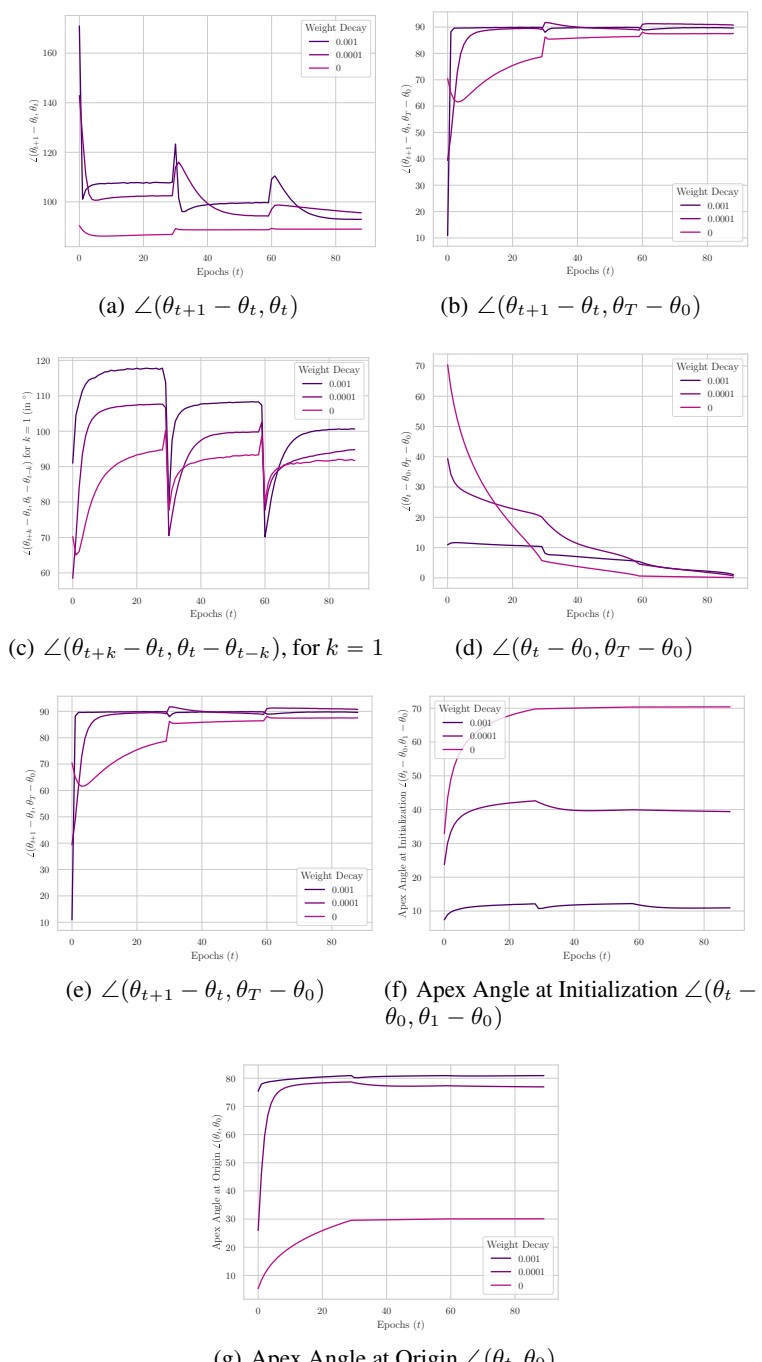

Figure 32: Angular measures of the Trajectory for ResNet50 trained on ImageNet

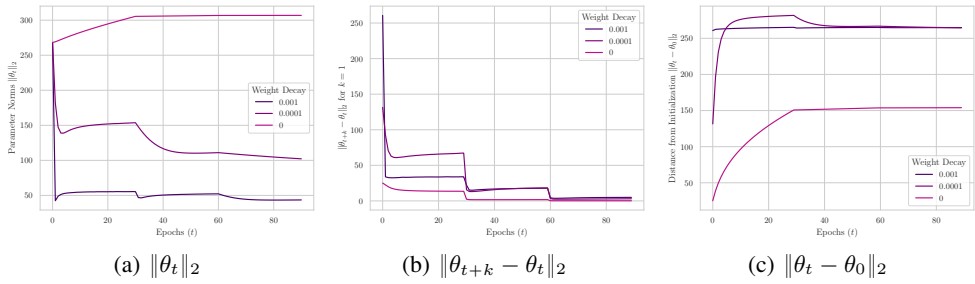

Figure 33: Norm-based measures of the Trajectory for ResNet50 trained on ImageNet

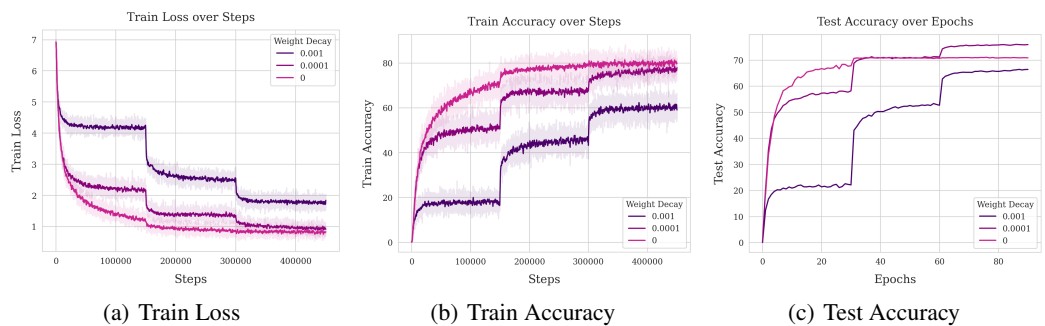

Figure 34: Loss and accuracy values for Weight Decay experiments. The final test accuracy values are $66.42, 75.96, 70.86$, in the order listed in the figure legend.

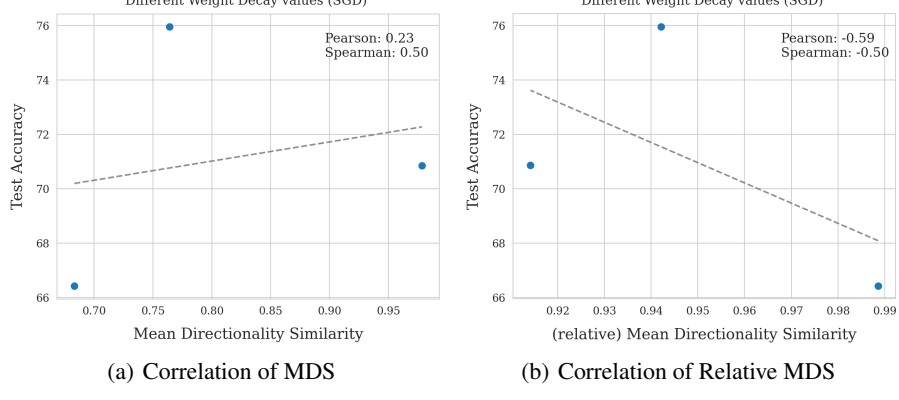

Figure 35: Correlation plots of MDS and relative MDS with test performance.

### F.3 RESNET50: SHARPNESS AWARE MINIMIZATION ANALYSIS

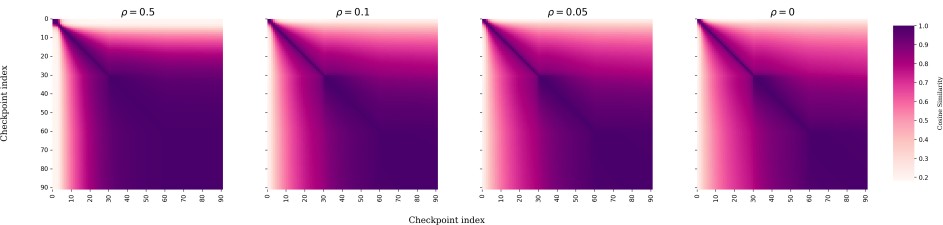

Figure 36: Trajectory Maps of ResNet50 models across different values of SAM regularization coefficient

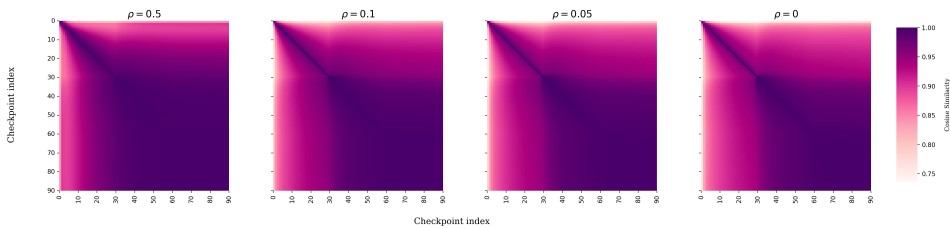

Figure 37: Relative Trajectory Maps, with respect to initialization, of ResNet50 models across different values of SAM regularization coefficient

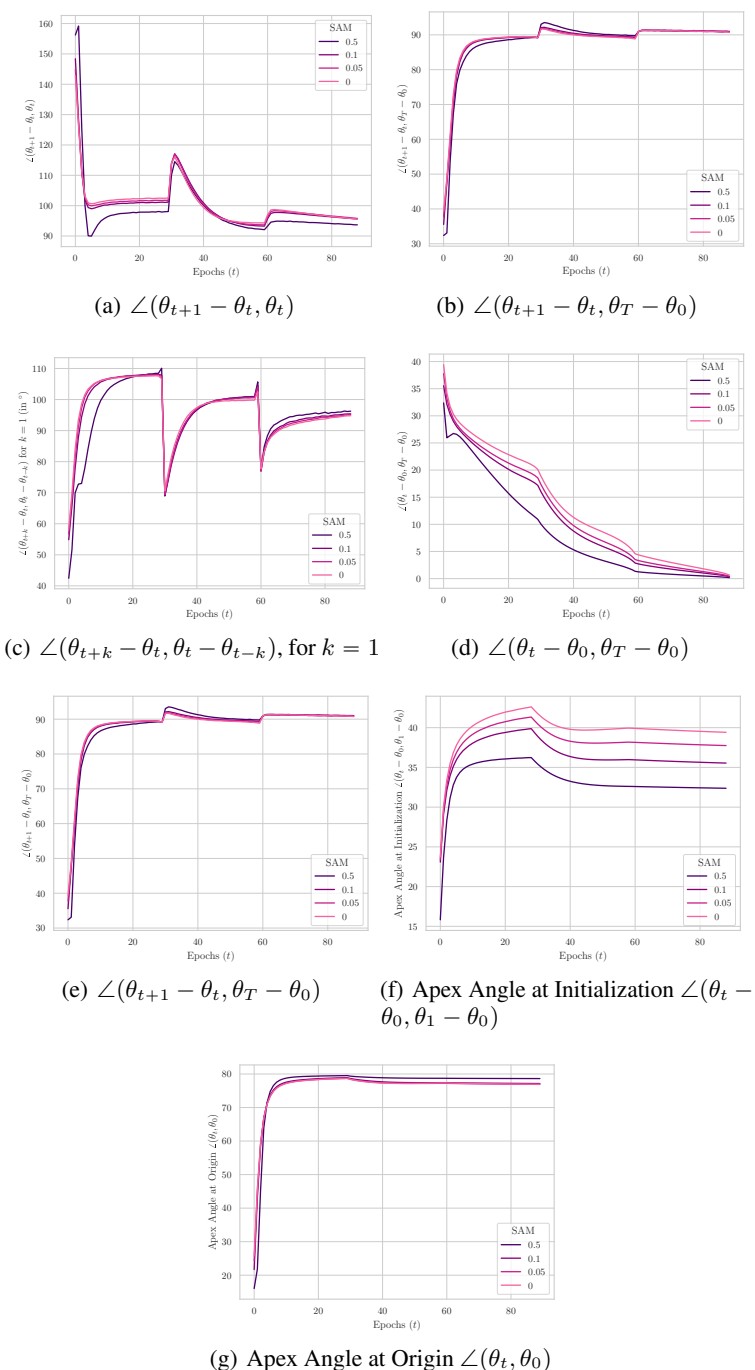

Figure 38: Angular measures of the Trajectory for ResNet50 trained on ImageNet

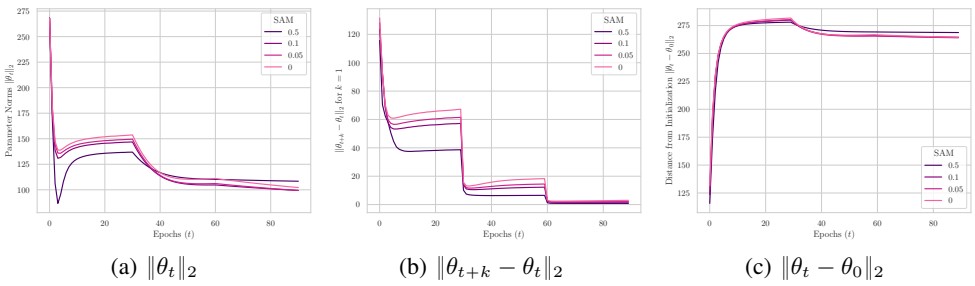

Figure 39: Norm-based measures of the Trajectory for ResNet50 trained on ImageNet

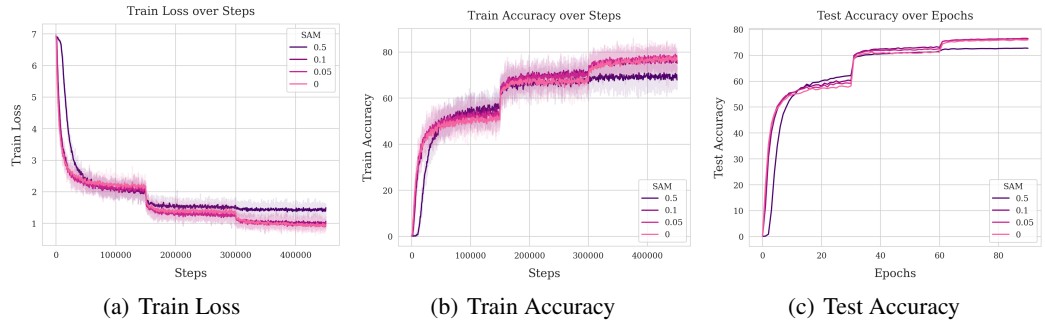

(a) Train Loss          (b) Train Accuracy          (c) Test Accuracy

Figure 40: Loss and accuracy values for SAM experiments. The final test accuracy values for $\rho = 0.5, 0.1, 0.05, 0$ are $72.72, 76.5, 76.3, 75.96$ respectively.

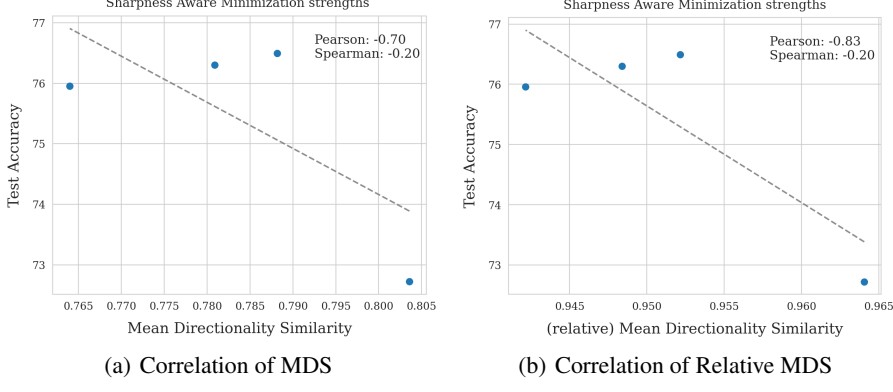

(a) Correlation of MDS          (b) Correlation of Relative MDS

Figure 41: Correlation plots of MDS and relative MDS with test performance.

### F.4 RESNET50: MOMENTUM ANALYSIS, LR 0.1, WD 0.0001

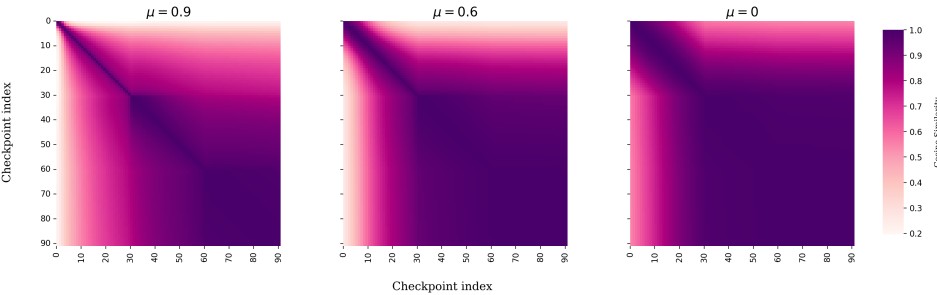

Figure 42: Trajectory Maps of ResNet50 models across different amounts of momentum

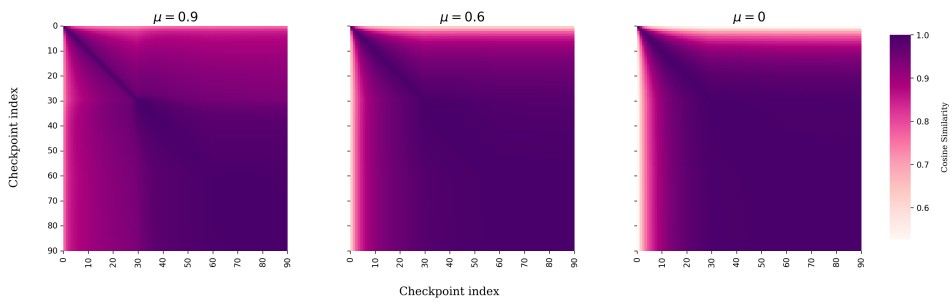

Figure 43: Relative Trajectory Maps, with respect to initialization, of ResNet50 models across different amounts of momentum

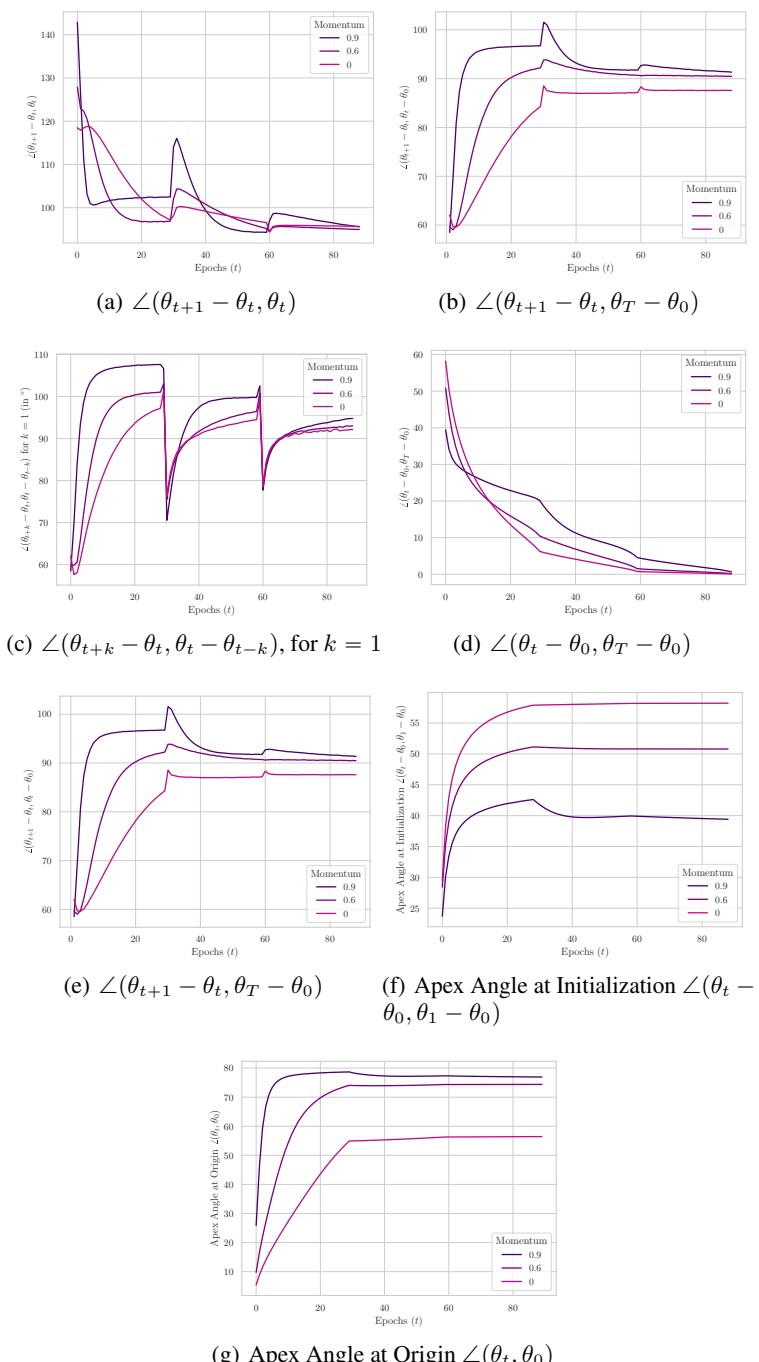

Figure 44: Angular measures of the Trajectory for ResNet50 trained on ImageNet

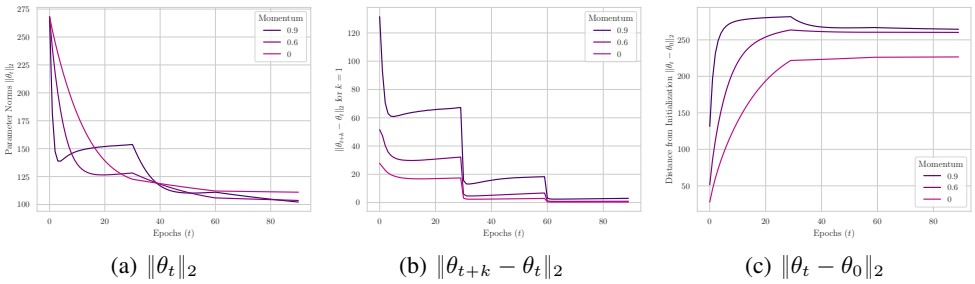

Figure 45: Norm-based measures of the Trajectory for ResNet50 trained on ImageNet

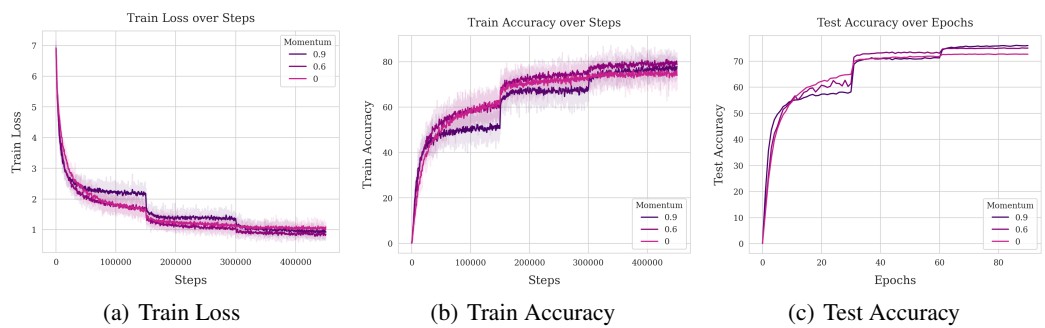

Figure 46: Loss and accuracy values for Weight Decay experiments. The final test accuracy values are $75.96, 74.98, 72.63$, in the order listed in the figure legend.

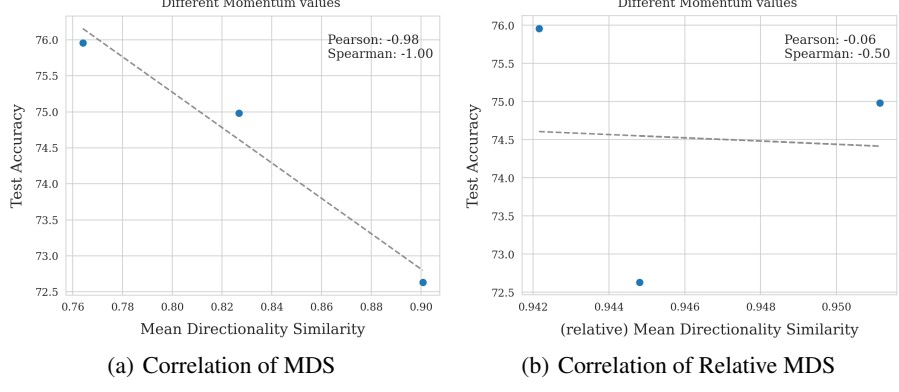

Figure 47: Correlation plots of MDS and relative MDS with test performance.

## F.5 VGG: MOMENTUM ANALYSIS, LR 0.1, WD 0.0001

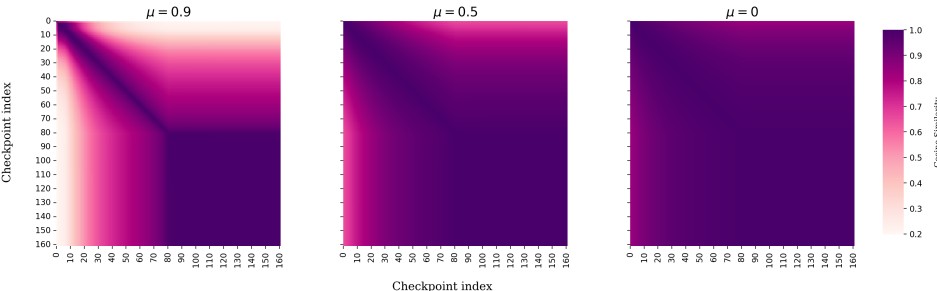

Figure 48: Trajectory Maps of VGG16 models across different amounts of momentum

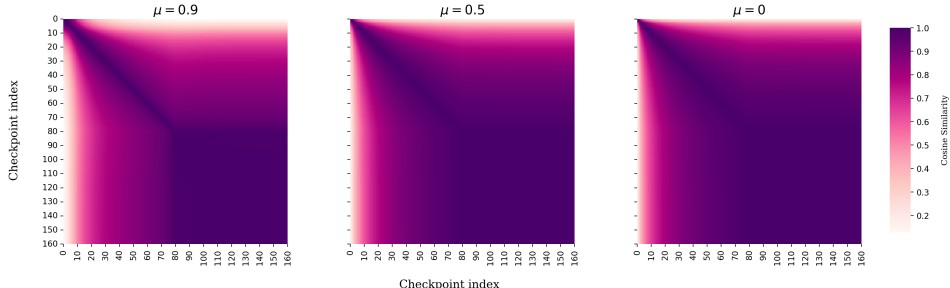

Figure 49: Relative Trajectory Maps, with respect to initialization, of VGG16 models across different amounts of momentum

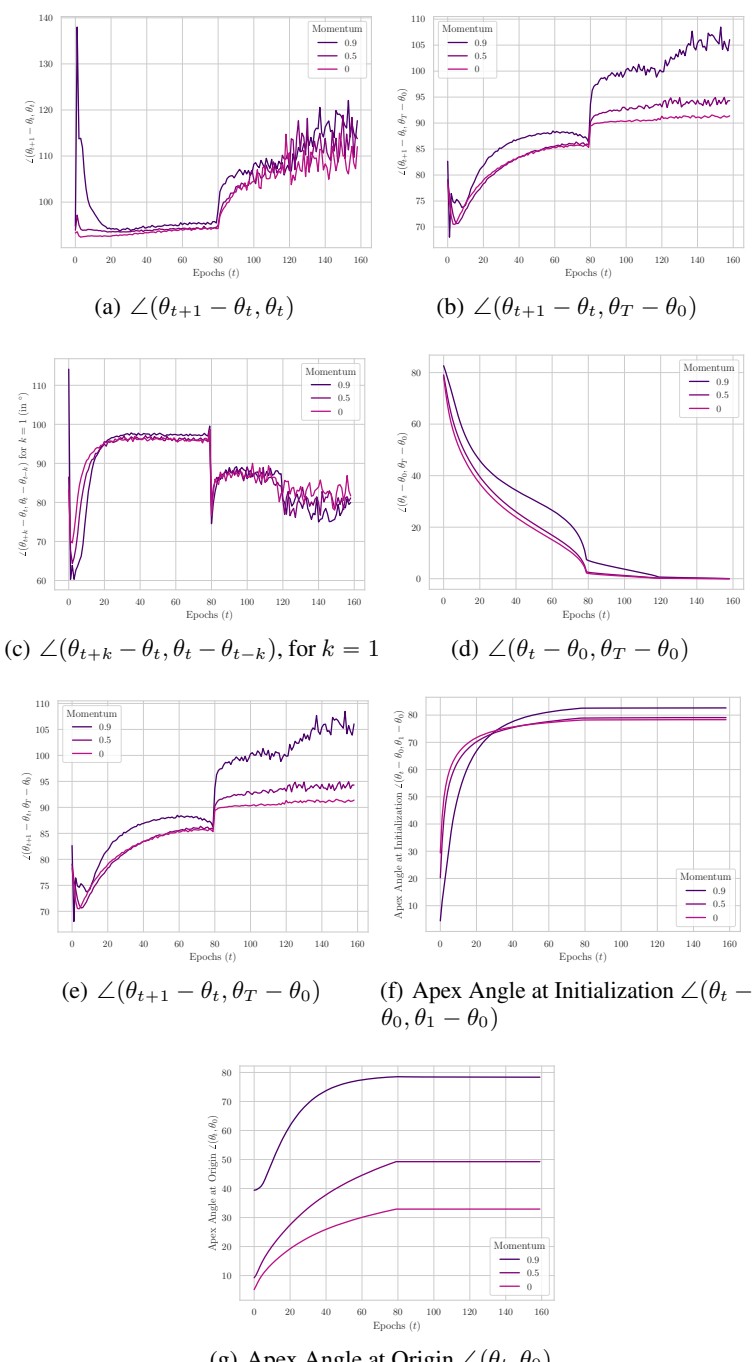

(a) $\angle(\theta_{t+1} - \theta_t, \theta_t)$

(b) $\angle(\theta_{t+1} - \theta_t, \theta_T - \theta_0)$

(c) $\angle(\theta_{t+k} - \theta_t, \theta_t - \theta_{t-k})$, for $k = 1$

(d) $\angle(\theta_t - \theta_0, \theta_T - \theta_0)$

(e) $\angle(\theta_{t+1} - \theta_t, \theta_T - \theta_0)$

(f) Apex Angle at Initialization $\angle(\theta_t - \theta_0, \theta_1 - \theta_0)$

(g) Apex Angle at Origin $\angle(\theta_t, \theta_0)$

Figure 50: Angular measures of the Trajectory for VGG16 models trained on CIFAR10.

## F.6 VGG: MOMENTUM ANALYSIS, LR 0.1, WD 0

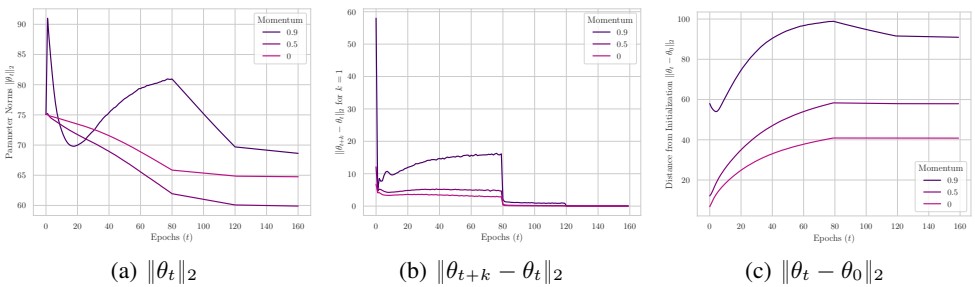

(a) $\|\theta_t\|_2$      (b) $\|\theta_{t+k} - \theta_t\|_2$      (c) $\|\theta_t - \theta_0\|_2$

Figure 51: Norm-based measures of the Trajectory for VGG16 models trained on CIFAR10.

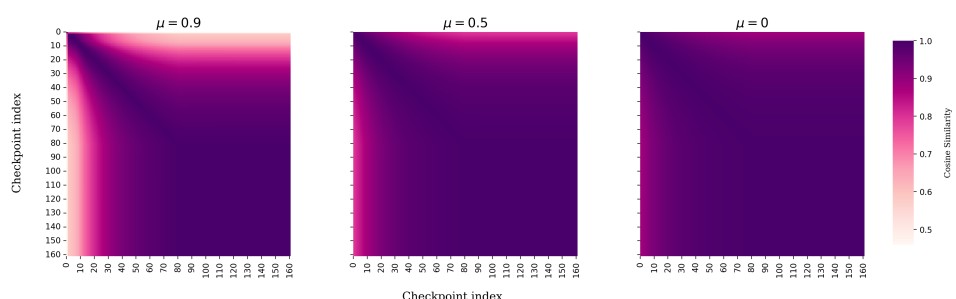

Figure 52: Trajectory Maps of VGG16 models across different amounts of momentum

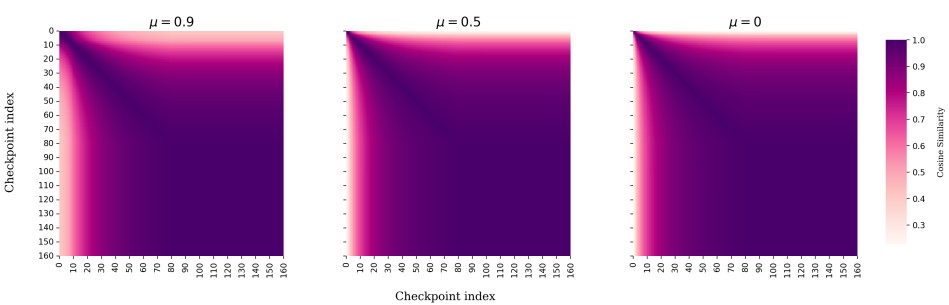

Figure 53: Relative Trajectory Maps, with respect to initialization, of VGG16 models across different amounts of momentum

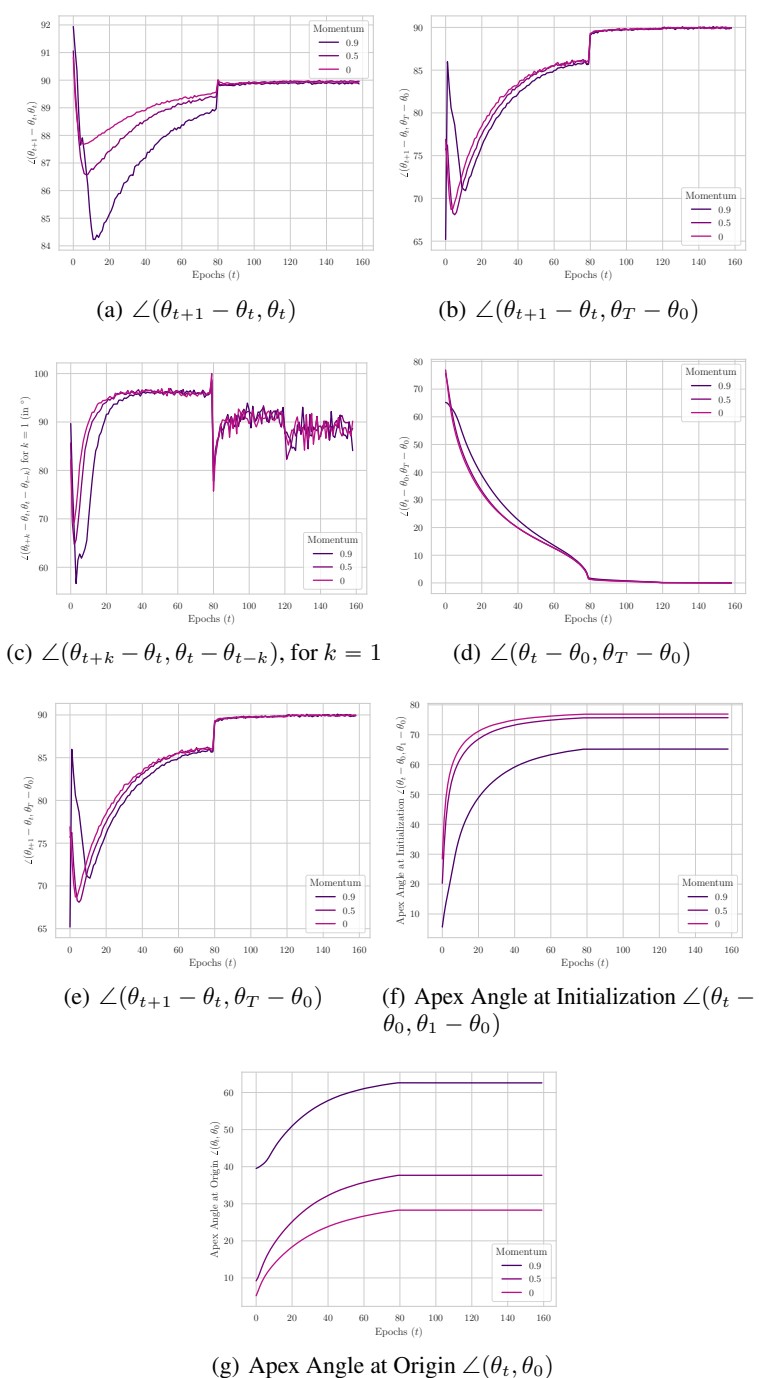

Figure 54: Angular measures of the Trajectory for VGG16 models trained on CIFAR10.

### F.7 VGG: MOMENTUM ANALYSIS, LR 0.01, WD 0.0001

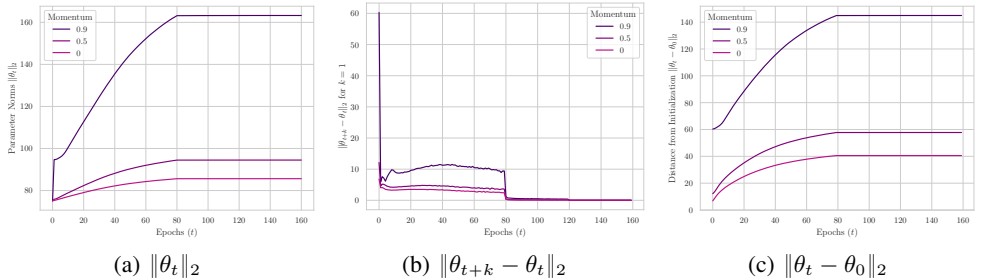

(a) $\|\theta_t\|_2$      (b) $\|\theta_{t+k} - \theta_t\|_2$      (c) $\|\theta_t - \theta_0\|_2$

Figure 55: Norm-based measures of the Trajectory for VGG16 models trained on CIFAR10.

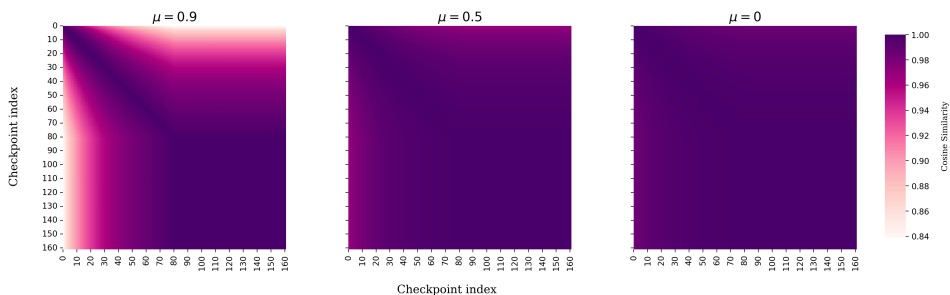

Figure 56: Trajectory Maps of VGG16 models across different amounts of momentum

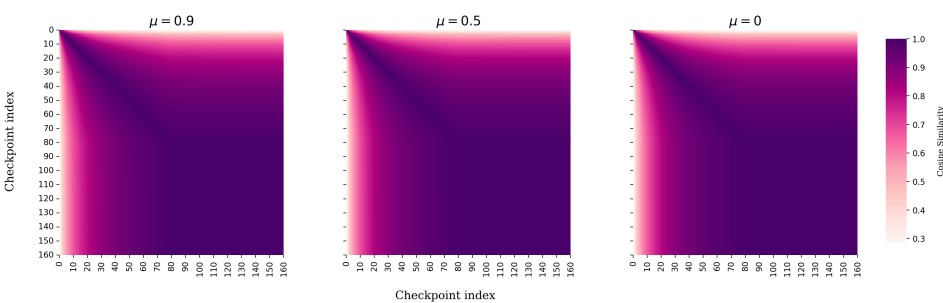

Figure 57: Relative Trajectory Maps, with respect to initialization, of VGG16 models across different amounts of momentum

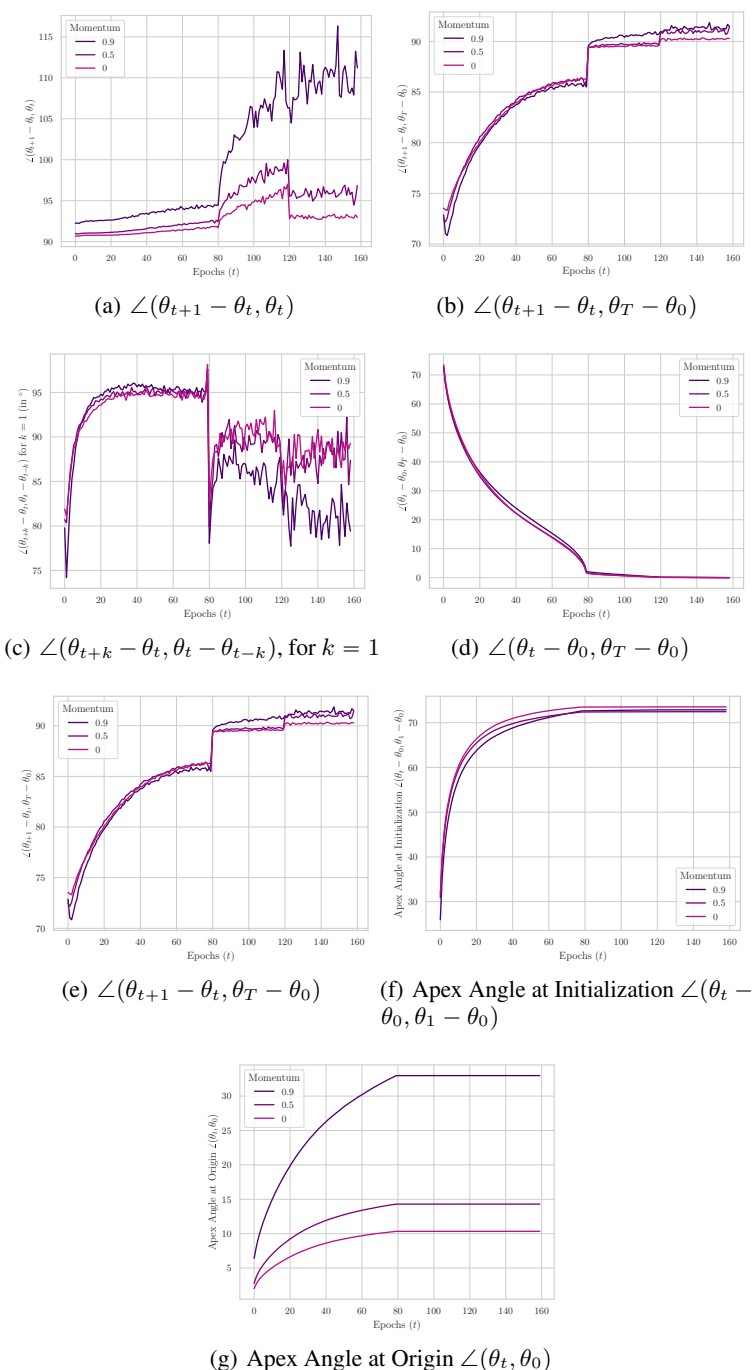

Figure 58: Angular measures of the Trajectory for VGG16 models trained on CIFAR10.

## F.8 VGG: MOMENTUM ANALYSIS, LR 0.01, WD 0

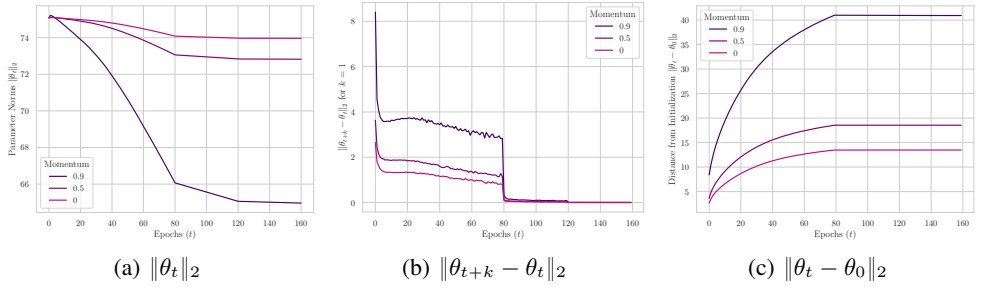

(a) $\|\theta_t\|_2$      (b) $\|\theta_{t+k} - \theta_t\|_2$      (c) $\|\theta_t - \theta_0\|_2$

Figure 59: Norm-based measures of the Trajectory for VGG16 models trained on CIFAR10.

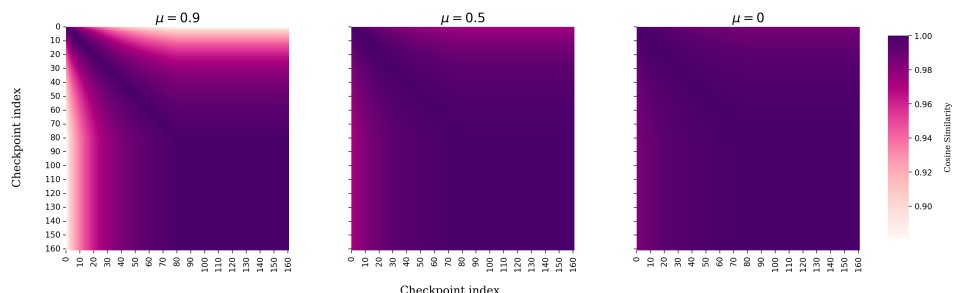

Figure 60: Trajectory Maps of VGG16 models across different amounts of momentum

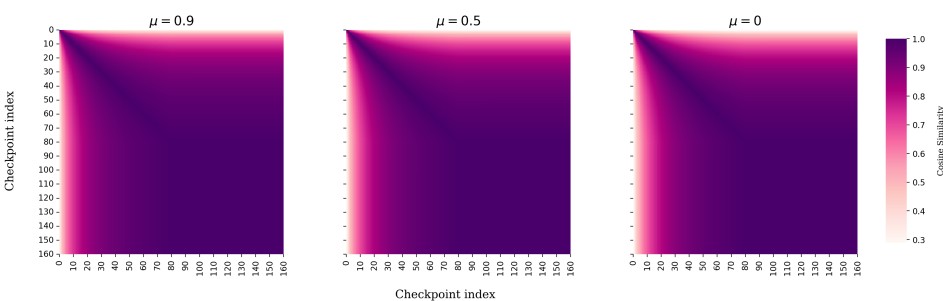

Figure 61: Relative Trajectory Maps, with respect to initialization, of VGG16 models across different amounts of momentum

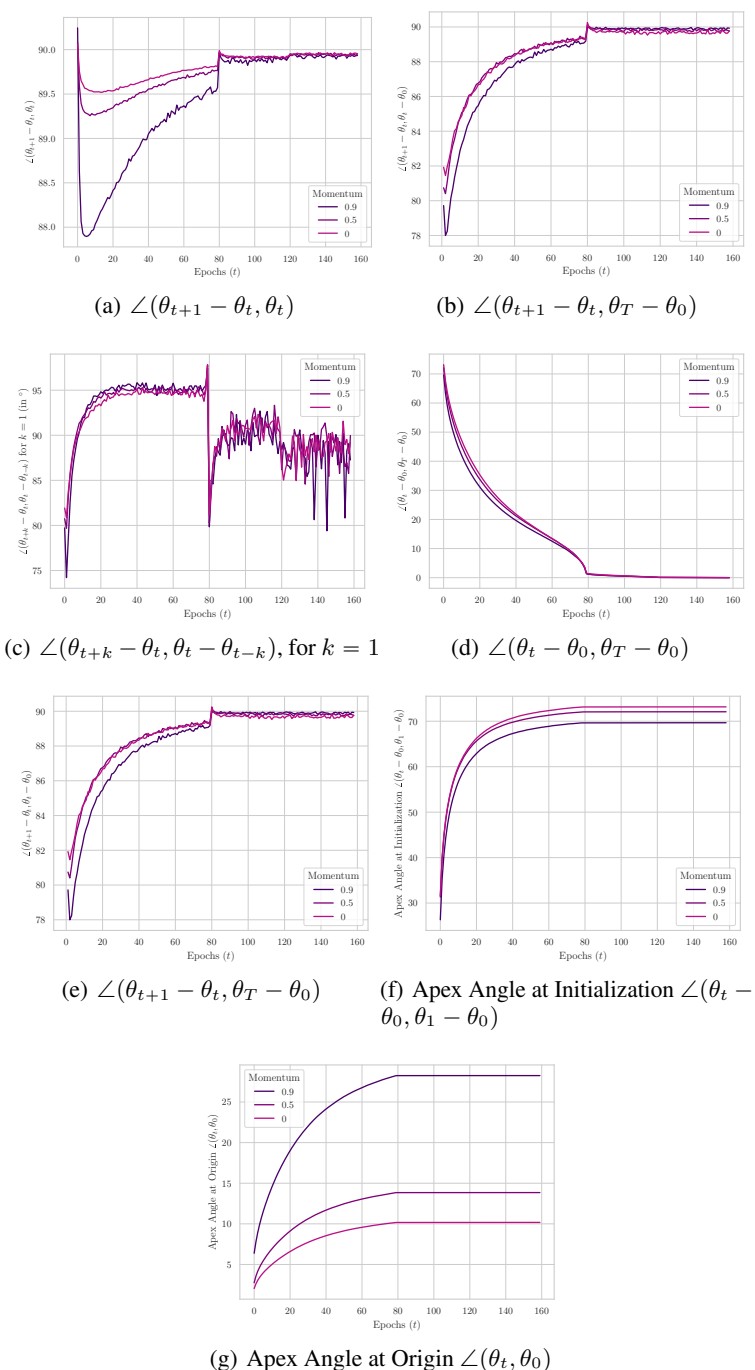

Figure 62: Angular measures of the Trajectory for VGG16 models trained on CIFAR10.

F.9    VGG16 BATCH SIZE ANALYSIS

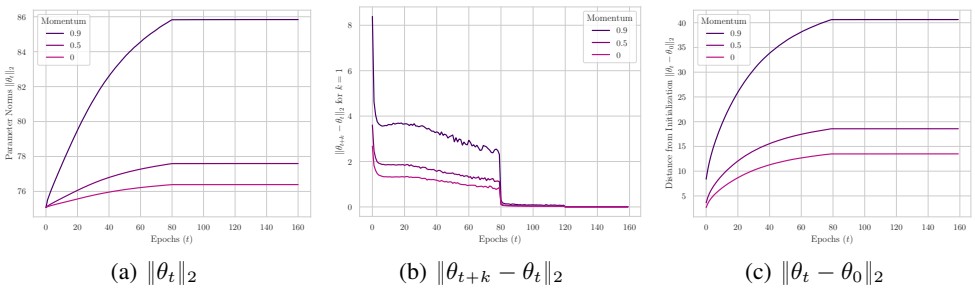

Figure 63: Norm-based measures of the Trajectory for VGG16 models trained on CIFAR10.

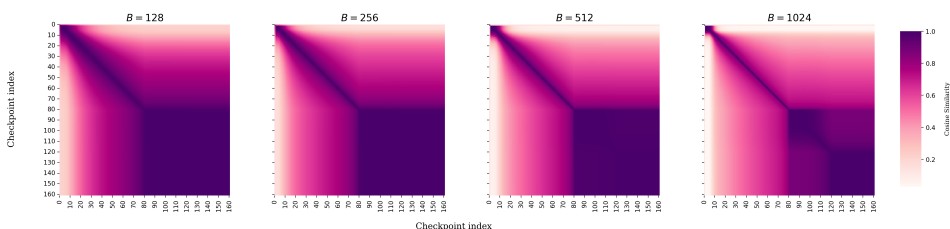

Figure 64: Trajectory Maps of VGG16 models across different batch sizes. The learning rates have been scaled in proportion to the batch size, and the training schedule was adjusted to ensure an equal number of steps (and not simply epochs) for all the runs. We also adjusted the learning rate schedule to drop learning rates at a corresponding number of steps across the experiments. The respective MDS values are $\omega = 0.753, 0.723, 0.660, 0.619$ and the test accuracies are $91.63\%, 91.82\%, 92.44\%, 92.39\%$.

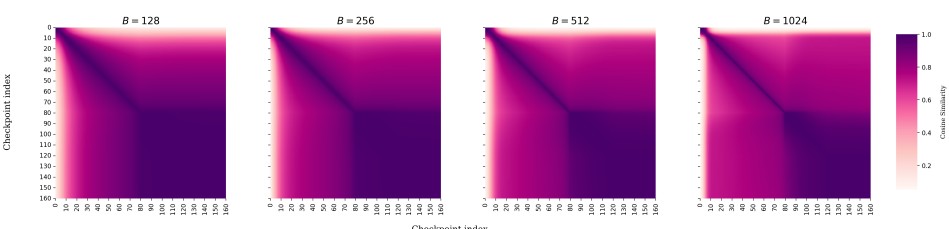

Figure 65: Relative Trajectory Maps, with respect to initialization, of VGG16 models across different batch sizes

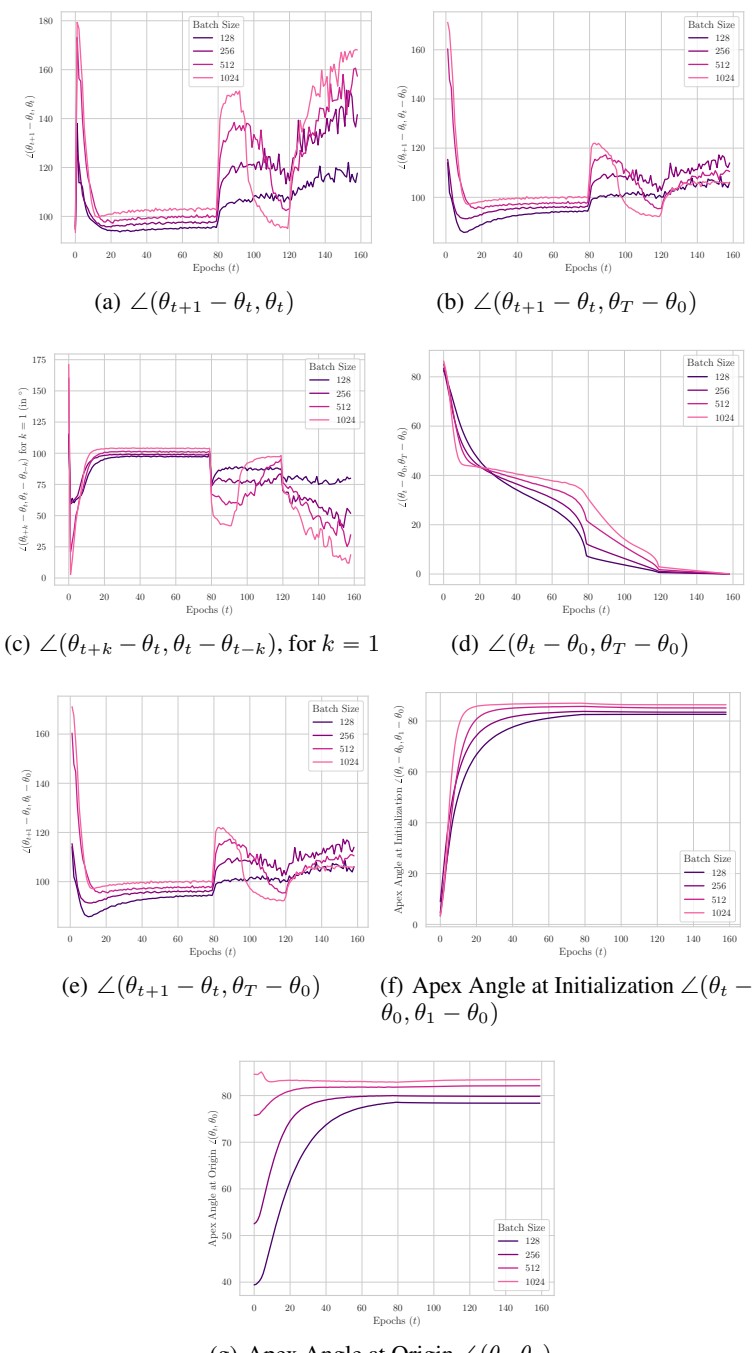

Figure 66: Angular measures of the Trajectory for VGG16 models trained on CIFAR10.

## F.10 TRAJECTORY MAPS IN THE PRESENCE OF LABEL NOISE

We observe that with increasing label noise, the network is required to undergo more directional exploration to find a solution that can interpolate the training set. The MDS scores decrease monotonically with increasing label noise.

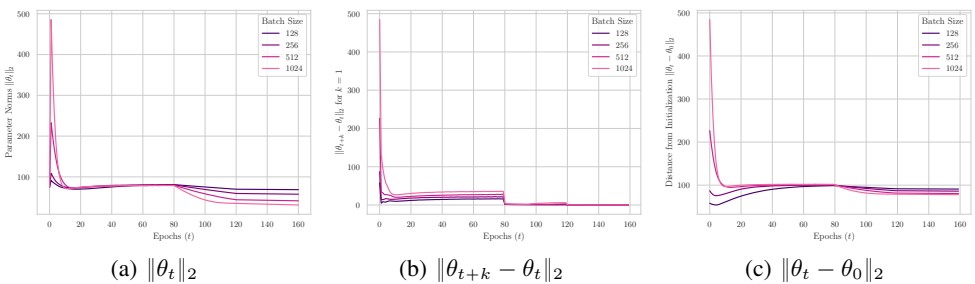

Figure 67: Norm-based measures of the Trajectory for VGG16 models trained on CIFAR10.

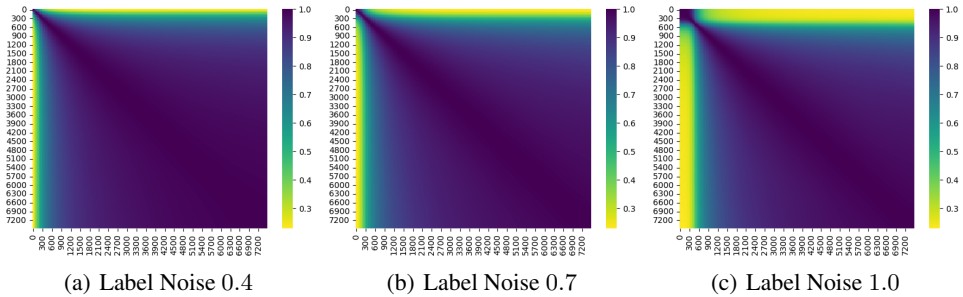

Figure 68: Trajectory maps when a CNN is trained on CIFAR10 with different amounts of label noise, i.e., what fraction of samples have been assigned random labels.

## G    GPT-NEOX TRAJECTORY ANALYSIS

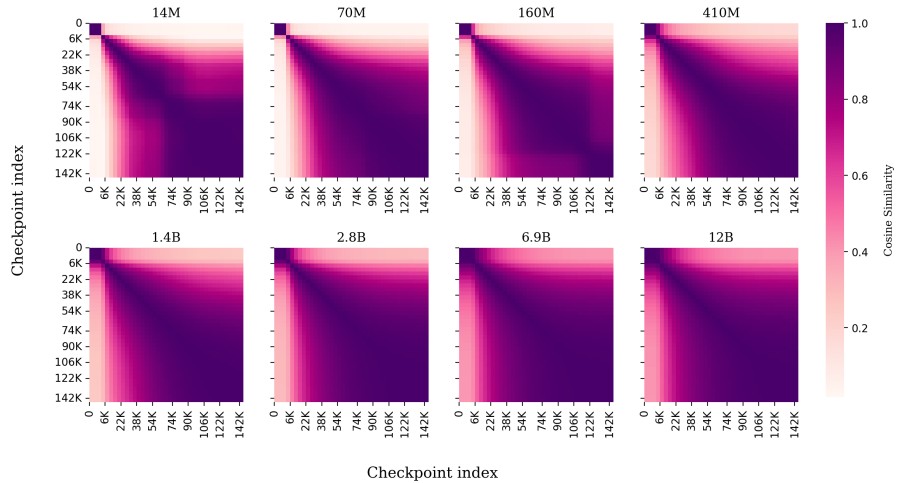

Figure 69: Trajectory Maps of Pythia GPT-NeoX models across two orders of model scales trained on Pile. The corresponding MDS values are $\omega = 0.650, 0.672, 0.678, 0.726, 0.759, 0.786, 0.818, 0.815$.

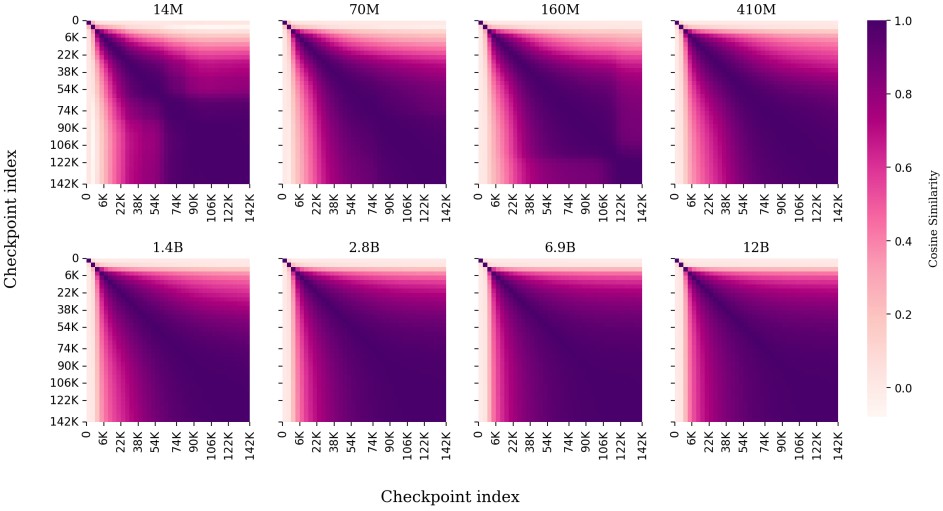

Figure 70: Relative Trajectory Maps, with respect to initialization, of Pythia GPT-NeoX models across two orders of model scales.

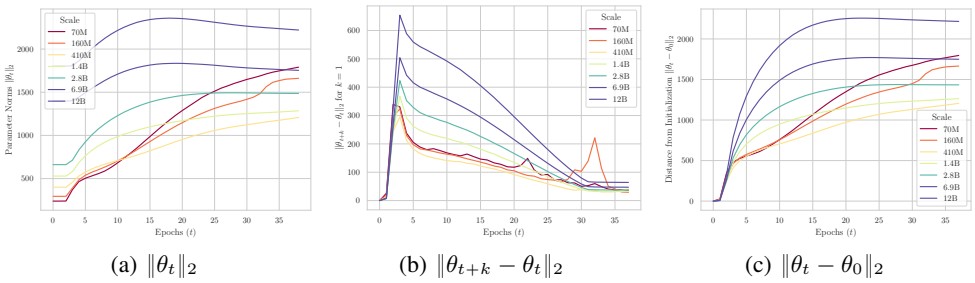

(a) $\|\theta_t\|_2$      (b) $\|\theta_{t+k} - \theta_t\|_2$      (c) $\|\theta_t - \theta_0\|_2$

Figure 71: Norm-based measures of the Trajectory for GPT-NeoX trained on the Pile dataset

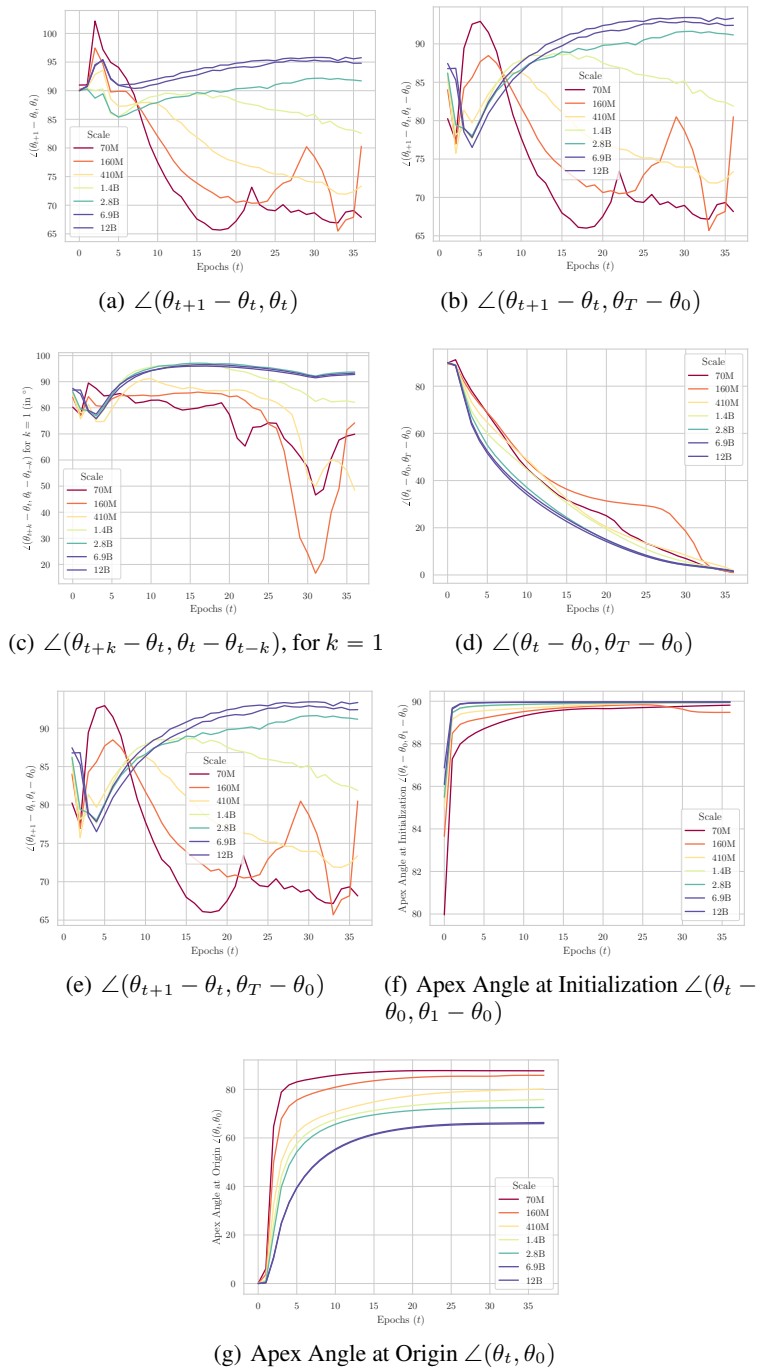

Figure 72: Angular measures of the Trajectory for GPT-NeoX trained on the Pile dataset

# H LAYERWISE-TRAJECTORY MAPS

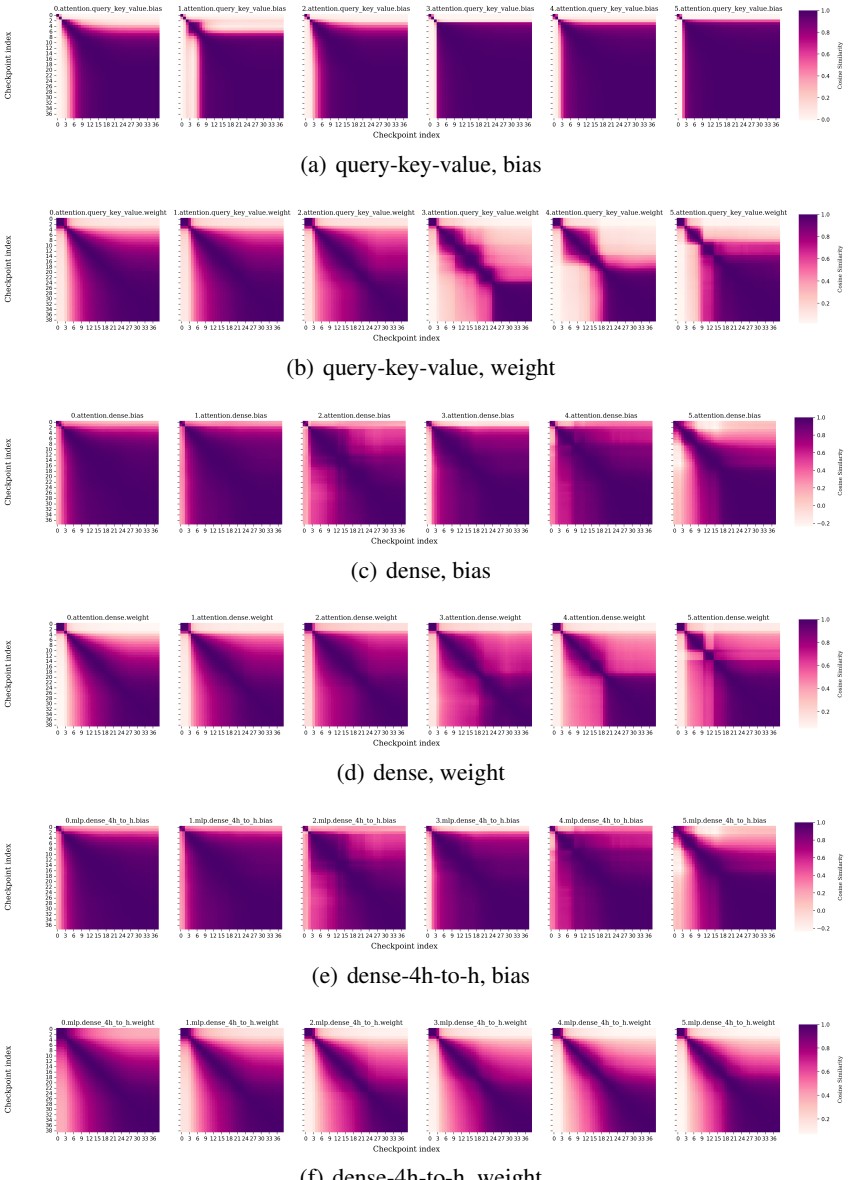

(a) query-key-value, bias

(b) query-key-value, weight

(c) dense, bias

(d) dense, weight

(e) dense-4h-to-h, bias

(f) dense-4h-to-h, weight

Figure 73: Layerwise Trajectory Maps, grouped by layer type, for the $14M$ GPT-NeoX model trained on the Pile dataset.

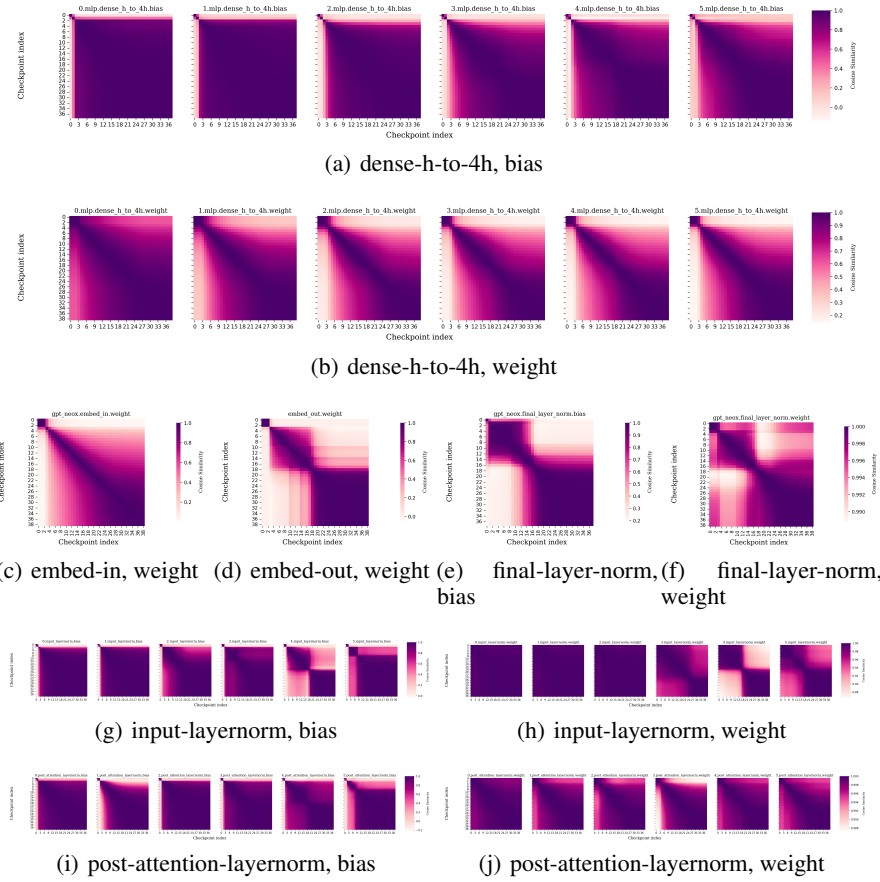

Figure 74: Layerwise Trajectory Maps, grouped by layer type, for the $14M$ GPT-NeoX model trained on the Pile dataset.

## H.1 Q,K,V Trajectory Maps across Scales

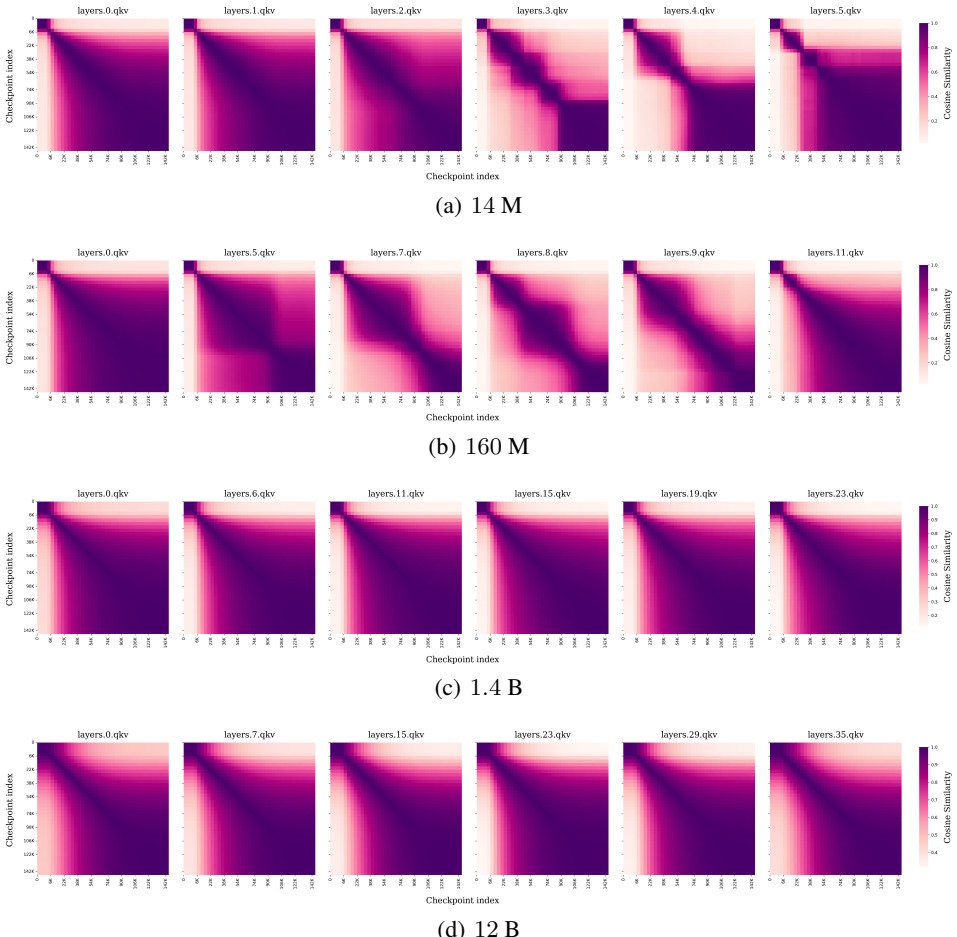

Figure 75: Trajectory maps of Q,K,V layers become homogenized over increasing scale.

## I  TRAJECTORY MAPS FOR GROKKING

In grokking (Power et al., 2022), we have that the performance on test samples significantly lags behind the training performance. Below, we look at the trajectory maps in this setting, considering the experimental setup of `https://github.com/teddykoker/grokking`. We can observe

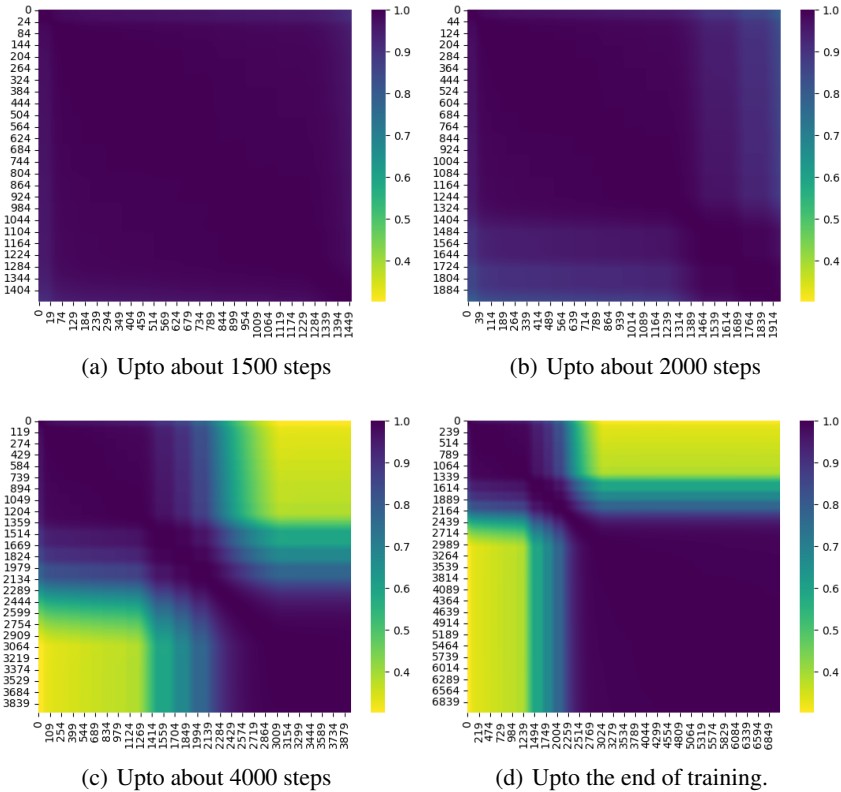

(a) Upto about 1500 steps  (b) Upto about 2000 steps

(c) Upto about 4000 steps  (d) Upto the end of training.

Figure 76: Trajectory maps during the course of learning. Grokking (Power et al., 2022), or sudden increase in test accuracy while training accuracy is already at a ceiling, occurs where the trajectory map also shows a transition point.

in Figure 76 that:

- Upto about 1500 steps: Everything is pitch blue. No directional exploration, test accuracy remains, more or less, random.
- Upto about 2000 steps: Some directional movement starts to happen, and some initial signs of improvement in test performance.
- Upto about 4000 steps: Transition point for directional exploration. Test performance visibly improves.

We think that without (appropriate) directional exploration, the training converges to a 'lazy'/'shortcut'/'dead-end' like solutions. Moreover, we believe that being 'lazy' in the directional sense is highly intertwined with being 'lazy' in the sense of feature learning (Chizat et al., 2020). Besides, the above experiments show that the resemblance with the lazy regime is more than an analogy. Kumar et al. (2024) have shown that grokking can be seen as the transition from the lazy to the non-lazy (rich) training regime. In particular, we find that the precise part of the training, where the test accuracy first shows a marked growth is also the part where the directional exploration starts to happen.

# J    PUTTING DIRECTIONAL REDUNDANCY TO TEST

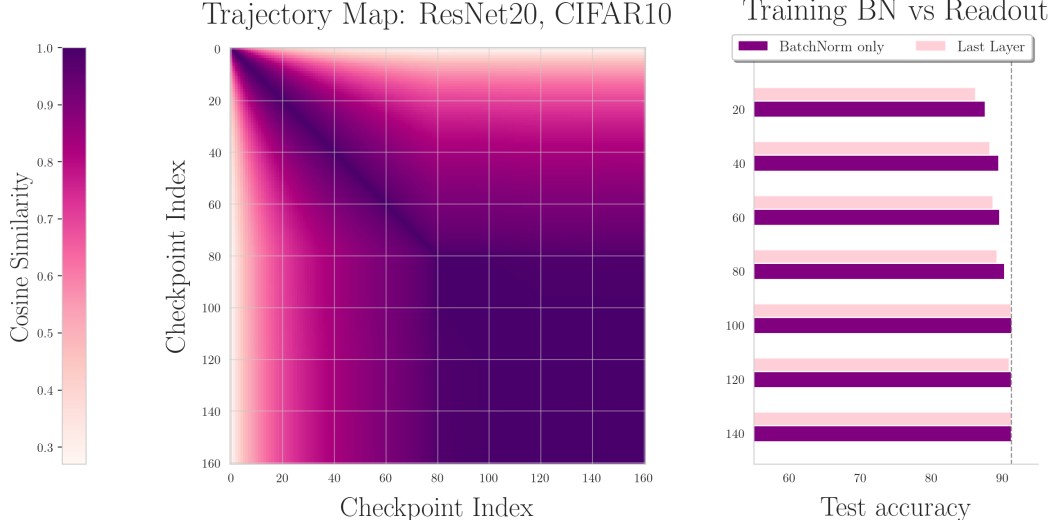

Figure 77: Comparison of training only batch norm parameters with training entire last layer (readout) parameters.

The presented results, and that in the main section, have been averaged over 3 seeds. The standard deviation is never more than 0.05 (for the batchnorm training, even less), and hence it is difficult to make it out in the plots and has been omitted.

## J.1    COMPARISON WITH (DISCRETIZED) TRAJECTORY MAP

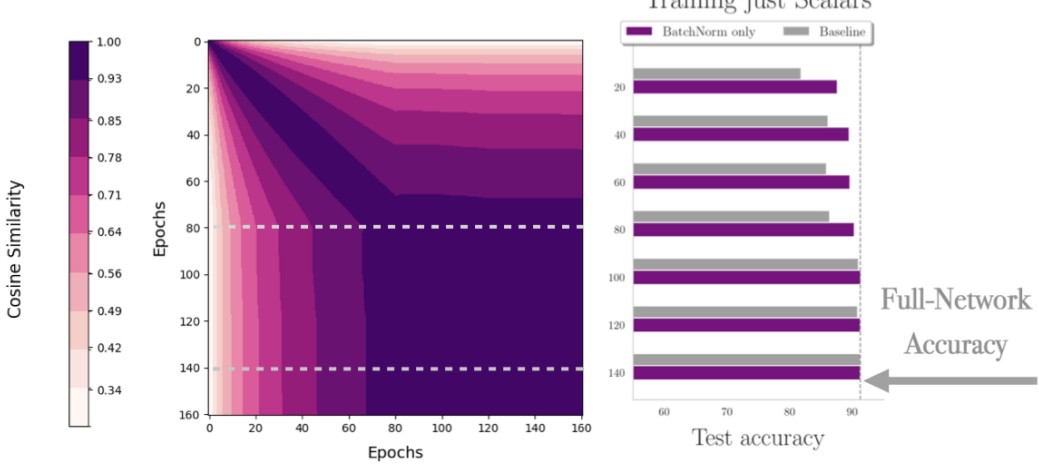

Figure 78: Trajectory Map and the corresponding decapacitation test via tuning only batch-norm parameters. Here we discretize the trajectory map to 10 color levels to facilitate comparison.

# K  EFFECT OF DIFFERENT LEARNING RATE SCHEDULES

In order to ensure that our broader trajectory map results are not sensitive to the particular choice stepwise learning rate schedule, we run a study for three other learning rate schedules: cosine, linear decay, and polynomial decay with power (2). We run this comparison both without and with an initial warmup for 5 epochs to the base learning rate of 0.1. Besides, to facilitate an easier comparison, we discretize the plots to 10 color levels. These resulting trajectory maps are illustrated below in Figures 80, 81, and the learning rate schedules plus the train/test loss are depicted in the Figure 83.

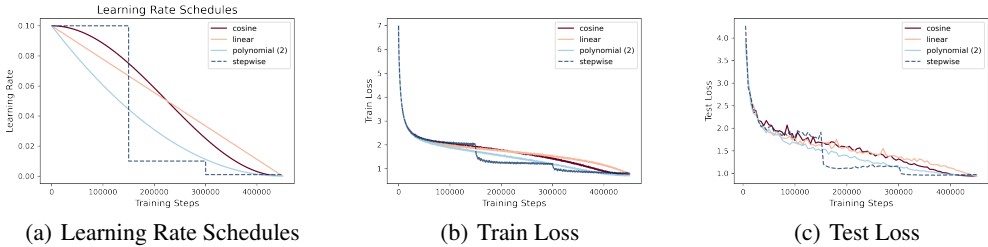

(a) Learning Rate Schedules        (b) Train Loss        (c) Test Loss

Figure 79: Visualization of different learning rate schedules and the accompanying train and test loss curves (in the non-warmup case of Figure 80).

## K.1  TRAJECTORY MAPS

Without Warmup

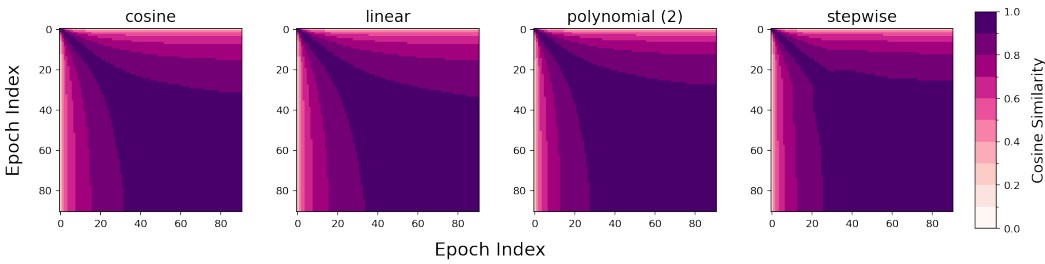

Figure 80: Comparison of trajectory maps for ResNet50 trained on ImageNet across different learning rate schedules.

### K.1.1  WITH WARMUP FOR 5 EPOCHS

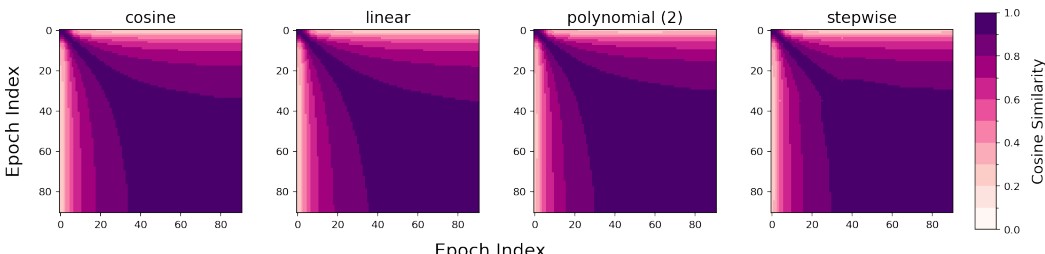

Figure 81: Comparison of trajectory maps for ResNet50 trained on ImageNet across different learning rate schedules.

Importantly, we find that the larger nature of the trajectory map is highly similar across the different schedules, despite minor variations. ***This demonstrates that our claims about directional similarity are robust to the choice of learning rate schedules.***

## K.2 ANGLE BETWEEN SUCCESSIVE EPOCHS

We also investigate the effect of different learning rate schedules on the angles formed between successive epochs, and, in particular, how it varies in the presence or absence of momentum and weight decay. Said differently, this is analogous to the setting of Figure 4(a), but where we swap out stepwise learning rate schedule for cosine, linear, and polynomial (degree 2) schedules. These results are located in the Figure 82.

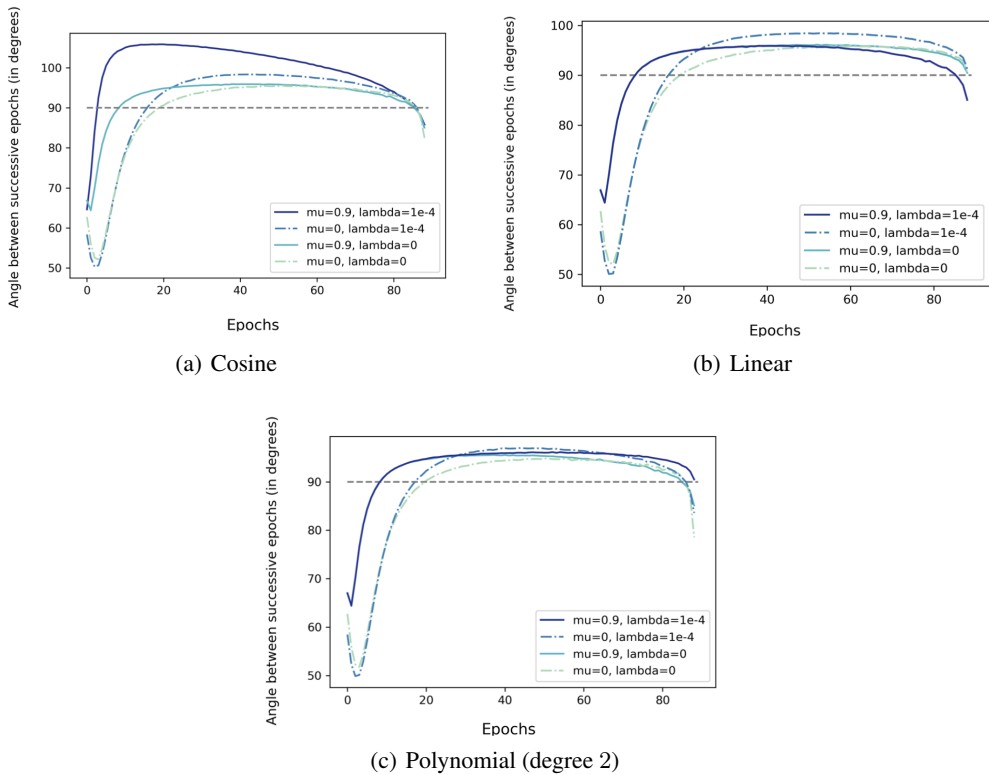

(a) Cosine        (b) Linear

(c) Polynomial (degree 2)

Figure 82: Effect of different learning rate schedules on the angle between successive epochs. We find that as in Figure 4(a), a significant part of the training shows obtuse angles between the updates (as marked by the part above the dashed gray lines), and which tend to be higher in the setting of having both momentum and weight decay, like seen previously.

## L EFFECT OF TRAJECTORY GRANULARITY

### L.1 TRAJECTORY MAPS WITH FINER GRANULARITY

We run our prototypical ResNet50 experiments at $2\times$ and $4\times$ granularities, each of which amounts to having 2 and 4 evenly spaced checkpoints for every epoch. This results in an overall (including initialization) 181 and 361 checkpoints, as opposed to 91 checkpoints considered previously. *The below-mentioned comparisons show that our results at the granularity of $1\times$ are produce highly similar trends as seen with the finer granularity.*

The above holds both for the visualizations of trajectory maps as well as angular hallmarks such as angle between successive checkpoints.

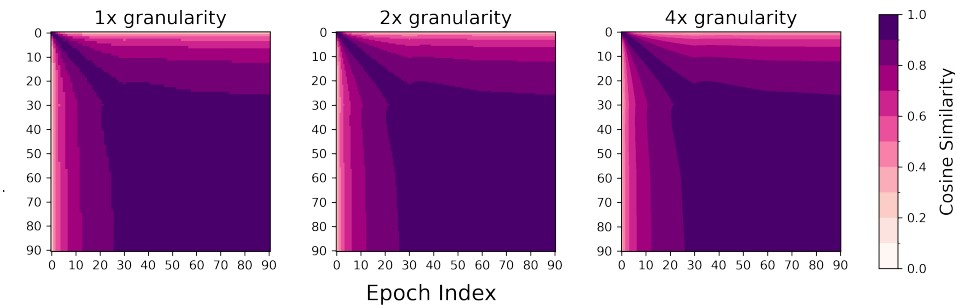

(a) Trajectory Maps with 10 color levels

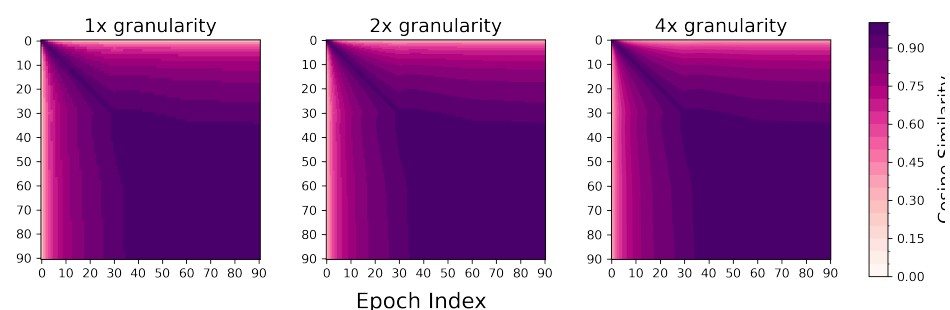

(b) Trajectory Maps with 20 color levels

Figure 83: Visualization of different learning rate schedules and the accompanying train and test loss curves (in the non-warmup case of Figure 80).

### L.2 ANGLE BETWEEN SUCCESSIVE EPOCHS

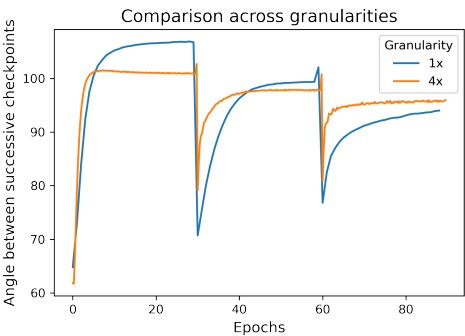

Figure 84: Direct comparison of building trajectory maps that are at $(4\times)$ finer granularity than previously used. Hyperparameter settings are identical between the two, i.e., momentum is $0.9$ and weight decay is $0.0001$.

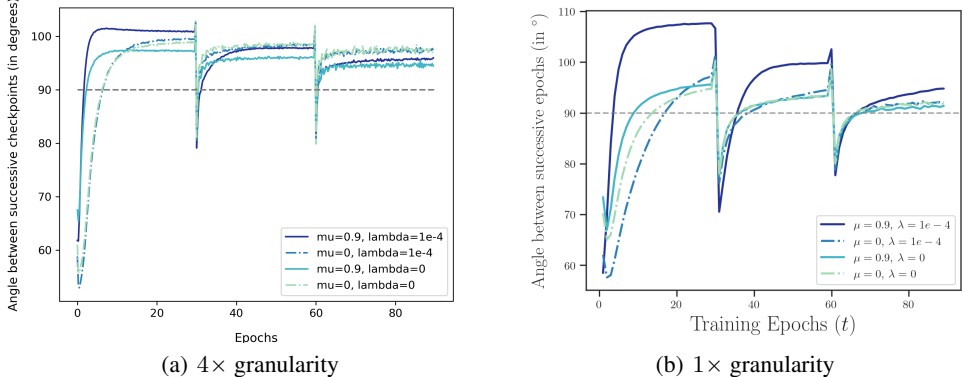

(a) $4\times$ granularity

(b) $1\times$ granularity

Figure 85: Effects of Momentum and Weight Decay at different granularities. The right figure is the same as the one in the main text. $1\times$ granularity refers to a checkpoint every epoch; while $4\times$ granularity would mean a checkpoint every quarter of an epoch.

## L.3 FINE-GRAINED VIEW INTO THE FIRST EPOCH

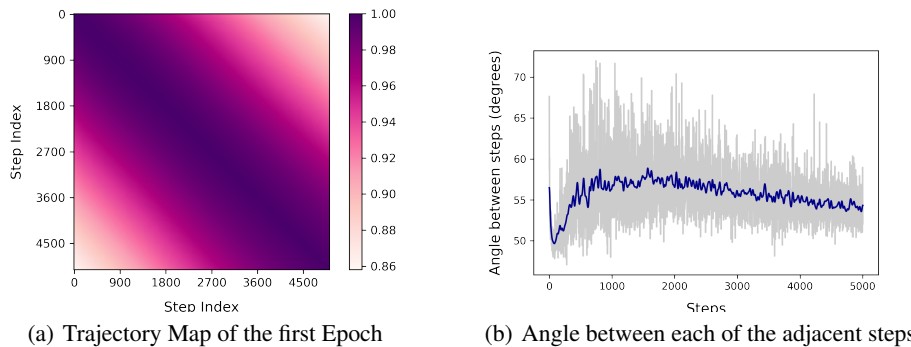

(a) Trajectory Map of the first Epoch

(b) Angle between each of the adjacent steps

Figure 86: Trajectory map and Angle between checkpoints take at fine intervals during the first epoch. The first epoch contains $5004$ steps, and so for the left subfigure, we consider checkpoints after every $18$ steps which suffices here since still finer granularity will make the already high resolution only marginally more high resolution. Hence the trajectory map comprises of a total of $279$ checkpoints (including the initialization). As far as the right subfigure is concerned, here **we process every single step** to yield a detailed view of the steps in the first epoch. The particular setting shown are for our prototypical setting with momentum $0.9$ and weight decay $0.0001$.

## M ANGLE BETWEEN SUCCESSIVE EPOCHS FOR VERY LARGE BATCH SIZE TRAINING

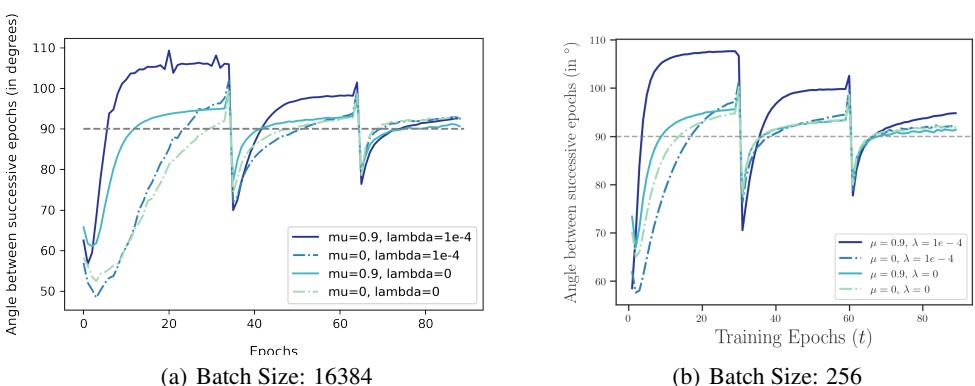

(a) Batch Size: 16384

(b) Batch Size: 256

Figure 87: Angle between successive epochs for networks trained with different batch sizes.

