# OpenReview forum: "The Directionality of Optimization Trajectories in Neural Networks"
_ICLR.cc/2025/Conference — ICLR 2025 Poster_

### Official Review · Reviewer_BhS2 · 2024-10-21

**Soundness:** 2
**Presentation:** 3
**Contribution:** 2
**Rating:** 3
**Confidence:** 2

**Summary:**

This paper introduces the concept of the trajectory map to study the dynamics of neural network training. Specifically, the trajectory map is a matrix where each element in $i$-th row and $j$-th column represents the cosine similarity between the model parameters at $i$-th iteration and the model parameters at $j$-th iteration.
Through visualization and quantitative analysis, the study uncovers patterns such as directional redundancy as reflected in the trajectory map. Additionally, the empirical influence of momentum and weight decay on the trajectory map is verified. Finally, the paper suggests utilizing directional redundancy to enhance the efficiency of neural network training.

**Strengths:**

(1) This work provides a relatively thorough validation of the proposed trajectory map concept, specifically for analyzing the training trajectory. The concept of the trajectory map is new and transforms extremely high-dimensional information into a more manageable and trackable form, making it suitable for visualization and enabling the meaningful application of statistical tools for further insights.

(2) The identified observations and corresponding patterns may offer valuable insights into the dynamics of neural network training, potentially revealing useful information about the optimization process and the network’s behavior during different phases of training.

**Weaknesses:**

(1) The introduction of customized properties in this work leads to findings relevant within those tailored domains. This work introduces new aspects of neural network training but does not build strong connections with the existing understanding of the optimization of neural networks. Thus, the new definitions and results in this work are less useful in motivating the development of new optimization methods or helping understand current optimization methods.

(2) The primary finding of directional redundancy is questionable. First, projecting high-dimensional information into lower-dimensional space may result in significant information loss, meaning that redundancy observed in low-dimensional representations may not necessarily imply that iterations during the later stages of training are redundant or unnecessary. Thus, the leveraging of directional redundancy (in section 6) lacks concrete theoretical support and sufficient assessment.

(3) Section 6 offers a relatively weak argument for the practical application of directional redundancy, specifically the claim that "once directional saturation is achieved, the full capacity of a network might no longer be required for further training." While the image on the right side of Figure 8 presents straightforward patterns, these patterns do not clearly align with the left side of the figure, and they seem more contingent on the percentage of iterations

**Questions:**

(1) What are the shared and outlined observations between Figure 2 and Figure 6? Despite the influence introduced by the learning rate schedule in Figure 6, for points along the diagonal, the cosine similarity exhibits a monotonically decreasing trend in both the horizontal and vertical directions, generally declining from 1 to 0 as observed from the upper-left point of the diagonal. While this observation is straightforward but introduces nothing new.

(2) Line383 to line388 and corresponding Figure 7.

(2.A) I believe that the observed trends are linked to variations in model size rather than performance across different metrics.
(2.B) However, I am unclear on the significance of this comparison beyond the linear relationship between the apex angle and model size. Could you elaborate further on the insights provided by Figure 7.a in this regard?
Furthermore, Figures 7.b and 7.c "fail to capture the expected monotonic trend," but the implications of this "failure" are unclear, other than that these metrics cannot be used as reliable predictors for model size.

(3) The experiments with large language models (LLMs) in Section 5 utilize Adam as the optimizer instead of SGD. Could you provide further insights on how the choice of different optimizers, such as Adam versus SGD, influences the training process and outcomes, particularly in relation to the dynamics of adaptive learning?

(4) Section 5 key takeaways (line398 - line402).

(4.A) What does the complexity measure refer to in this context? Is it related to the complexity of the neural loss landscapes, the architecture of the model, or something else? Could you clarify how this measure is utilized to assess the optimization process?
(4.B) While I agree that the complexity metric reveals a correlation with model size, claims regarding other correlations seem unsupported or underexplored in the current analysis.

---

> ### Author Response · Authors · 2024-11-28
>
> We would like to thank you for your feedback and for voicing your concerns. We understand that you see the promise of our “relatively thorough validation” that can help towards “useful information about the optimization process and the network’s behavior”. There is merit in some of the raised questions which we seek to address below, and we would also like to offer clarifications at other places so that we are on the same page. We hope you will allow us the chance.
>
> &nbsp;
>
> > “connections with the existing understanding of the optimization of neural network”
>
> We don’t think this is a fair assessment of our work.
>   - We draw our connections to the recently uncovered Edge-of-Stability (Cohen et al, 21)  phenomenon, and adding evidence to its interplay with momentum and weight decay.
>   - We also showcase how many of the optimization hyperparameters have choices well and beyond their sole intended purpose (Andruishchenko et. al., 2023; Zhang et al. 2023. This can be seen clearly by the directional exploration encouraged by both momentum and weight decay.
> - Our work also ties in intimately with the emerging paradigm of implicit regularization (Dherin et al., 2021; Gunasekar et. al., 2018), but from the perspective of directionality.
>
> &nbsp;
>
>
>  > “projecting high-dimensional information into lower-dimensional space may result in significant information loss”
>
> Trajectory maps, as you have mentioned yourself, transform extremely high-dimensional information into a manageable form. However, there is nothing that prevents us from honing into some specific parts of this information, when needed.
>
>   - Namely, we already allude in the main text and also discuss in detail in Appendix H that trajectory maps can also be defined at the level of layers or groups of parameters. So one would track simultaneously multiple trajectory maps that each focus on certain subsets of the parameter space.
>
>   - This can indeed over some additional insights or finer details beyond those available in the pan-network trajectory map, as evident in Figure 74. Gathering information about fringe parameters (like biases) in the pan-network trajectory map is not the most suitable place.
>
> *Importantly, however, the trajectory map successfully represents the shared information across the majority of parameter groups, as is evident in Figure 73 (as well as Figure 75). Thus we can be confident that the crucial information is being retained even in the simple pan-network trajectory maps.*
>
> &nbsp;
>
>
> > “redundancy observed in low-dimensional representations may not necessarily imply that iterations during the later stages of training are redundant or unnecessary”
>
> Yes, and neither do we ever claim that late-stage training is unnecessary or trivially redundant. This is also precisely why in Section 4, under Question 1 Initial Thoughts, we explicitly say that “A tempting first guess is that perhaps the trajectories are just linear in this phase. But, if that were the case, we would be just using line-search”.
>
>    - We dedicate the whole of Sections 4.1 and 4.2 to understand the nature of directional movement that occurs in this late-stage. In particular, we explain that a lot of the directional movement occurs in the form of oscillations and noise. And, of course, one cannot just omit this late phase.
>
>   - However, our crucial point in leveraging redundancy relies on the fact that the overall trajectory direction (as seen from the origin) is largely maintained after some point in training (which is what we call directional saturation). And moreover, certain interventions, such as only training the normalization layers, preserve this macro-level trajectory direction, as is evident from our experiments in Section  6.
>
> *In brief, we are saying that there is redundancy inherent at the macro-scale which can be utilized to improve the late-stage training process.*
>
> &nbsp;
>
>
> > “While the image on the right side of Figure 8 presents straightforward patterns, these patterns do not clearly align with the left side of the figure”
>
> The left subfigure is indeed not the cleanest figure. We now provide in Figure 78, a more clear-cut demonstration of this fact.

---

> ### Author Response · Authors · 2024-11-28
>
> &nbsp;
>
> ### Questions:
>
> > “cosine similarity exhibits a monotonically decreasing trend in both the horizontal and vertical directions …. introduces nothing new”
>
> Our point is surely not to just say cosine similarity is monotonically decreasing. Rather, the information lies in being able to gauge how fast or slow this decrease in cosine similarity is.
>
> And indeed, then trajectory maps are not just about a single row or column, but rather how it captures the information about the pairwise inter-relations of each of the points in the trajectory.
>
> &nbsp;
>
> > “the observed trends are linked to variations in model size rather than performance across different metrics, …. Could you elaborate further on the insights provided by Figure 7.a in this regard? figures 7.b and 7.c "fail to capture the expected monotonic trend ”
>
> - Many of the other angular and norm-based metrics could also be expected to scale with model size, but their trends with increasing scale are non-monotonic. This is in opposition to the apex angle at the peak which has a clean monotonic trend.
>
> - Even if it is just a more reliable indicator of model size, there is also value in that.
>
> - And, yes these figures also serve to show that some of the most natural complexity measures and which are also related to the extent of feature learning (distance from initialization) do not work as well in this setting — which is not entirely evident from prior work
>
> &nbsp;
>
> > Adam and SGD, in regards to LLMs
>
> We would like to offer insights into this, however SGD is well-known to work worse in the LM setting. However, in our submission we do have experiments with AdamW and SGD on a shared setting of ResNet50 ImageNet classification that enables comparison across optimizers. For example, we see shared similarities across optimizers in their trajectory maps in appendices E.3 and F.2, with regards to the effect of weight decay.
>
> &nbsp;
>
>
> > “What does the complexity measure refer to in this context?”
>
> It is in the generalization sense, i.e., complexity of the function/model class.
>
> &nbsp;
>
> ----
>
> We hope the above answers the questions you had, but please let us know if there is something you would like us to further elaborate. *We'd be more than happy to address. Besides, we also hope that in light of the above response, you will give your score a second thought. Thanks!*

---

> > ### Comment · Reviewer_BhS2 · 2024-11-28
> >
> > Thank you for the response!
> >
> > After carefully reviewing all the feedback, I remain uncertain about the necessity of introducing the trajectory map to characterize optimization trajectories. This directly relates to the first identified weakness: the findings appear relevant only within those tailored domains.
> > Regarding other weaknesses, as previously mentioned, the leveraging of trajectory map such as directional redundancy lacks robust theoretical support and has not been sufficiently assessed experimentally.
> > Given these considerations, I will maintain my score but lower my confidence level to 2.

---

### Official Review · Reviewer_eAxJ · 2024-11-03

**Soundness:** 3
**Presentation:** 2
**Contribution:** 3
**Rating:** 8
**Confidence:** 4

**Summary:**

This paper flattens the neural network parameters into a vector, then analyzes the evolution of that vector over the course of training. They show across a few architectures and domains that there is large directional stability, affected by hyperparameter choices. They also note that such stability can be leveraged to finetune fewer parameters, in keeping with prior literature.

**Strengths:**

- The choice of quantity to analyze is quite simple, and it's nice to have a fleshed out analysis of such a quantity
- It was a good idea to include the analysis of an additional domain (Pythia) so as to make the arguments firmer beyond the overparametrized setting (can get training accuracy to 0) that happens in image classification

**Weaknesses:**

## Overall

1. I think the motivation for why to study this quantity in the introduction is a bit sparse. As someone who works in the field I think this is an interesting quantity to look at, but I think a reader from elsewhere may not feel similarly and the paper does not make that easier. For a work on a similar topic that I think makes a better case, see https://arxiv.org/abs/2106.16004
2. We as a field obviously don't have a complete understanding of NN optimization, so I think it's a bit unfair to judge this work by saying "it's not clear why this analysis connects to things we care about" as it might be useful in the future, or for correlating with other observations, still I think the paper could make a better attempt to connect to many existing works on the topic to further the scientific dialogue. I'll try to detail this below.
3. I think the experiments could use a few more hyperparameter settings, it's a bit difficult to make claims about the importance or role of hyperparameters like momentum or weight decay with only two settings (and no sense of how they connect to performance, see next point).
4. In a paper ostensibly introducing itself about the loss landscape, it was a bit frustrating not to find a single loss curve. If the network fails to fit the loss, then that discussion should occur in concert with increased directional stability. I think it's difficult to understand these things in isolation and I wish the paper made it easier to do so.

## Specific

### Section 1
1. Implicit bias should be defined in the introduction, at least in a short sentence.
2. The introduction to the motivating question feels very fragmented.
3. "Benefits of the Trajectory View..." - I think it would be helpful to explicitly contrast this with prior metrics others have considered to make the case stronger.
4. "Decapacitation Hypothesis" - I think this is an odd name, it would imply to me that networks lose capacity, but the actual results later in the paper suggest that networks have sufficiently learned the important features such that you don't need every parameter any more, not that you never needed every parameter. Also I believe that without a discussion of prior work on this topic, giving this a new name is not good hygiene. See https://arxiv.org/abs/2312.11420

### Section 2
1. I think the motivation for why to flatten the parameters is not very clear. Especially as prior work like muP has explicitly noticed different evolution for different parameters (e.g. bias vs. matrix) given the structure of the network, it would be nice to make an explicit argument contrasting with existing work why this is a good idea. It would also be useful to discuss linear interpolation works like https://arxiv.org/abs/2106.16004 and many works cited in that paper to bolster the case.

### Section 3
1. "epochly" -> intermediate?
2. I feel that convergence in direction should probably be considered as convergence given that CE loss can never be minimized because of the exponential.
3. Training loss curves for Fig 2 + 3 are missing: is directional similarity mostly a result of the model failing to learn well at all?
4. Directional movement saturates exactly when learning rate is decayed, so it'd be nice to have a hyperparameter setting without this additional confounding factor...
5. It might be nice to rescale the trajectory maps to be on separate color scales. For example in the case with no weight decay, the increased similarity could be due to the fact that WD removes superfluous weights, but without a lot of the initialization sticks around that isn't used much for the actual prediction task. This effect could drown out any signal in the trajectory map.

### Section 4
1. The "Initial Thoughts" are confounded by the lack of training loss to correlate with the plots. It could be that the model is stuck at a local minimum.
2. Certainly in Fig. 4 the updates are oscillating, but this might be expected for (relatively) small batch sizes as the examples are likely very different. It would be interesting to know how the overall gradient oscillates.
3. In 4.2 it's very unclear to me why this is the right model to consider for NN as so much is simplified here. Certainly the stable Hessian assumption seems unfounded in the early stage of training given the trajectory differences shown in this work, without a measurement that it's stable... It's fine to try to simplify to basics but I think the presentation should be amended.
4. Also as far as I understand, EoS is for gradient descent, while here we look at practical SGD, this seems like a huge difference no?
5. Would be good to discuss related work in this section on linear interpolation, see previous links.

### Section 5
1. "Directional measures predict..." feels really too strong a claim for the existing evidence. I would want to see performance for many different hyperparameter choices vs. the apex angle. Besides there is already work on practical generalization measures (https://arxiv.org/abs/1912.02178) and it would be nice to talk about how the stronger ones there connect.

### Section 6
1. There is already work on tuning just normalization layers in the finetuning setting (https://arxiv.org/abs/2312.11420) that should be referenced and discussed here. I also don't love the naming of this phenomenon as I feel it could be stated simply as what you did.
2. It would be nice to have experiments on Pythia here as well.

**Questions:**

Please see Weaknesses.

---

> ### Author Response · Authors · 2024-11-28
>
> Thank you for your very comprehensive review. We truly appreciate this!
>
>
> Below, we address each of your comments section by section, and point by point. Some of the raised comments have already been incorporated to the submission pdf, so for brevity we’ll omit further discussion.
>
> ---
>
> ### Overall
>
> &nbsp;
>
> > “ it's a bit difficult to make claims about the importance or role of hyperparameters like momentum or weight decay with only two settings”
>
> While the main text includes only a handful of settings for the sake of readability, we would like to emphasize that our supplementary material already contains an extensive analysis of hyperparameters with multiple settings. For clarity, we briefly outline them here:
>
> - Section E.3 on Effect of Weight Decay with AdamW, ResNet50 contains 5 different settings of weight decay.
> - Section F.1: Effect of Weight Decay with AdamW, ViT contains 3 settings of weight decay. Likewise Section F.2 looking at the effect of weight decay with SGD, ResNet50 also contains 3 settings of weight decay.
> - Section F.3 on SAM also contains experiments on three sharpness coefficients.
> - Section F.4, on the effect of momentum shows three values (0.9, 0.6, and 0)
> - Section F.9 contains 4 different values of batch sizes.
>
> So it is definitely the case that we have multiple settings/values of hyperparameters. We’ll certainly make the presence of these experiments clearer in the main text.
>
> Besides, if there are some specific hyperparameters where you’d like to see more exploration, please let us know.
>
> &nbsp;
>
> > “no sense of how they connect to performance … loss curves”
>
> Thanks for this point. We have included the loss and accuracy curves/plots in **Figures 16, 22, 28, 34, 40, 46** as well as correlation of MDS with test performance in **Figures 17, 23, 29, 35, 41, 47**.
>
> &nbsp;
>
> ----
>
> ### Section 1
>
> &nbsp;
>
> > ‘Decapacitation Hypothesis’ & prior work
>
> Thanks for bringing this recent work to our attention. We now refer to it.
>
> We do want to mention while it is similar to our particular instantiation of decapacitation, it is important to note that:
> - Our work is more geared towards (pre-)training, and not as much fine-tuning. It could be argued the late-stage might be thought of as being similar to fine-tuning, but there can still be differences.
> - Also, as we mentioned, tuning only (batch/layer) normalization layers is just one particular example of how we aim to preserve the broader trajectory direction, while taking away capacity or locking in certain parameters.
>
> > Motivation from linear interpolation works
>
> We really like your idea of motivating this in the style of Vlaar & Frankle. We will add a detailed related work discussion in the appendix that covers this and other related work on the cross-sectional geometry of loss landscapes
>
> &nbsp;
>
> ----
>
> ## Section 2
> &nbsp;
>
>
> > motivation for why to flatten the parameters is not very clear.
>
> Thank you for raising this point. Our motivation for trajectory maps was to provide a method to compactly represent high-dimensional information in a more manageable form. While our current approach aggregates parameters across the network, *it is important to note that nothing fundamentally prevents us from focusing on specific subsets of parameters when needed.*
>
>  - Specifically, as we mention in the main text and elaborate further in Appendix H, trajectory maps can be defined at the level of layers or groups of parameters. This allows us to track multiple trajectory maps simultaneously, each tailored to specific subsets of the parameter space.
>
>  - Such an approach can indeed offer additional insights or capture finer details beyond those accessible through a pan-network trajectory map, as demonstrated in Figure 74. For instance, while the pan-network trajectory map provides a holistic view, it may not be the ideal tool for isolating dynamics specific to biases or other parameter types.
>
>  - Consistent with views from muP about bias, we observe that trajectory maps for bias parameters sometimes exhibit slight differences, but these differences do not appear to critically impact the overall network dynamics.
>
> *Importantly, the pan-network trajectory map effectively captures shared information across the majority of parameter groups, as shown in **Figures 73 and 75**. This indicates that crucial information about the network's dynamics is retained, even in the aggregated and compact representation.*
>
> &nbsp;
>
> ---
>
> ### Section 3
> &nbsp;
>
>    > Directional movement saturates exactly when learning rate is decayed, so it'd be nice to have a hyperparameter setting without this additional confounding factor
>
> That’s a great question. In **Appendix K**, we have additional experiments with cosine, linear, and polynomial schedulers. We show that the similar behaviour of the trajectory maps, as in similar onset of directional saturation, can also be witnessed for these other learning rate schedulers. Thus verifying that LR scheduler is not a confounding factor.

---

> ### Author Response · Authors · 2024-11-28
>
> &nbsp;
>
> ### Section 4
>
> &nbsp;
>
> > “"Initial Thoughts" are confounded by the lack of training loss”
>
> This should now be addressed by the loss curves in each of the scenarios. The particular figure of relevance here is Figure 16.
> As can be seen, the loss is slowly but surely going down.
> Also, from Figure 3b, we can see that the trajectory length is still increasing.
> The gradient norm in this setting is also non-zero (actually much bigger), so technically it won’t quite yet be at a local minimum.
>
> &nbsp;
>
>
> > “Fig. 4 the updates are oscillating, but this might be expected for (relatively) small batch sizes”
>
> That’s a valid concern. In **Appendix M**, we now showcase ImageNet experiments with a much larger batch size **16384**, while previously we had a batch size of 256. When we increase the batch size, we also increase the learning rate by the same linear factor and warm it up in the first 5 epochs (as suggested in Goyal et al. 2017).
>
> We can draw the following observations about Figure 87:
> - Figure 87 (a) and 87 (b) match quite closely, despite the significant differences in batch size.
> - In both cases, we see obtuse angles between updates for a significant part of training, and also that the presence of momentum and weight decay leads to still bigger angles.
>
> This shows that the oscillation also occurs at large batch sizes.
>
> &nbsp;
>
> > “Certainly the stable Hessian assumption seems unfounded in the early stage of training”
>
> We would like to clarify ourselves.
>
> - We are not claiming the onset of obtuse angles from the very beginning of training, but after some time into training (like around 5-10 epochs in Figure 4a). Thus, we are more near the vicinity of EoS.
>
> - Besides, historically a lot of intuition in NNs has been gained from toy models given complexity. E.g. noisy quadratic model (Zhang et al. 2019, https://arxiv.org/abs/1907.04164) predicts batch size-LR scaling. Even infinite width (Jacot et al. 2018) is "simplified" but provides a lot of intuition for practical NN training dynamics
>
> &nbsp;
>
> > “EoS is for gradient descent … no?”
>
> Past work [1] and similar show that it is possible to extend EoS to mini-batch gradient setting, but with a slightly generalized EoS criterion that defaults to 2/\eta \approx \lambda_max in full-gradient setting.
>
> [1] Lee & Jang et al., ICLR 2023: A new characterization of the edge of stability based on a sharpness measure aware of batch gradient distribution
>
>
> &nbsp;
>
> ----
>
> &nbsp;
>
> ### Section 5
>
> &nbsp;
>
> > Directional measures predict.
>
> We have switched to “correlate with”, but are also happy to tone it in a different way which you prefer.
>
> &nbsp;
>
> > Besides there is already work on practical generalization measures (https://arxiv.org/abs/1912.02178) and it would be nice to talk about how the stronger ones there connect
>
> We appreciate the observation regarding existing work on practical generalization measures (https://arxiv.org/abs/1912.02178) and agree that it would be valuable to discuss how the stronger measures identified there relate to our findings. We will incorporate this discussion in the revised version.
>
> - That said, we wish to clarify that our focus on the efficacy of the apex angle at the origin is primarily a side observation rather than a core claim or contribution of our work.
>
> - Furthermore, we would like to highlight that many of the generalization measures explored in prior work have not been thoroughly studied in the context of language modeling.
> - Among these, the sharpness-based measures, which perform particularly well in the cited work, are not data-free. In contrast, the quantities we consider here, while data-dependent in nature, are effectively data-free in practice.
>
> &nbsp;
>
> ----
>
> &nbsp;
>
> ### Section 6
>
> &nbsp;
>
>
> > It would be nice to have experiments on Pythia here as well.
>
> While we would love to pursue this, conducting training experiments with Pythia checkpoints would require a substantial compute budget, which is unfortunately beyond our current resources.
> That said, we believe this line of inquiry is an exciting and valuable offshoot that deserves a dedicated and thorough investigation. The promising results we have obtained on CIFAR-10 and ImageNet, combined with the generalizable nature of trajectory maps, make extending this work to Pythia a compelling direction for future research.
>
> &nbsp;
>
> ----
>
> Thank you once again for your thoughtful suggestions. We hope we have been able to address your remaining concerns. Should you have more questions or concerns, we remain at your disposal.

---

> ### Comment · Reviewer_eAxJ · 2024-11-30
> **Thank you for your response (1/2)**
>
> Thank you for the response. I appreciate the additional experiments and clarifications. I'll go point by point to address the rebuttals.
>
> ## Overall
>
> > it's a bit difficult to make claims...
>
> Thank you for pointing me to these results. I believe they should be highlighted in the main text, though I completely understand that it is impossible to include them all due to space constraints.
>
> > no sense of how they connect to performance...
>
> Thank you for providing the loss/accuracy curves. I feel that these help in interpreting the current set of results. However, I don't see them mentioned in the text at all. It would be helpful to reference (perhaps in the figure captions), so as to make reasoning easier.
>
> ## Section 1
>
> > Decapacitation Hypothesis & prior work
>
> I don't believe that late-stage finetuning is spiritually different than pretraining -> finetuning, given lots of prior results show similar properties in both settings (LMC as a big one [1, 2]). If the claim is that tuning only normalization layers is just one instantiation, then it is necessary to provide other instantiations. I don't think there's anything in this rebuttal that has convinced me it is worthwhile to give this phenomenon a new name and I believe the original criticism still stands.
>
> - [1] https://proceedings.mlr.press/v119/frankle20a
> - [2] https://arxiv.org/abs/2008.11687
>
> > Motivation from linear interpolation works
>
> The entire point of this was to make the *introduction* more clear to the reader. I was not providing this to pad the related work far from where readers would see it, but rather to provide a model introduction that I enjoyed reading on how to motivate such fundamental scientific work. I do appreciate the additional changes providing definition for implicit bias.
>
> ## Section 2
>
> > motivation for why to flatten the parameters is not very clear
>
> Two points:
> 1. certainly nothing prevents considering individual parameters as vectors, but that is not what the majority of the experiments are about, so I don't think this is a reasonable line of arugment
> 2. even when flattening individual parameters, the point of this criticism was to ask *why* it makes sense to flatten e.g. convolutional tensors, when the natural linear-algebraic operation is happening in more dimensions. Without backing out a connection from the vector to the structure, it's unclear to me why this is a good idea.
>
> I still maintain that this motivation is quite unclear, and I don't think the paper has yet addressed this. The goal is not to change the results or provide restricted results, the goal is to justify this choice, instead of taking it as a given.
>
> ## Section 3
>
> > Directional movement saturates exactly when learning rate is decayed, so it'd be nice to have a hyperparameter setting without this additional confounding factor
>
> Thank you for the additional results. This helps clarify the original experiments.
>
> I'm also not sure if the additional concerns here were cut off in the initial response, or what happened.
>
> ## Section 4
>
> > "Initial Thoughts" are confounded by the lack of training loss
>
> It would be good to reference the training loss figures at the site of this confusion. Otherwise I can see that it would be similarly confusing.
>
> > Fig. 4 the updates are oscillating, but this might be expected for (relatively) small batch sizes
>
> Thank you for the additional experiment. I'm not sure this really addresses the original concern. The point of the original concern was that what oscillation may really be tracking is that each individual update could be directionally defined according to the few examples that comprise it. ImageNet has a much larger number of examples than 16k, so I'm not sure how close this behavior is to gradient descent, or if it's still (relatively) small. The natural control to this is to look at the gradient direction after each update, or to take an estimate of the gradient, but for a fixed sample of examples that is *shared across all timesteps*. Without this control, I'm not sure that the concern is answered.
>
> > Certainly the stable Hessian assumption seems unfounded in the early stage of training
>
> Thank you for clarifying that the theory is intended to address late-stage. It would be helpful to modify the text near the statement of Lemma 1 to make this explicit.
>
> Also I don't want to open this can of worms, but I'm not sure that I buy that infinite-width NNs provide "a lot of intuition for practical NN training dynamics." They do not perform like training, and they assume something that even this work shows is false: dynamics are not linear around the origin.
>
> > EoS is for gradient descent... no?
>
> Thank you for the additional reference.

---

> ### Comment · Reviewer_eAxJ · 2024-11-30
> **Thank you for your response (2/2)**
>
> Continuing the response to the rebuttal below.
>
> ## Section 5
>
> > Directional measures predict
>
> Thank you, I think this is more appropriate.
>
> > Besides there is already work on practical generalization measures...
>
> Regardless of whether it is a side observation or not, there is a long literature on generalization measures. Presenting a new one without any context is inappropriate. It is fine not to comprehensively compare, but I think this should be couched in the understanding that this may or may not be competitive as a generalization measure, and it is not currently presented that way.
>
> Also, there are measures besides sharpness which perform well in that work, and there are certainly other data-free generalization measures. Without a comparison on this paper's settings it is impossible for me to judge how reasonable the proposed one is.

---

> ### Author Response · Authors · 2024-12-04
>
> Thanks for your response. To answer pointwise:
>
> &nbsp;
>
> ### Section 1
>
> > give this phenomenon a new name
>
> We understand your concern. We'll remove this naming, and reframe the discussion without it.
>
> &nbsp;
>
> > Motivation from linear interpolation
>
> Yes, as mentioned before, we like this idea and do want the motivation to appeal to a wider readership. We'll incorporate the elements of their motivation in the camera ready.
>
>
> &nbsp;
>
> ### Section 2
>
> > why it makes sense to flatten
>
> Thanks for further explaining your question. There are theoretical and empirical reasons why flattening makes sense:
>
> - Mathematically, in a finite-dimensional vector spaces,  all norms are equivalent in the sense that they define the same topology. Please see https://en.wikipedia.org/wiki/Norm_(mathematics)  or https://kconrad.math.uconn.edu/blurbs/gradnumthy/equivnorms.pdf
>
> - Empirically, the usage of flattening parameters has successful precedents in optimization, and even in the linear interpolation literature. Optimization convergence analyses in deep learning default to working with flattened vectors as parameters, e.g., see AdaGrad https://www.jmlr.org/papers/volume12/duchi11a/duchi11a.pdf and they are directly lifted as such in applications. Also, more recently in the line of work on schedule-free methods by Defazio et al., like in D-Adaptation https://arxiv.org/pdf/2301.07733 (ICML'23, outstanding paper award) directly estimates the size of the domain $\|\theta_0 - \theta_\ast\|$ based on flattened calculations.
>
> Also, in the weight-matching methods used in the extended linear interpolation literature, such as for model merging modulo permutations (https://arxiv.org/pdf/2209.04836, https://arxiv.org/abs/1910.05653), it is a common practice to flatten convolutional tensors into vectors for weight matching.
>
> &nbsp;
>
> ### Section 3
>
> In the interest of time, we didn't touch on some concerns that we perceived to be somewhat tangential, but we can discuss them here briefly.
>
> > "epochly" -> intermediate?
>
> Has been already done.
>
> &nbsp;
>
>
> > I feel that convergence in direction should probably be considered as convergence given that CE loss can never be minimized because of the exponential.
>
> - Convergence in direction is relative to the minimum, which is unknown, so it is not trivial to ascertain.
> - What we show here as directional saturation occurs well before the interpolation regime, i.e., much before achieving zero-training error.
> - As far as post-zero training error (terminal phase), often training is still carried out as some works show that it confers benefits (such as that on the Neural Collapse https://www.pnas.org/doi/10.1073/pnas.2015509117 and related works.)
>
> With all this in mind, we're simply resorting to the standard training lengths as reported in the literature, like 90 epochs for ImageNet.
>
> &nbsp;
>
>
> > Training loss curves for Fig 2 + 3 are missing: is directional similarity mostly a result of the model failing to learn well at all?
>
> We have added these loss curves. As far as the particular question is concern, No --- as clearly evident in the training loss plots.
>
> &nbsp;
>
>
> > It might be nice to rescale the trajectory maps to be on separate color scales. For example in the case with no weight decay, the increased similarity could be due to the fact that WD removes superfluous weights, **but without a lot of the initialization sticks around that isn't used much for the actual prediction task.**
>
> This was unfortunately incomprehensible to us (see the part in bold which seems chopped off). Modifications to trajectory map visualizations would nevertheless be something to look at in future works; but we are confident that our trajectory map visualizations have been tested out in various settings and are highly robust.
>
> &nbsp;

---

> > ### Author Response · Authors · 2024-12-04
> >
> > ### Section 4
> >
> > > It would be good to reference the training loss figures at the site of this confusion
> >
> > Yes, surely that's in the plan for the camera ready. We didn't have the chance to make all such cosmetic changes given the limited time duration.
> >
> > &nbsp;
> >
> > > each individual update could be directionally defined according to the few examples that comprise it.
> >
> > To ensure there is no misunderstanding, the update under consideration is the net parameter change over one epoch. So *all the examples in the training set do get seen* to form this aggregate/net parameter change, and not "few examples".  Also, in appendix L, we consider the case when this formed over less than an epoch as well with similar results.
> >
> > > look at the gradient direction after each update, or to take an estimate of the gradient, but for a fixed sample of examples that is shared across all timesteps. Without this control, I'm not sure that the concern is answered.
> >
> > Having made the above remark, we still went ahead with the recommended experiment. We compute the full gradient, i.e., gradient aggregated over the **entire ImageNet training set** at each epoch (with gradient accumulation), and then look at the angles. The results can be found here  https://imgur.com/a/angle-between-gradients-computed-on-entire-imagenet-training-set-kz0L4FD (we overlay a smoothened curve for ease) corresponding to our prototypical setting of ResNet50 with momentum and weight decay enabled and when trained with batch size 256.
> >
> > At the full dataset level, the angles are smaller. But we have to be careful when interpreting this, as this study suggested in your feedback is an observational one, different from how optimization trajectory is actually proceeding.
> >
> > In any case, this matches your requested setting exactly and hopefully addresses your concerns.
> >
> >
> > &nbsp;
> >
> > >  It would be helpful to modify the text near the statement of Lemma 1 to make this explicit. Also I don't want to open this can of worms, ....
> >
> > Yes, and Yes.
> >
> > &nbsp;
> >
> > ### Section 5
> >
> > > Thank you, I think this is more appropriate.
> >
> > Good to know.
> >
> > &nbsp;
> >
> > >  but I think this should be couched in the understanding that this may or may not be competitive as a generalization measure, and it is not currently presented that way.
> >
> > Agree, we will present in the suggested way since anyways the focus over here is not on generalization.
> >
> >
> > &nbsp;
> >
> > ---
> >
> > *We believe with the response above we have answered all pending concerns.*

---

### Official Review · Reviewer_2KNQ · 2024-11-04

**Soundness:** 3
**Presentation:** 2
**Contribution:** 2
**Rating:** 6
**Confidence:** 3

**Summary:**

This paper introduces a novel approach to understanding neural network training trajectories by examining their directionality, particularly through the concept of a "TM: Trajectory Map." This approach sheds light on the consistency and detailed behaviors of optimization paths throughout training. The authors propose that once directional movement saturates, the model’s capacity can be effectively reduced ("decapacitation"), potentially improving computational efficiency. The study presents evidence of directional redundancy within optimization trajectories across both image classification tasks and large language models.

**Strengths:**

- Novelty: The Trajectory Map provides a fresh perspective on neural network training by visualizing and quantifying directional movement, potentially contributing to the development of more efficient training techniques.
- Empirical Validation: Experiments span multiple architectures and hyperparameter configurations, supporting the proposed approach.
Applicability to Efficiency: The concept of "decapacitation," where computation resources can be reduced once directionality saturates, is promising for the design of more resource-efficient training strategies.

**Weaknesses:**

- Dependency on Specific Configurations: The initial results for ResNet50 appear heavily dependent on a specific learning rate decay schedule. It remains unclear whether author's findings can be replicated across different learning rate schedules (e.g., constant, cosine decay), suggesting additional experiments would be valuable to verify generalizability.
- Correlation Between MDS and Performance: The proposed Mean Directional Similarity (MDS) metric lacks clear quantitative correlation with test accuracy or loss, which weakens the evidence of its effectiveness. Further analysis to clarify this relationship would strengthen the contribution.
- See questions section.

**Questions:**

- Details on Accuracy and Loss: Could specific values for train loss, test loss, and accuracy be provided for each experiment?
- Correlation Between Mean Directional Similarity (MDS) and Performance: What is the observed correlation between MDS and test accuracy or loss? Detailing this would clarify the effectiveness of MDS as a performance predictor.
- Learning Rate Schedule Diversity: In the appendix, including a comparison of Trajectory Maps under different learning rate schedules (e.g., constant, cosine decay, step decay) would be beneficial, as the results suggest that Trajectory Maps may vary depending on the schedule used.

#### Comments
- Missing Figure References: Figures referenced on lines L1090, L1120, and L1126 are absent.

---

> ### Author Response · Authors · 2024-11-28
>
> Thanks for your concrete and helpful feedback. We’re pleased to hear that you found our work sheds light on the consistency and detailed behaviour of optimization paths, as well as the supporting empirical results to be convincing.
>
> We understand the raised concerns. Please allow us the opportunity to address them in detail below.
>
> &nbsp;
>
> ### Learning Rate Schedule Diversity:
>
> In brief, we have added new experiments with cosine, linear, and polynomial schedules in Appendix K.
>
> - Overall, we find that the broader nature of the trajectory map as well as the angular metrics is highly similar across these schedules (barring minor variations).
>
> - Hence, our proposed approach reflects the underlying characteristics of optimization and is not overly sensitive to the learning rate scheduling.
>
> &nbsp;
>
>
> ### Correlation of Mean Directional Similarity with Performance:
>
> That’s a valid question.  We have now added the correlation of MDS (as well as a relativized version of it) with test accuracy in all the ImageNet settings. **These plots being, namely, Figure 17, 23, 29, 35, 41, 47.**
>
> **Observations:**
>
>   - We do find that MDS numbers correlate well with performance when models are of the same size and the hyperparameters are varied. In particular, as MDS decreases, and the over-reliance on a specific direction (or set of them) decreases, and therefore the performance increases. Hence, in most of these figures we find a strong negative correlation.
>
>   - On average across these settings, MDS obtains a Pearson correlation of $-0.71$. The magnitude of this average Pearson correlation coefficient further increases to $-0.90$ if we do not consider a couple of outlying cases where the network underfits severely due to high regularization in these cases. Thus, it demonstrates that MDS can serve as a promising indicator of performance.
>
>   - However, we believe further refinements of MDS could potentially also enable comparison across networks of different sizes. Presently, MDS as a notion depends on the loss landscape and trajectories that can be taken within it, thus it is better suited to comparing performance/generalization for a given sized network.
>
> Nevertheless, the full-extent to which MDS and other angular notions (such as promising ones like the apex angle at origin) can serve as a performance predictor deserves a separate paper of its own. But given these initial results, this would form an excellent future direction.
>
> &nbsp;
>
>
> ### Others:
>  - Accuracy and Loss Numbers have now been provided. In particular, we have provided loss and accuracy curves, as well as exact final test accuracy for all ImageNet experiments are available in the figure captions. **Please see Figures 16, 22, 28, 34, 40, 46** for more.
>
> - References have been fixed, thank you for pointing these out.
>
>
> &nbsp;
>
> ----
>
> *We hope that we have been able to address all your pending concerns. But please let us know, in case you still have some other questions or comments. Besides, in light of the above response,  it would be great if you could reconsider your score. Thank you!*

---

> > ### Comment · Reviewer_2KNQ · 2024-11-28
> > **Official Comment by Reviewer 2KNQ**
> >
> > I thank the authors for their detailed response. I am satisfied with the answers provided, so I will raise my score accordingly.

---

### Official Review · Reviewer_idmu · 2024-11-04

**Soundness:** 4
**Presentation:** 4
**Contribution:** 4
**Rating:** 10
**Confidence:** 2

**Summary:**

The paper presents several tools for visualizing, quantifying, and comparing neural network training trajectories. The optimization paths of neural networks are represented by a Trajectory Map (TM), which consists of the cosine similarities between parameters in checkpoint pairs of a neural network over training time. The angle between adjacent samples of the trajectory is also evaluated. Theoretically, this angle is found to be bounded by the Hessian eigenvalues so that in most training settings it is either obtuse (negative cosine) or orthogonal (0 cosine). Unlike parameter distances, the angular similarity of parameters with their initializations are also found to reduce as models get larger - a result which is used to predict relative performance between models. Finally, the "decapitation hypothesis" is introduced with supporting empirical evidence, which hypothesizes that the majority of parameters can be frozen without impacting performance, once a network's training trajectory becomes fixed in cosine similarity relative to the initialization.

**Strengths:**

Originality: the paper neatly ties together several previous works related to phases and trajectories of optimization. Although some of the findings confirm results from other work (e.g. larger networks are closer to initialization), other findings are novel and surprising (e.g. momentum and weight decay increasing rather than decreasing variability in the optimization paths).

Quality: the paper is overall very thorough and addresses almost all of the pertinent concerns that a reader may raise (see Weaknesses for issues). Although the decapitation hypothesis is not very well elaborated on, this is fine since the paper already covers a large amount of ground.

Clarity: the paper is very clearly laid out and neatly presented, with helpful color choices to enhance the figures and accompanying text. The writing has a succinct, engaging style, and is enjoyable to read. Having said that, there is some excess complexity in some sentence structures (e.g. contribution 1) and vocabulary choices (e.g. "epochly") which are difficult to understand.

Significance: the paper gives a nice disambiguation between distance travelled in parameter space during training, versus the straightness of the trajectory. The proposed measure of directionality in section 5 seems to be a useful metric for model capacity and training progress. Finally, the decapitation hypothesis does open up a lot of possible methodological work in the future, and potentially a new perspective on existing empirical methods for reducing training costs (e.g. LoRA).

**Weaknesses:**

Appendix C is important in that it looks at the baseline where the trajectory is a random walk, since naturally one would expect trajectories to become more similar over time due to learning rate decay and having travelled further from the origin. In general, it would be more informative if *all similarity values, including in the main text* were reported as increases/decreases over the random walk baseline (with decaying step size), and plots showed such relative differences instead of absolute similarities. Specifically:
- It is not obvious from figures 9-10 how much added similarity/dissimilarity is in training trajectories versus random walks - for instance, is the random walk with decay more dissimilar than training trajectories at all points in time, or do they invert in order of similarity at some point?
- What is the non-relative TM for random walks (to compare with figure 2)?
- Similiarly, what are the expected MDSs of random walks for a given step size schedule?
- Are real (empirically determined) step sizes used to derive/simulate the random walk TMs in figure 9? This seems to be the fairest method, and thus it would also be necessary to consider all combinations of momentum/weight decay and their effect on step size.

The per-epoch sampling has an unclear interaction with the analyses of adjacent steps (i.e. figure 4), because the choice of how often to sample the network could change the measured results. For example, it could be that no momentum/no weight decay actually has larger angles when measured between successive iterations, but the sum of such steps over the epochs results in smaller angles. For the sake of thoroughness, it would be nice to know how results such as figure 4a depend on the sampling rate.

**Questions:**

On a related note to the aforementioned weakness, neural networks are known to be much less stable during the early training period. What does the TM and angles between adjacent samples look like when sampled at a higher frequency in the first few epochs (e.g. every iteration of epoch 1)?

How much of the observations related to phases of training (figure 2 left, figure 4) are due to discontinuities in the learning rate schedule? For instance, what if a linear decay or cosine annealing schedule is used instead?

There is a similarity between the TM and notions of representational similarity which also consider distances and angles among the cosine similarities of activations (e.g. [1] Lange et al. 2023) . Could the authors discuss if either the methods of representational similarity could be applied to TM, or if there is an empirical or theoretical connection between a network's trajectory in parameter space vs representation space?

[1] Lange, R. D., Kwok, D., Matelsky, J. K., Wang, X., Rolnick, D., & Kording, K. (2023, September). Deep Networks as Paths on the Manifold of Neural Representations. In Topological, Algebraic and Geometric Learning Workshops 2023 (pp. 102-133). PMLR.

---

> ### Author Response · Authors · 2024-11-28
>
> We thank you for your thorough review and your appreciation of our work. In particular, we are grateful that we have been able to communicate the originality of our perspective as well as its significance. While we are glad to hear that we were able to address “almost all of the pertinent concerns that a reader may raise”, we would also like to iron out the last few wrinkles:
>
> &nbsp;
>
> ### Effect of sampling rate on Fig 4a (angle between checkpoints), and more broadly.
>
> This is a valid concern. To address this, in Appendix L, we present a detailed study of the effect of sampling rate, whose main findings are:
> - Firstly, at a broad level, comparing Figures 85 (a) and 85 (b), the obtuse nature of the angle between successive steps can still be seen.
> - The first 30 epochs have a very similar trend across both levels of granularities, and the setting with both momentum and weight decay enabled has on average $\sim 18^\circ$  bigger angle between successive samples in this duration of 30 epochs, than in the setting where both momentum and weight decay are absent.
> - The last 30 epochs show a slight deviation between the granularities. Here the finer granularity, lets us see that the latter setting (without momentum and weight decay) results in a slightly bigger angle albeit  $~\sim 0.85^\circ$ than in the former setting of enabling both these hyperparameters.
> - While the above observation in the last 30 epochs does echo your intuition, given this significant gap between the angular differences across these settings ($18^\circ$ vs $-0.85^\circ$), our larger finding of momentum and weight decay leading to an overall larger directional movement stands.
>
>
> Additional Comments:
> - Lastly, we also recommend you to have a look at Figure 84 where both granularities are shown in the same plot for the prototypical experimental setting of having both momentum and weight decay.
>
> &nbsp;
>
> ### Early Training Period: Trajectory Map and Angles b/w adjacent samples.
>
> Interesting remark. In Appendix L.3, we perform this suggested experiment and detail the results.
>
> **Observations:**
> - The TMap in Figure 86 (a) at this ‘high resolution’ bears a coarse-grained resemblance to the (tiny) 5x5 grid at the top-left of Figure 2 (left). This seems to suggest that some trajectory characteristics could repeat at different scales.
> - While TMap essentially presents an ‘aerial’ view of the trajectory (origin will be high up in the loss landscape), the angle between adjacent samples on the other hand presents a ‘ground’/surface level view of the trajectory.
> - The latter trend in Figure 86 (b) shows the trend of the angle formed between every pair adjacent updates. The raw trend is a bit noisy, fitting in with your view about the less stable nature of the early training period, so we additionally visualize a smoothened version of it.
>
> &nbsp;
> ### Effect of Different learning rate schedules.
>
> Thanks for this question. We have added new experiments with cosine, linear, and polynomial schedulers in Appendix K.
>
> - In subsection K.1, we explore the effect on trajectory maps, where we find that the phase in training where directional movement saturates is common to all these settings.
> - In subsection K.2, we study how the angle between adjacent updates changes when using these different learning rate schedulers as opposed to the stepwise learning rate scheduler used for Figure 4a. We find that while there are no stepwise transitions (which would understandably be specific to stepwise scheduler), the angles remain obtuse for a significant part of training as well as that the setting with both momentum and weight decay enabled results in a more obtuse angle, especially in the initial training phase (~20-30 epochs).
>
> *Overall, given these similarities in results, we can rest assured that our findings are general and not sensitive to the choice of learning rate schedulers.*
>
>
> &nbsp;
> ### Relation between methods of representational similarity and trajectory maps:
>
> That’s an excellent comment, and thanks for sharing this very interesting work. There are surely similarities and avenues for cross-fertilisation in both these research directions. Some interesting threads at the top of our heads:
>
> - Kernel matrices, like trajectory maps in our work, and CKA for representations are the key objects of focus in these directions, so there are definitely parallels.
> - It would be interesting to see how well trajectory-based quantities can indicate the evolution of representations for a given network, while being data-free. Of course, some interesting questions might explicitly hinge on the given inputs, but it would be worthwhile to investigate the extent of representational dependence contained within the trajectories.
> - In the opposite direction, it would make sense to explore a different, more geometry-aware, metric to compare the trajectories, than the plain Euclidean that we use here.

---

> ### Author Response · Authors · 2024-11-28
>
> ### Discussion on random walks:
>
> We have added similarity in Trajectory Maps over random walks: The added Figure 11 visualizes the difference between the trajectory map for the ResNet50 experiments and that obtained for a random walk, with the same learning rate schedule as used in practice.
> - Interestingly, the biggest difference is the further influence of the initial training phase as opposed to the random walk scenario.
>
> - We agree that this view of the trajectory map offers added insights; and we are happy to add similar plots for the rest of the experiments. In fact, we did think of this, but we chose not to do so since the random walk is a baseline model — one that makes assumptions that are perhaps not entirely accurate (e.g., isotropic covariance for the steps). Thus, as a foray into this line of work, we went for the trajectory map as it is – but, of course, it is not the definitive representation of a trajectory :)
>
> &nbsp;
>
> *Expected MDS:*
>
> That’s a great question, thanks. We have added some calculations in Appendix C.2 that do for a simple constant schedule, and whose corresponding trajectory map is the one in Figure 12.
> - This analysis in C.2  is extendible to more general schedules, such as stepwise schedules. The general idea is to compute MDS between every pair of phases through similar calculations (which can get a bit lengthy and out of scope). When the later phases employ smaller step sizes, this leads to added MDS, as is also pictorially evident in the dark grids in Trajectory maps that correspond to the later parts of training.
> - We use the precise (stepwise) learning rate scheduler used when training ResNet50, with the same hyperparameters. It would also be interesting to carry out a comparison with the exact lengths of steps, as in the grafting literature in optimization [1] in future work, as well as more broadly explore trajectory maps in this context.
>
> [1] Agarwal et. al., 2022: Learning Rate Grafting: Transferability of Optimizer Tuning

---

> > ### Comment · Reviewer_idmu · 2024-12-02
> >
> > I have read the rebuttals and comments by other reviewers.
> >
> > I find the points raised by eAxJ very pertinent, and agree that the paper suffers from some overclaiming (including proposing many new terms that may not be needed) and insufficiently strong experimental controls (e.g. my point about the random walk baseline having the same step sizes as training, and eAxJ's point about comparing gradient directions of identical batches to detect oscillation). Having said that, I think there is some confidence that these controls will not invalidate the paper's observations. I also find the results to be overall more observational in nature than predictive (in terms of generalization) which I think is a valid perspective, although the paper's language should reflect this more.
> >
> > I have kept my score but reduced my confidence.

---

> > > ### Author Response · Authors · 2024-12-04
> > >
> > > > my point about the random walk baseline having the same step sizes as training
> > >
> > > Please see https://imgur.com/a/trajectory-map-comparisons-with-different-baseline-random-walks-iUvfqyH for the results with the empirically determined, exact, stepsizes as observed in training. This is for our prototypical RN50 ImageNet setting.
> > >
> > >   - These results are slightly better at capturing the empirical trajectory maps, as you might expect
> > >   - however, after some initial epochs, they are quite similar to random walk baseline with step sizes as employed in the LR scheduler. This can be seen theoretically as the cosine similarities depend on the ratio of cumulative squared learning rates (lines 882-900)
> > >
> > > We'll add these results, as well as similar plots in other scenarios, in the paper.
> > >
> > > &nbsp;
> > >
> > > ---
> > > &nbsp;
> > >
> > > Besides, we've responded to reviewer eAxJ's concern in our latest response, as well as the experiment requested by them of identical batches. Also, we're happy to drop the name of hypothesis as decapacitation.
> > >
> > > &nbsp;
> > >
> > > ---
> > >
> > > *We hope this helps!*

---

### Author Response · Authors · 2024-11-28

We sincerely thank all the reviewers for their time, effort, and thoughtful feedback. We are delighted that the reviewers had overall a positive impression of our paper, noting it as *“very thorough and addressing almost all pertinent concerns a reader may raise,” “very clearly laid out and neatly presented”* (Reviewer **idmu**), *“a fleshed-out analysis of such a quantity”* (Reviewer **eAxJ**), *“introducing a new and transformative concept of trajectory maps”* (Reviewer **BhS2**), and *“promising for the design of more resource-efficient training strategies”* (Reviewer **2KNQ**).

&nbsp;

----


In response to the concrete questions and suggestions raised, we have revised the paper to include additional results for the many interesting ImageNet-level experiments that were asked by the reviewers as well as other analyses, which we believe have further strengthened the work. Specifically, we have made the following key additions:

&nbsp;

   - **Appendix K: Effect of Different Learning Rate Schedulers**. New experiments with cosine, linear, and polynomial schedules demonstrate the robustness of our findings across various scheduling strategies, confirming that they are not tied to the previously used stepwise schedules.

&nbsp;

  - **Appendix L: Trajectory Maps and Angular Measurements at Finer Granularity**. Our analysis at finer granularity reaffirms that per-epoch checkpoints are representative of the underlying phenomena.

&nbsp;

  - **Appendix M: Angular Measurements at Very Large Batch Sizes (16K) on ImageNet.** These experiments reveal that oscillatory movement is not limited to smaller batch sizes but persists at very large scales.

&nbsp;

  - **Appendix C.2: Additional Analysis on the Expected MDS for the Random Walk Scenario**. This analysis offers deeper insights into the presented values of Mean Displacement Scale (MDS) and enriches the theoretical understanding.

&nbsp;

- **Figures 17, 23, 29, 35, 41, 47: Evidence of strong correlation of MDS with test accuracy in all the ImageNet settings.** These findings suggest that a smaller MDS value is linked to better test performance, and shows its potential to measure performance.

&nbsp;

*These additions further validate the generality and applicability of our findings across different scenarios and conditions.*

&nbsp;

----

In addition to this general response, we have also provided detailed responses to address the specific concerns and suggestions raised by each reviewer individually.


&nbsp;

Once again, we thank the reviewers for their constructive engagement and helpful suggestions. *We remain available to address any further questions or clarifications. Thank you!*

---

### Author Response · Authors · 2024-12-04
**Final Remarks**

We thank the reviewers and the AC for their time and thoughtful engagement with our work. Before signing off, we would like to make the following remarks:

&nbsp;

1. All reviewers recognize the perspective of trajectories offered in this paper as valuable. Some more, some less, as is the norm; but the general observation holds.


2. We have conducted extensive experiments across both vision and language domains, **evaluating models with millions and billions of parameters on large-scale datasets**. The consistency of our findings across these diverse settings underlines the robustness of our work.


3. This paper represents a substantial undertaking. As a simple demonstration of this:
   - Our **appendix** has grown from (pre-rebuttal) **30 pages** to approximately **40-42 pages** with the additional experiments and plots and theoretical analyses requested during the rebuttal.
   - Although our paper is not just an empirical paper, but contains theoretical analyses as well, a quick tally of the **number of figures in the paper comes out to be ~ *240***.


&nbsp;

---
*All in all, we hope that the committee will adjudicate this work appropriately.*

---

### Meta-Review · Area_Chair_BVTP · 2024-12-21

**Metareview:**

This work investigates the behavior of optimization trajectories through the introduction of a "trajectory map" (measuring pairwise cosine similarity across the entire trajectory) and a few other complexity measures. They find that (a) well-tuned neural networks exhibit less random trajectories and (b) in the late stages of training, optimization is largely unaltered even when freezing >99% of the parameters.

Reviewers were left divided on the work: two extremely positive, one mixed, and one negative. It was generally agreed that the presented manuscript constitutes a highly novel and impressively substantial piece of work. The underlying concepts appear practically useful, and the report itself is well-written and engaging. Controversy appears to stem from some overclaiming, given the reliance on qualitative observations calls the generality of the findings into question. This seemed especially true with the introduction of new complexity measures. Balancing these aspects left reviewers with lower confidence scores than average. My opinions, especially post-discussion, reflect those of the more positive viewpoints, some of which came late in the discussion period. The authors incorporated a significant amount of reviewer feedback into their revision and discussion, addressing pitfalls that often come with empirical studies. I believe this is fascinating work, and do not see how it could be substantially improved. I recommend acceptance.

**Additional Comments On Reviewer Discussion:**

Reviewer idmu provided a highly positive assessment of the work with strong confidence, that was later lowered in the discussion period in light of comments raised by Reviewer eAxj, who initially left a more mixed response.

Reviewer eAxj commented on lack of generality, insufficient motivation, lack of training loss curves, and insufficient connections to the existing literature. The authors engaged in a lengthy discussion with Reviewer eAxj to address these concerns, including many new figures, additional appendices, and incorporating several additional references and further discussion thereof. Along with a few other minor points, the reviewer still had concerns about adherence to small/large data regimes in the ImageNet example, and how the new generalization measures were introduced without appropriate context. The authors responded in agreement that these aspects would be changed, and Reviewer eAxj increased their score substantially, acknowledging their concerns had been addressed.

Reviewer 2KNQ gave an initially skeptical score before the discussion period, acknowledging novelty, but criticizing dependence on specific experimental setups and lacking quantitative evidence for the correlation between claimed performance measures and performance itself. In response to several questions raised by Reviewer 2KNQ, the authors provided six additional figures to the revision. Reviewer 2KNQ raised their score slightly in response.

Reviewer BhS2 remained skeptical throughout the discussion period, expressing concern about the limitations of the work to tailored domains, and questioning the use of lower-dimensional representations. The authors fundamentally disagreed with these assessments, and the discussion was left at an impasse. Reviewer BhS2 reduced their confidence level after seeing other reviewer feedback.

In their rebuttal, the authors made a substantial number of changes and improvements to the working document, including three new appendices and several new figures. The authors themselves have done a nice job of summarising these inclusions in the general comments.

---

### Decision · Program_Chairs · 2025-01-22

Accept (Poster)